# Enhancing hydrovoltaic power generation through coupled heat and light-driven surface charge dynamics

Tarique Anwar & Giulia Tagliabue ⬡ ✉

Harnessing natural evaporation offers a sustainable pathway for next-generation energy technologies. We present a unified physical and experimental framework for evaporation-driven hydrovoltaic (EDHV) systems that decouples and controls the key interfacial processes underlying electricity generation from heat and sunlight. An intermediate ion-conducting layer separates the evaporative top interface from the silicon–dielectric nanopillar array, enabling independent modulation of evaporation, ion transport, and interfacial chemical equilibrium. This strategy enhances performance and clarifies mechanisms governing thermal and photo-induced charge generation, improving ion migration and electricity output. We develop a predictive equivalent-circuit model that captures process coupling through an analytically derived transfer capacitance. Our results show that capacitive photo-charging and thermally modulated surface equilibria—rather than faradaic or photothermal effects—dominate energy conversion. The device achieves 1 V open-circuit voltage and 0.25 W/m² power density, with silicon doping and dielectric choice further boosting performance. These findings inform EDHV optimization across environmental and material conditions.

Evaporation, a natural process with an average global energy flux of 80 (W/m²)[1], has significant potential for energy harvesting. Hydrovoltaic technology, which generates energy through the direct interaction of materials with water[2–4], has recently emerged as a promising avenue for sustainable energy generation[5–8]. In particular, evaporation-induced hydrovoltaic devices (EDHV) stand out due to their ability to produce a continuous (24 h) electricity output, operating without external mechanical energy input across a wide range of environmental conditions. This is achieved through the synergy of spontaneous capillary action and evaporation, enabling autonomous, low-intensity energy supply solutions ideal for portable devices, Internet of Things applications[9–11] and harvesting of low-grade waste heat (below 100 °C) produced by industrial, agricultural, and domestic processes[12,13]. When integrated with solar-driven evaporators that utilize photothermal effects[14] to significantly enhance evaporation rates, EDHV systems have the capability to also convert solar energy into electrical power. This advancement underscores their potential as a reliable and sustainable energy source. However, as the demonstration of these devices expands to encompass a wider variety of materials and architectures, it is imperative to engage in the ongoing debate surrounding the fundamental mechanisms, optimal micro/nanostructures, and operating conditions of EDHV devices. In particular, beyond enhancing the evaporation rate through photothermal effects, it is crucial to comprehend how external heat and light sources impact the interfacial processes that are essential for effective hydrovoltaic energy conversion.

In recent years, EDHV devices utilizing micro-nanoporous materials have garnered significant attention for their ability to harness energy through the movement of electrolytes within partially wetted regions[2,15,16]. This phenomenon, which occurs ahead of a liquid meniscus, facilitates the electrokinetic streaming of ions. Previous studies have identified several contributing phenomena, including the

Laboratory of Nanoscience for Energy Technologies (LNET), STI, École Polytechnique Fédérale de Lausanne (EPFL), Lausanne, Switzerland.
✉e-mail: giulia.tagliabue@epfl.ch

streaming potential[4,6,17], ionovoltaic effect[15,16,18], and evaporating potential[2,19]. Central to these mechanisms is the critical role of directional electrolyte flow near the liquid-solid interface. Additionally, going beyond the intricate micro- and nanostructures of typical EDHV devices[4,17,20,21], we recently utilized a controlled array of silicon nanopillars to demonstrate the significant impact of the geometrical and interfacial chemical properties of the nanostructures, particularly the role of the chemical equilibrium of the surface groups in enabling high salinity operation[6], challenging the reliance on deionized water for high performance[19]. Furthermore, recent work introduced a passive hydrovoltaic device characterized by a limited evaporation rate and fluid permeation, capable of producing electricity through upstream proton diffusion[22]. This proton diffusion arises from the chemical potential difference between the wet and dry sides of the material, resulting in sustained electricity generation owing to the gradual permeation of water. Despite these advancements, it remains unclear how the various mechanisms identified in different nanomaterials and device geometries can be effectively integrated into hydrovoltaic systems[21], or how to design structures that harness them simultaneously. Similarly, previous studies have indicated that combining light and heat can synergistically improve hydrovoltaic performance through enhanced photothermal evaporation[7,14,23]. On the other hand, in nanofluidic systems with charged interfaces, thermo-osmotic flows can convert thermal gradients into electrical currents[24,25]. However, one vital area remains largely overlooked: how photocharging and (photo)thermal effects contribute to ion migration, which is a key process for the performance of hydrovoltaic devices[6,14]. This lack of understanding emphasizes the importance of a fundamental study of light- and heat-driven ion dynamics, potentially revealing new methods to boost hydrovoltaic efficacy.

This work unravels the complex influence of heat and light on solid-liquid interfaces in EDHV systems, ultimately demonstrating a unified concept for EDHV architectures that transcends traditional mechanisms focused solely on ion streaming at the solid-liquid interface of the evaporating surface. By employing a top evaporating surface and a bottom uniformly structured cm-scale silicon-dielectric (core-shell) nanopillar array (SiNPs, see Fig. 1a), we report major improvements in power output under external heating and solar illumination due to ion thermodiffusion in the electrolyte layer, as well as a combination of the photovoltaic effect and thermally enhanced surface charge at the silicon-electrolyte interface. Through a combination of experiments, numerical calculations, and theoretical modelling, we reveal a mechanism in which thermally and light-assisted ion migration plays a pivotal role in enhancing electricity generation, achieving a state-of-the-art open-circuit voltage of 1 V and an output power density of 0.25 W/m² under optimal conditions with 0.1 M concentrations. Notably, these results are achieved without the need for an additional black absorbers[14,23].

Overall, our EDHV architecture introduces a device concept and design strategy that leverages thermal and photovoltaic effects to enhance interfacial processes essential for hydrovoltaic energy conversion. By structurally decoupling the top evaporating surface from the bottom nanostructured layer, we establish that the system can operate in a decoupled manner. This architectural innovation represents a significant advancement in hydrovoltaic device design, allowing full exploitation of developments in solar-driven interfacial evaporation[14,26] while simultaneously enabling the integration of optimized hydrovoltaic components.

## Introduction to key interfacial processes

Understanding the complex influence of heat and light on EDHV devices requires control over the solid-liquid interfacial properties as well as the possibility to disentangle different phenomena. In particular, as schematically represented in Fig. 1a, light and heat inputs can modify (1) the evaporation rate at the evaporating interface, (2) the ion transport within the liquid induced by chemical potential difference, and (3) the chemical equilibrium at the solid-liquid interface, which controls the surface charge. In typical EDHV devices, these effects cannot be easily decoupled as a single material serves all these functions simultaneously. Instead, we devised an EDHV architecture where the top evaporating surface, consisting of an Ag/AgCl electrode, and the hydrovoltaic component, consisting of an array of SiNPs (Fig. 1b), are spatially decoupled by an intermediate ion-conducting layer, allowing us to address these phenomena and their interplay in a controlled manner.

The SiNP's solid-liquid interface, in particular, plays a key role in the system behavior and performance due to the presence of a net surface charge[6], which can originate from both electronic and ionic contributions[27,28]. In this work, we focus on the ionic contribution arising from dissociation reactions on the surface. More specifically, as shown in Fig. 1a, upon wetting, any oxide layer on the surface of the SiNPs (Fig. 1c, d) will dissociate, usually resulting in a net negative surface charge σ (orange circles). Concurrently, positive ions will adsorb on the surface (pink circles), and an electrical double layer (EDL) develops within the liquid[29]. This results in voltage across the EDL in the liquid and space charge layer in silicon as shown in Fig. 1e. For oxides-aqueous interfaces, the surface dissociation can be quantitatively described using the complexation model[30], which depends on the surface oxide material, i.e., $SiO_2$, $Al_2O_3$, or $TiO_2$. For example, surface groups on aqueous $TiO_2$ dissociates according to: $TiOH + H_2O \Longleftrightarrow TiO^- + H_3O^+$. Mathematically, the above equilibrium reaction is governed by the equilibrium constant $K_a$, which exhibits temperature dependencies ("Methods", Eq. (5)). To a first approximation, we can then recast the reaction equilibrium condition in a form that explicit the total surface charge σ as:

$$\sigma = \frac{-e\Gamma}{1 + \frac{[H^+]_S}{K_a}} \qquad (1)$$

where Γ is the total number surface site density. As $K_a$ increases with temperature (see "Methods" and **SI 2**), we can then see that the surface charge (σ) also rises with temperature.

Importantly, Eq. (1) shows that the surface charge σ can also be varied at a constant temperature due to dynamic changes in the surface concentration of protons ($[H^+]_S$). This can occur because of any changes in the local pH, but interestingly also under irradiation. The equilibration of the Fermi level[31] across the solid-liquid interface, primarily driven by surface states, determines the band bending at the silicon-oxide interface (Fig. 1a panel iii). Consequently, under illumination, photogenerated charges (electrons or holes) accumulate at the interface[32,33]. Due to the capacitive effect of the oxide layer, a concurrent change in the surface proton concentration will occur, leading to a change in surface charge. This also highlights the importance of the oxide layer in passivating the silicon surface[34] and preventing any chemical reaction between silicon and water. Our photoelectrochemical test on the device indeed shows no evidence of faradaic activity, thereby confirming a capacitive charging, rather than faradaic, process (Figs. S3 and S4).

Overall, the chemical equilibrium at the interface is strongly dependent on the material properties as well as the temperature and the ion concentration in the EDL. Light and heat triggers can have a multi-faceted influence on the interfacial chemical equilibrium, therefore affecting device behavior and performance. With our architecture, we can specifically decouple the effect of light and heat on the interfacial chemical equilibrium, disentangling it from a purely photothermally driven evaporation enhancement.

## Experimental platform

All our devices feature a bottom cm-scale regular array of SiNPs (Fig. 1a), fabricated using a combination of nanosphere lithography

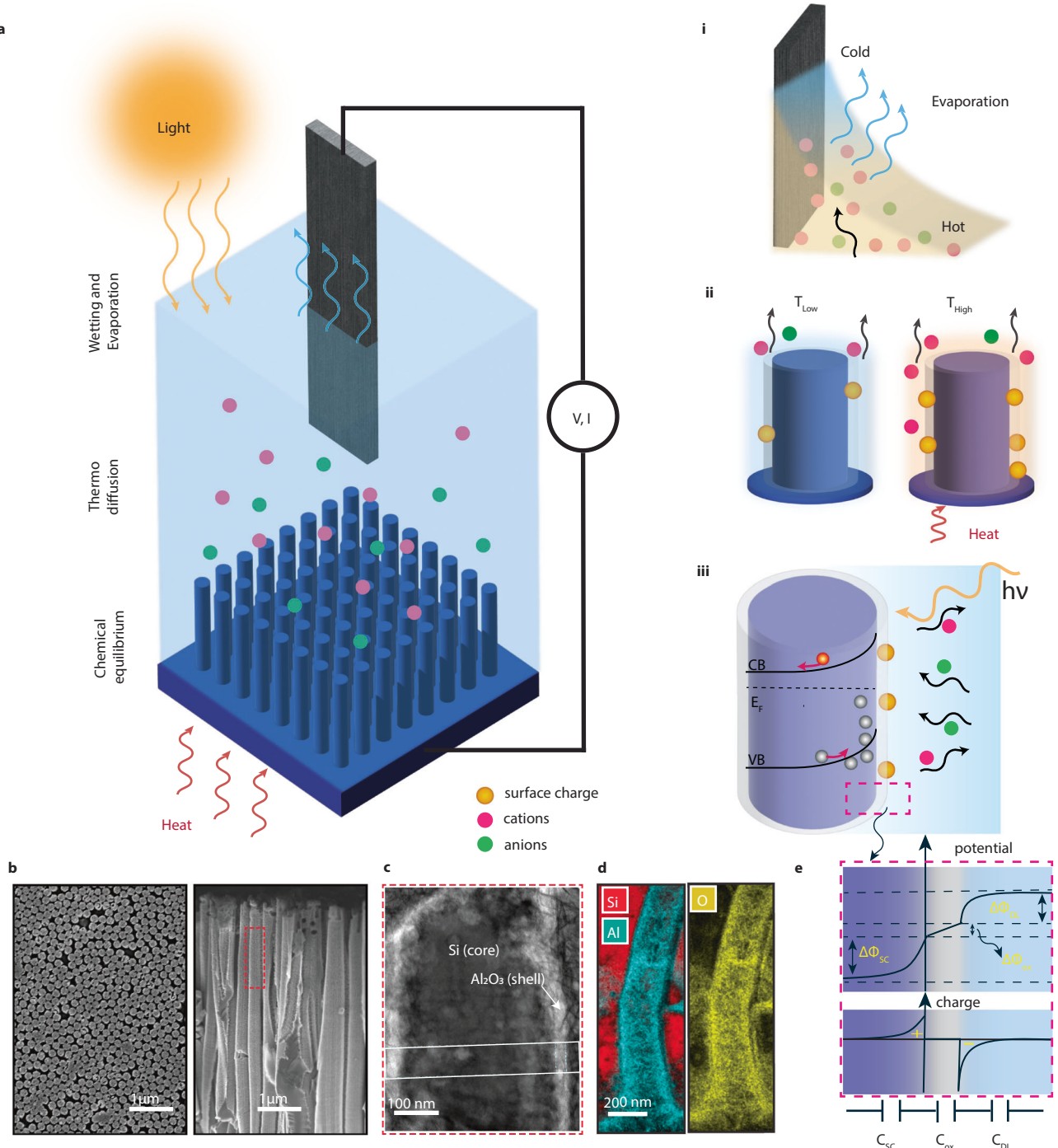

**Fig. 1 | Evaporation-driven hydrovoltaic device architecture, mechanisms, and materials. a** Schematic representation of the hydrovoltaic device featuring a top evaporating electrode surface and a bottom array of SiNPs immersed in water. The top and bottom components do not physically contact but are electrochemically connected through the water. The inset displays the three effects contributing to the device's performance. **i** A side view of the evaporating surface with the liquid meniscus and the thermal gradient across the liquid layer. **ii** An intermediate electrolyte layer and thermally tuned chemical equilibrium at the bottom nanostructures, resulting in a higher surface charge at increased temperatures. **iii** photoactive nanostructure-electrolyte interface depicting the enhanced surface charge under irradiation due to electron-hole pair generation. CB conduction band, VB valence band, $E_F$ Fermi level. **b** Scanning electron microscopy (SEM) image of the SiNPs array. (left) Top view, and (right) cross-sectional view. **c** Scanning transmission electron microscopy (STEM) image of a single NP. The cross-sectional cut of a single NP reveals the presence of a silicon core and $Al_2O_3$ shell. Intensity mapping was performed in the rectangular region (see Fig. S1). **d** TEM-EDX image of the NP displaying the elemental maps of aluminum, silicon, and oxygen. **e** (Top) A detailed view of the interface potential and free space charge in the silicon and electrolyte mediated by the oxide layer. The total potential is the sum of the potential difference in the space charge layer of silicon ($\Delta\Phi_{SC}$), the oxide layer ($\Delta\Phi_{ox}$), and the double layer ($\Delta\Phi_{DL}$) of the electrolyte (bottom), as well as the free charge profile in the respective regions and the corresponding equivalent capacitance.

and metal-assisted chemical etching of a Si wafer (see Figs. SI 4 and 1b). To passivate the surface and control its properties, we use atomic-layer deposition to coat the Si/SiO$_2$ core, where SiO$_2$ is the native oxide layer, with few-nm thick dielectric shells of Al$_2$O$_3$ or TiO$_2$ (see Figs. 1c, d and SI 4). These two materials, in fact, have distinct chemical equilibria. In addition, as any band bending at the silicon-oxide interface will be affected by the Fermi level of the semiconductor, and hence its doping, we specifically used three silicon doping, Low N-doping (1–20 Ω.cm) and high N-doping ( < 0.005 Ω.cm), and P-doped (0.1–0.5 Ω.cm), which has different space charge layer thickness and capacitance, C$_{SC}$. For hydrovoltaic testing, the sample is placed inside a custom HV cell and wetted with 250 µl of deionized water containing KCl salt of varying concentrations (from 10 µM to 0.1 M). In ambient conditions (T = 22–24 °C and humidity = 25–30%), evaporation readily occurs (**SI 5**). Our testing HV cell is uniquely designed to prevent contact between the solution and the bottom (Aluminum) electrode, which could lead to unwanted chemical reactions, and to ensure that only the central part of the silicon (1 cm$^2$ area) is in contact with the electrolyte (**SI 6**). Next, the HV cell is positioned on a microbalance to track the evaporation rate. The electrical response is measured using a top Ag/AgCl or graphite electrode (**SI 7**), placed in the liquid right above the Si NPs, and an aluminum contact previously deposited on the back surface of the Si wafer (Fig. 1a and "Methods"). During the electrical measurements, the top electrode and bottom silicon substrate are decoupled, and the electrical circuit is complete as soon as the liquid is dispensed; thereafter, voltage and current can be measured. Heat and light stimuli are applied using a Peltier cell and a Solar simulator, respectively, while measuring the open circuit voltage, V$_{OC}$, or power output of the device under different temperatures and irradiation (see "Methods" for more details).

## Results

Due to the chemical equilibrium-controlled surface charge, at the oxide-liquid interface, an EDL is formed, and any resulting imbalance in the ion distribution along the SiNP length contributes to the measured electrical potential difference. Interestingly, we previously observed that, due to the closed bottom surface of the nanochannel, the studied geometry presents an intrinsic asymmetry in the surface-charge distribution (**SI 8**). As a result, even in the absence of an evaporation-induced flow, a chemical potential difference (Φ) exists, and therefore, a V$_{OC}$ can be measured between the top electrode and the bottom SiNPs surface[6]. Φ is quantified by Eq. (2)), which depends on the temperature (T$_s$) and surface charge (σ) of the bottom silicon substrate:

$$\Phi = \frac{2k_B T_s}{e}\sinh^{-1}\left(\frac{\sigma}{\sqrt{8000\epsilon_0\epsilon_r c_0 k_B T_s}}\right) + \frac{\sigma}{C_{Stern}} \quad (2)$$

where, $k_B$, $T_s$, e, $\epsilon_0$, $\epsilon_r$, $c_0$ are Boltzmann constant, surface temperature of solid, electronic charge, dielectric permitivity of free space, relative permittivity of the medium, and molar concentration of bulk electrolyte, respectively. σ is the surface charge and $C_{Stern}$ is the capacitance of the Stern layer[35]. In the following sections, we will demonstrate how the performance metrics of the EDHV devices, specifically open circuit voltage and power density, are influenced by alterations in chemical potential under conditions of external heating and irradiation.

### Effects of temperature change on chemical equilibrium

Using a previously validated COMSOL Multiphysics model[6], which accounts for the liquid nanoconfinement and incorporates the temperature-dependent equilibrium constants for the oxide/liquid interface (see "Methods"), we calculated the electrostatic potential difference between the bulk electrolyte and bottom SiNPs as a function of temperature. We obtained a linear increase in potential difference

with an increase in temperature up to 50 K above room temperature of 298 K (**SI 9**). Therefore, we first conducted a series of experiments explicitly analyzing the open circuit voltage generated at different temperatures of the bottom SiNPs electrode, denoted as T$_s$. To trigger the temperature changes, we utilized an external Peltier heater placed beneath the SiNPs electrode (see "Methods" and **SI 10-I**). Concurrently, we monitored the temperatures using an infrared camera positioned above the sample.

Fig. 2a provides a time trace of the open circuit voltage of a SiNPs device with N-type low-doping silicon core nanopillars coated with Al$_2$O$_3$. At the beginning of the experiment (dry sample), the voltage was zero. As soon as the electrolyte wets the sample, we observe a gradual increase in voltage until a steady-state value is reached (0.25 V). When we activate the heater (at 125 s), the voltage experiences a noticeable rise, eventually approaching twice the initial value (0.5 V). The corresponding time-trace for T$_s$, as recorded by the infrared camera, is depicted in Fig. 2b (empty symbols). As the heater operates, the temperature steadily climbs and ultimately stabilizes around 70 °C for the maximum power applied to the heater. Once the heater is turned off, we observe an immediate drop in temperature and V$_{OC}$. Notably, the cooling rate is slower than the heating phase, which can be attributed to restricted pathways for heat dissipation (Fig. S10-I). It is important to highlight that the temperature of the top electrode, located at the air-water interface, is observed to be slightly lower than the measured T$_s$ due to the cooling effect of evaporation and the differing thermal environments present at the bottom (heating source) and top (ambient air) of the setup. We quantified this temperature difference as ΔT (see Fig. SI 11). Figure 2b illustrates the time trace of ΔT as T$_s$ is increased from the ambient temperature of 25 °C to 70 °C, peaking at around 7 K.

We further examined the slope of the voltage-temperature curve (Fig. S9) across various conditions, including pH, electrolyte concentration, and the initial equilibrium constant ($K_{aO}$), obtaining minimal dependence on these external factors. Indeed, the dV/dT slope is predominantly influenced by the enthalpy of dissociation of the surface groups, $Δ_H$, which is the chemical characteristic of the material (Fig. SI 2). While this relationship is derived numerically, it underscores a significant physical dependence on the key thermodynamic parameters at play. Thus, we experimentally explored how different surface properties affect the device's open-circuit performance metrics in relation to T$_s$. Figure 2c presents the voltage-temperature profile for two distinct samples that share the same silicon N-type low-doping core but differ in their outer shells, composed of Al$_2$O$_3$ (pink curves) and TiO$_2$ (purple curves), respectively. These experiments were conducted at two concentrations of KCl, specifically 1 mM and 100 mM. Based on the experimental results, Al$_2$O$_3$ displayed a more considerable voltage increase compared to TiO$_2$ for equivalent temperature rises (Fig. 2c), and the slopes remain insensitive to changes in concentration, consistent with the simulation. We observe that the experimental curves present a linear range (approx. up to 20 K), consistent with our COMSOL model, and a super-linear regime at higher temperatures (Fig. SI 12 and Table S1). Within the linear regime, we can extract the dV/dT slope and relate it to the enthalpy of dissociation, confirming the expected trend due to the lower enthalpy of TiO$_2$ compared to Al$_2$O$_3$[36] (Fig. 2d). This behavior clearly stems from the distinct chemical properties of the two materials. Furthermore, the thermal conductivity of the oxide surface—commonly low in TiO$_2$—has minimal impact on the system's overall behavior (see Fig. SI 10-III).

As we show later, to model the experimental $V_{oc}(T)$ data beyond the linear regime, it is necessary to account for the complex interplay of the chemical equilibria and the enhancement of the evaporation rates with temperature. This is discussed in the decoupling strategy section, where we develop a comprehensive equivalent electrical circuit and establish an expression for V$_{OC}$ by integrating three distinct phenomena.

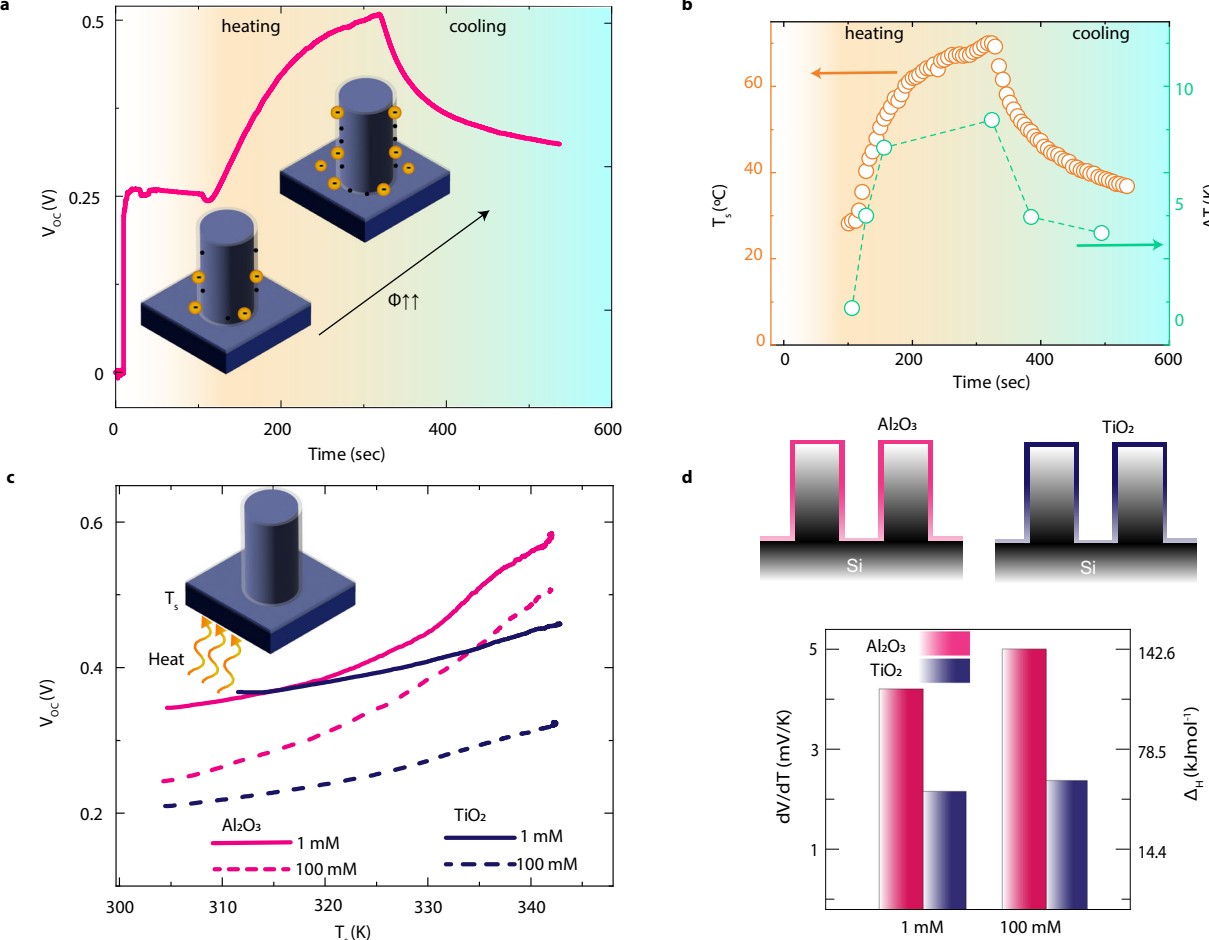

**Fig. 2 | Role of temperature for different coatings and salinity levels. a** The time trace of the measured open circuit voltage at ambient temperature is presented when the silicon surface temperature is increased and then allowed to cool down. The inset illustrates a qualitative rise in the chemical potential difference ($\Delta\Phi$) for a single NP that has been wetted with electrolyte. **b** The time trace of the silicon surface temperature when the heater is turned on and turned off once the maximum is reached. The right axis shows the corresponding temperature difference between the bottom silicon surface and the top electrode (at the end of the liquid meniscus). **c** The voltage-temperature profile of the two devices was measured using the same top electrode at varying salinities. Each device consists of an identical silicon core and a dielectric shell made of $Al_2O_3$ and $TiO_2$: 1 mM (solid lines) and 100 mM (dashed lines). **d** (Top)Schematic representation of the core-shell nanopillars with $Al_2O_3$ (pink) and $TiO_2$ (purple) shells. (bottom) The slope of the $V_{OC}$-temperature curves for the linear regime (up to a 20 K increase in temperature) and the corresponding estimated values of enthalpy $\Delta_H$.

## Effect of irradiation on chemical equilibrium

As described earlier, the surface charge σ can also be varied at a constant temperature due to dynamic changes in the surface concentration of protons ($[H^+]_s$) under irradiation. Thus, we conducted a comprehensive series of measurements to understand the device performance under various illumination conditions and different solid and interfacial properties. We begin by assessing the open circuit voltage of the same samples of the temperature-dependent study, namely N-type low-doping silicon core nanopillars coated with $Al_2O_3$ and $TiO_2$. The $V_{OC}$ time trace is shown in Fig. 3a (purple for $TiO_2$ and pink for $Al_2O_3$), omitting the transient phase where it rises from 0 V to a steady-state value for improved clarity.

Under ambient conditions, before illumination, the devices show a stable voltage (0.36 V $Al_2O_3$, 0.15 V $TiO_2$). Upon exposing the samples to solar illumination (AM 1.5, 1 kW/m²), we observed an instantaneous increase in the measured voltage, which stabilizes at a significantly higher steady-state value (0.55 V $Al_2O_3$, 0.33 V $TiO_2$). Conversely, when we turned off the light source, the voltage immediately decreased, reverting to its initial value recorded in the dark. To validate the consistency of this response, we performed multiple cycles of switching the light on and off, and each cycle demonstrated a reliable and

pronounced rise and fall in voltage. We note here that due to the instantaneous nature of the $V_{OC}$ change upon illumination, photothermal effects are expected to play a minor role. Instead, the photogenerated charges contribute to the observed behavior.

When n-doped silicon is subjected to illumination, upward band bending occurs, accumulating holes at the silicon-oxide interface[31] (Fig. 1a panel iii). Simultaneously, on the liquid side, excess charges at the capacitive interface drive the dissociated cations of water, primarily hydronium ions ($H_3O^+$), away from the interface. It also creates an attractive force for hydroxide ions ($OH^-$), drawing them toward the interface to neutralize the accumulated holes. Consequently, this process leads to a reduction in the concentration of $[H^+]_s$ under illumination, increasing the surface charge (σ) that in turn produces a higher $V_{OC}$ or a positive photovoltage ($Vph$). Moreover, a differential capacitance of the double layer, $C_{DL}$, can be defined as the change in surface charge (σ) due to a change in $\Phi$, and mathematically expressed as by $C_{DL} = (\partial\Phi/\partial\sigma)^{-1}$, which increases with surface charge (see Fig. SI 14 for derivation). To quantify the time-dependent changes in total capacitance, which is a combination of the three capacitances as shown in Fig. 1e, we measured the device's complex impedance in real time at a fixed frequency of 1 kHz (see Fig. SI 15 for details on the

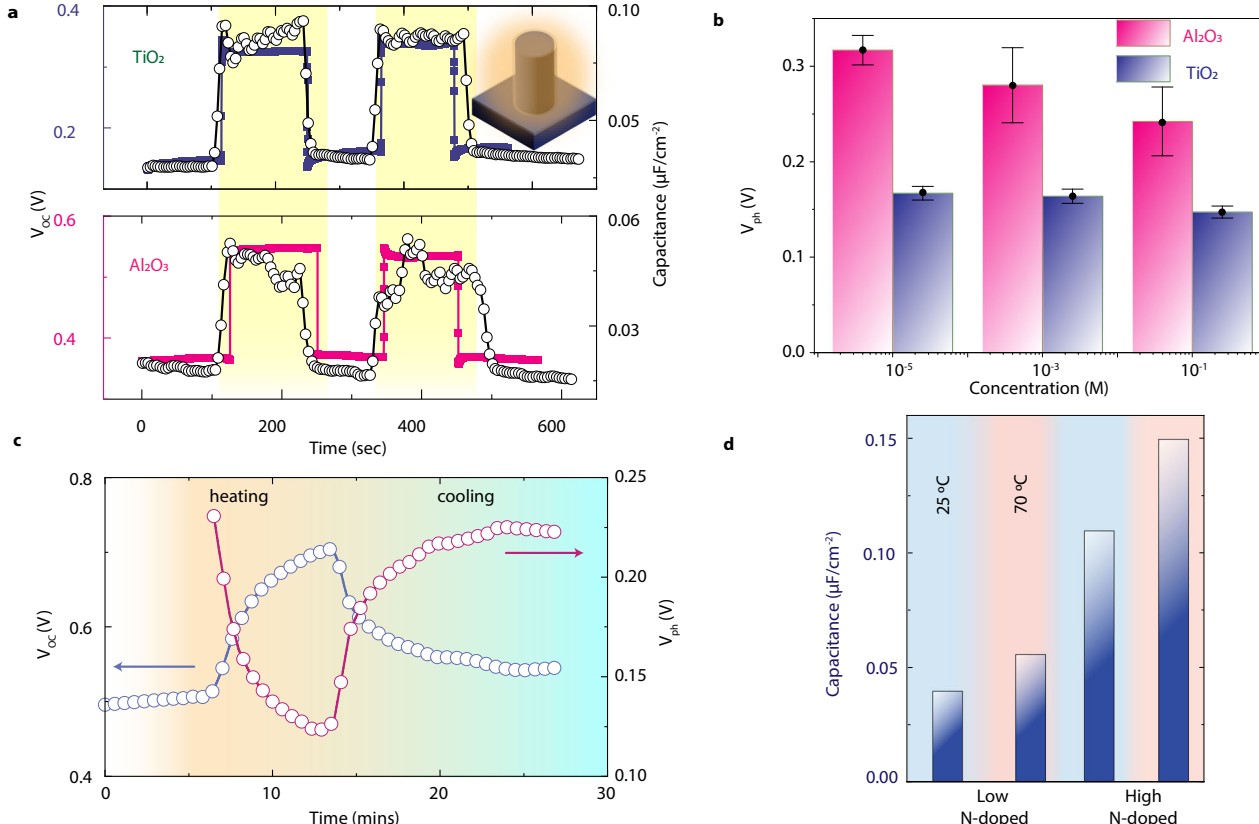

**Fig. 3 | Role of Light for different coatings and salinity levels. a** Time trace of the measured open-circuit voltage and capacitance at 1 mM KCl for two devices with different dielectric shells and low N-doping (1–20 Ω.cm). The test was conducted under ambient conditions and 100 mW/cm² solar illumination (yellow-shaded region). **b** Measured photovoltage under 100 mW/cm² intensity for samples with Al₂O₃ and TiO₂ shells (same low N-doped silicon core) at different concentrations of KCl. Error bar represents the standard deviation of three measurements.

**c** Measured open circuit voltage in the dark and the corresponding photovoltage measured as the device is heated and then allowed to cool down. **d** Steady-state capacitance (blue bars) values for two devices with Al₂O₃ shell, but with different doping of silicon core (low N-doped: 1–20 Ω.cm and high N-doped: <0.005 Ω.cm) at 1 mM KCl. The blue and red shaded region is measured at a surface temperature of $T_S = T_{ambient} = 25\ °C$ and $T_S = 70\ °C$, respectively.

measurement and estimating the capacitance). Figure 3a clearly shows that alterations in capacitance triggered by light irradiation, due to changes in the surface charge dynamics, directly correlate with observed increases in $V_{OC}$.

**Photocharging of charged interfaces**

We then measured the open circuit voltage of the same two low N-doping Si (1–20 Ω.cm) samples for different salinities (from 0.01 mM to 100 mM). Figure 3b presents the photovoltage, defined as $V_{ph} = V_{OC}^{light} - V_{OC}^{dark}$, for both materials across varying salinities. Our observations clearly demonstrate that the photovoltage recorded for samples with Al₂O₃ shells consistently outperformed that of the TiO₂ samples across all salinity levels tested. Our observations also reveal a significant decline in photovoltage as salinity levels increase for both sample types. Intriguingly, we further noted that High N-doped silicon samples exhibit photovoltage values that are consistently over 150 mV lower than their low N-doped counterparts (Figs. S15 and S16). Finally, we assessed the photovoltage as a function of combined heating/cooling and irradiation (Figs. 3c and SI 17). Interestingly, we observed that, contrary to the open circuit voltage, which increases with temperature, the photovoltage decreases with increasing temperature.

To better understand all of these critical findings and their relationship to the solid-liquid surface charge, we conducted capacitance measurements under all these different conditions. As shown in Fig. 1e, the measured capacitance is linked to three capacitances in series. Firstly, the space charge layer capacitance is directly proportional to

silicon doping[37]. We thus compared the response of two samples with different silicon doping, specifically Low N-doping (1–20 Ω.cm) and high N-doping ( < 0.005 Ω.cm). As shown in Fig. 3d (blue ($T_s = 25\ °C$) and red ($T_s = 70\ °C$) shaded areas), the capacitance for the low-doped silicon sample (left two columns) was significantly lower than that of the high-doped silicon samples (right two columns), in agreement with the expected trend. Secondly, the capacitance of the oxide layer is given by $C_{ox} = \varepsilon_0 \varepsilon_r / d$, where $\varepsilon_0$ is the permittivity of free space, $\varepsilon_r$ is the dielectric constant of the material, and d is the thickness of the oxide. The higher dark capacitance of TiO₂ compared to Al₂O₃ (Fig. 3a, right axis) can thus be related to the dielectric constant of anatase TiO₂ being 3-5 times higher than that of Al₂O₃[37], explaining the consistently lower photovoltage for TiO₂ samples across all the salinity values (Fig. 3b). Thirdly, the EDL capacitance must increase with surface charge (see Fig. SI 14 for the analytical expression). By measuring the capacitance at different electrolyte concentrations and temperatures (Fig. SI 18), we confirm that it rises with increased electrolyte concentration and surface charge. This is also in agreement with the observed trend of decreasing photovoltage with an increase in electrolyte concentration due to higher capacitance (Fig. 3b) as well as with the increase in capacitance with temperature for both Low- and High N-doping of Si (Fig. 3d, blue bars, red shaded areas). Overall, these results confirm the key role of light-triggered alterations of the chemical equilibrium at the interface via capacitive photocharging, concurrently underscoring the complex interplay between doping levels, salinity, and capacitance.

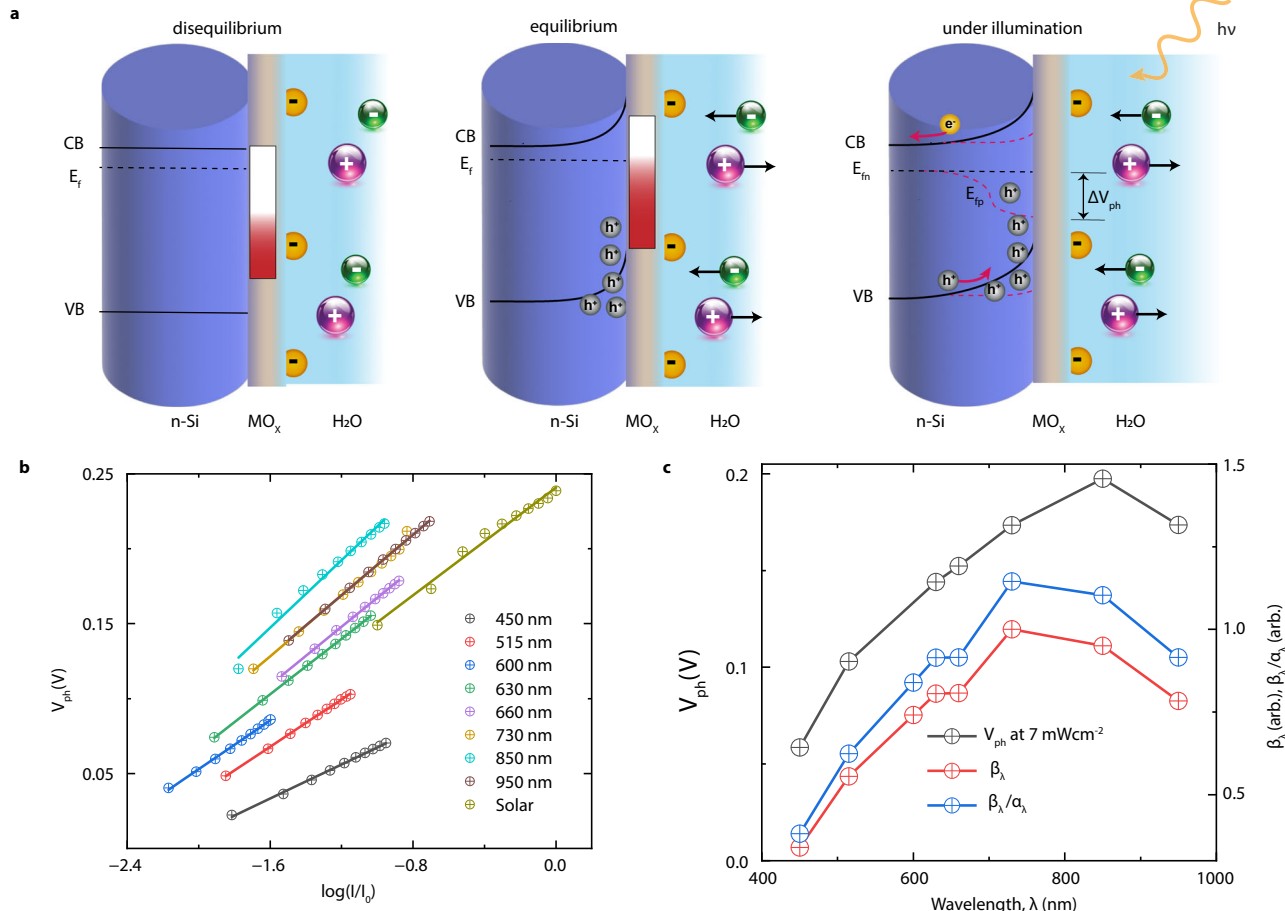

**Fig. 4 | Role of Light for different coatings and salinity levels. a** Band diagram of the semiconductor-metal oxide-electrolyte interface. (left) The liquid-solid interface is formed, but equilibrium has not yet been reached. (middle) Equilibrium has been established, resulting in Fermi-level alignment across the interface. The red-white gradient represents the filling of surface states leading to the Fermi level ($E_f$) pinning. (right) Under illumination, Fermi-level splitting occurs. The difference in the quasi-Fermi levels of electrons ($E_{fn}$) and holes ($E_{fp}$) equals the measured photovoltage. CB conduction band, VB valence band. **b** Measured photovoltage for

different monochromatic incidence as a function of light intensity, plotted in logarithmic scale. $I_0$ is equal to 100 mW/cm². The points corresponding to solar is the measured photovoltage for the full solar spectrum. The experimental data points are shown in circle, while the line is the linear fit. Fitting parameters are given in Table S2. **c** The black curve is the measured photovoltage for different at 7 mW/cm². The red curve is the estimated $\beta_\lambda$ from the linear fit. The blue curve is the value of $\beta_\lambda$ normalized absorptance $\alpha_\lambda$ of the sample (see Fig. S19). The subscripts $\lambda$ signifies wavelength-dependent physical quantities.

To further investigate the interplay between photocharging and interfacial processes, we finally examined how variations in light intensity influence the photovoltage response. As shown in Fig. 4a, due to the capacitive effect, accumulating holes at the silicon-oxide interface drive the dissociated cations of water, primarily hydronium ions ($H_3O^+$), away from the interface. This leads to a reduction in the concentration of $[H^+]_s$ enabling higher surface charge ($\sigma$) that in turn produces a higher $V_{OC}$ or a positive photovoltage ($V_{ph}$). The number of electron-hole pairs generated is directly proportional to the intensity of the light. However, not all generated carriers will lead to the charging effect as there will be a surface or bulk recombination process, which depends on the interface structure and the energy of generated carriers[38]. While the detailed analysis of these phenomena is beyond the scope of this manuscript, we notably observe that the number of generated charge carriers is directly proportional to the light intensity, expressed mathematically as $n_h \sim \beta_\lambda I_\lambda/E_\lambda$, where $\beta_\lambda$ is the effective number of carriers generated per unit incident photon, $I_\lambda$ is the incident intensity, and $E_\lambda$ is the energy of the photon. On the other hand, the concentration of protons on the liquid side is given by the Boltzmann distribution[37] as $[H^+]_s \sim \exp(-\frac{e\Phi}{k_B T})$, which influences the chemical potential according to the relationship[30] $\Phi \sim -\log([H^+]_s)$. Assuming the change in $[H^+]_s$ is directly proportional to the change in $n_h$ we can

establish the dependence of photovoltage on intensity

$$V_{ph} \sim \Delta\Phi \sim \log\left(\frac{\beta_\lambda I_\lambda}{E_\lambda}\right) \tag{3}$$

We conducted measurements of photovoltage across various monochromatic light wavelengths, ranging from 450 to 950 nm, as well as the full solar spectrum, across a broad intensity range. The results reveal that the photovoltage exhibits a clear logarithmic dependence on light intensity, with a linear trend observed when plotted on a logarithmic scale (Fig. 4b and Table S2 for fitting parameters). Additionally, we can derive estimates for the parameter $\beta_\lambda$ at different wavelengths, as illustrated in Fig. 4c, highlighting the selectivity of the wavelengths. Lastly, we have shown in Fig. 4c (blue line) the value of $\beta_\lambda$ normalized to the absorption of the sample $\alpha_\lambda$ (Fig. S19) to exclude the variation in the amount of light reaching the surface for different wavelengths.

Overall, for a deeper understanding of the underlying mechanisms that contribute to these pronounced light-sensitive behaviors, we need to develop a more comprehensive model of the system, capable of capturing not only the solid-liquid interface but also the light response of the silicon core. In the following section, we introduce an

equivalent circuit model and subsequently use it to gain a complete description of the device's behavior.

## Decoupling strategy and rationale for equivalent electrical circuit

Drawing from the mechanistic insights related to the interface, our goal is to enhance the understanding at the device level. We can break down our decoupled device into three distinct regions, each characterized by unique interfacial processes. As illustrated in Fig. 5a, the studied system consists of three key components: the top electrode, the electrolyte layer, and the bottom nanostructured electrode, each defined by its solid/liquid interfaces. At the bottom nanostructured electrode, the coupling of the three surface capacitances (Fig. 1e) leads to a surface charge regulation, whenever either the solid or the liquid side of the interface is perturbed, establishing a non-linear dynamic feedback loop (Fig. SI 20). It is essential to recognize that the interaction of surface charges with ion distribution in the electrolyte layer further results in a coupling between the bottom and top electrodes. As we outline further, the equivalent circuit depicted in Fig. 5b accurately reflects this multi-level coupling, capturing the complex behavior of the hydrovoltaic devices.

*Evaporation-induced Ion Streaming at the Three-phase Boundary of the Top Electrode*: as depicted within the green rectangle, the top electrode, which serves as the positive terminal for electrical measurements, is only partially immersed in the electrolyte, with the remaining portion exposed to air. This arrangement creates a liquid meniscus and a three-phase contact line around the perimeter of the electrode. The evaporation from this region generates fluid flow along the meniscus. To assess the role of the meniscus at the top electrode, we measured the evaporation rate from our working device with a 1 cm² area SiNP bottom electrode (Low N-doping, coated with $Al_2O_3$) under three different conditions: (a) without any top electrode, (b) with a W = 0.8 mm (perimeter) top electrode, and (c) with a W = 3.2 mm top electrode. We observed that the evaporation rate nearly doubled when introducing the small electrode compared to the scenario without an electrode. Additionally, the evaporation rate increased almost fourfold with the larger electrode, which correlates with the perimeter of the three-phase contact line (Fig. S5).

The flow induced by evaporation drives free ions in the liquid's EDL to stream, which can be represented by a current source, $I_{Evap}$, with an appropriate resistance in parallel, named $R_{3\phi}$. Equivalently, it can also be represented by a voltage source with $R_{3\phi}$ in series (Fig. S21). Physically, the voltage source originates from the three-phase contact boundary that generates an evaporation-rate-dependent contact potential difference between the wet and dry sides of the top electrode[39].

We obtained an expression for the current source $I_{Evap}$ in the equivalent circuit ("Methods", Eq. (7)) that shows that it is indeed proportional to the evaporation velocity ($v_f$). This expression shows that $I_{Evap}$ depends on the charge flux ($\bar{\sigma}$) induced by the direct interaction of polar solvent molecules with the evaporating surface[2], as well as ion separation arising due to the chemical potential difference ($C_{tr}\Phi$) across the electrolyte. To confirm the role of polar solvent molecules in the process, we tested the device using water or ethanol under otherwise identical conditions. The measured $V_{OC}$ values for DI water and ethanol at $T_S$ equal to 25 °C (65 °C) are 0.552 V (0.690 V), and 0.195 V (0.336 V), respectively (see Fig. SI 22 for more details on the effect of solvent), confirming a clear dependence on the polar solvent molecules' interaction with the charged surfaces. Overall, these observations indicate the importance of ion streaming at the three-phase boundary created at the top electrode in the system and the need to include it explicitly in the equivalent circuit model.

*Chemical Potential Gradient across the Electrolyte Layer*: as illustrated in the yellow rectangle, the wetted portion of the top electrode and the bottom SiNPs have different surface charges, resulting in a

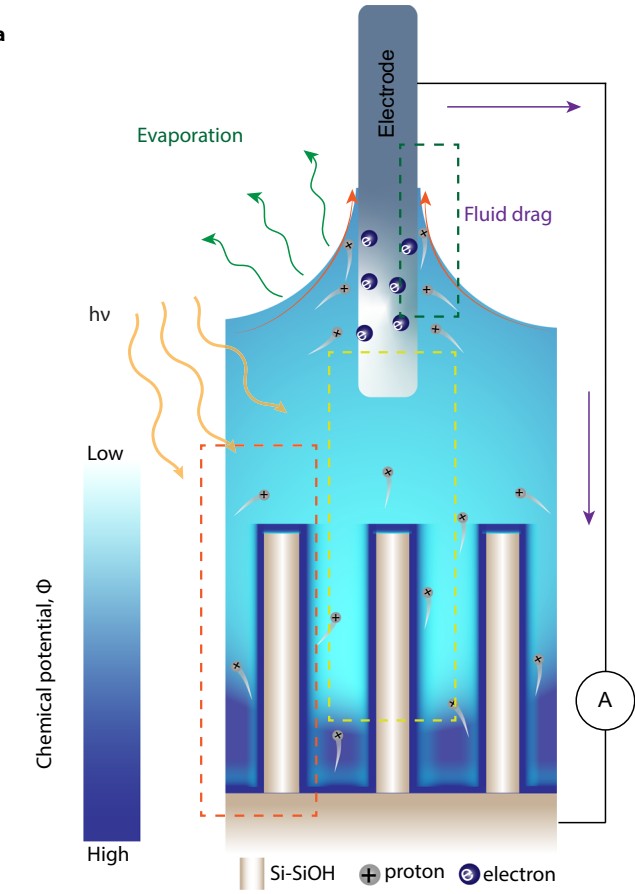

**a**

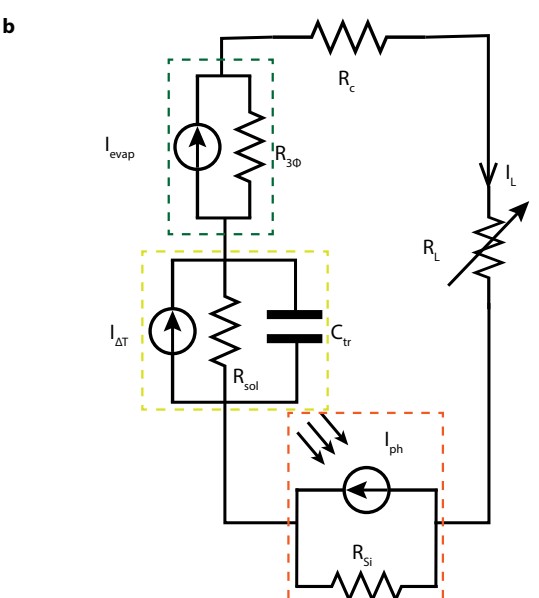

**b**

**Fig. 5 | Equivalent circuit model of the EDHV device. a** Schematic representation of the device working principle. The three regions enclosed in the box highlight the different aspects of the device that govern the electrical output. The green box illustrates the generation of a potential difference across the liquid wetting front caused by evaporation-driven fluid flow along the meniscus region. The yellow box depicts the existence of a chemical potential gradient (due to the differences in chemical characteristics of the surface and temperature) between the wetted parts of the top and bottom electrodes. The orange box shows the light-active region that generates photovoltage under solar illumination. **b** An equivalent circuit representation of the device is outlined above. A contact resistance is included alongside a variable load resistance. All variables are defined in the "Methods".

chemical potential difference ($\Phi$) across the electrolyte layer[6]. Due to this, the electrolyte region exhibits capacitive effects as the surface charges of either electrode are perturbed by thermal effects or light irradiation. We thus define a transfer capacitance $C_{tr}$ of the electrolyte, which is the capacitance arising from ion transfer across it[39]. We have obtained an analytical expression for transfer capacitance that depends on the geometrical parameters of the bottom nanostructures as well as the size of the top electrode and the features of the related meniscus regions (see Fig. SI 23 for detailed derivation).

Under equilibrium conditions, the chemical potential difference would cease to exist. However, due to the temperature gradient $\Delta T$ across the electrolyte layer, the non-zero chemical potential difference persists in the steady-state. Overall, the yellow region can be modeled as a current source driven by the thermodiffusion, $I_{\Delta T}$, in parallel with a resistor, which accounts for the ionic resistance of the electrolyte solution $R_{sol}$, and a capacitor, $C_{tr}$.

*Photovoltage at the Silicon-Oxide-Water Interface*: To account for the influence of light, we need to examine the changes in interfacial band bending at the silicon-oxide-water interface as the sample is illuminated. As illustrated in Fig. 4a (right panel), the generation of electron-hole pairs creates a quasi-Fermi level splitting that results in a photovoltage ($V_{ph}$), as shown in Fig. 3c. As highlighted in Section 3.3, critical factors such as doping, salinity, and the oxide layer non-linearly influence the capacitive photocharging mechanism, which, in turn, has a profound effect on the magnitude of $V_{ph}$. Therefore, the strength and effectiveness of the photocharging capacitive component are fundamentally encapsulated in the equivalent circuit by the value of $V_{ph}$, underscoring its crucial role in this phenomenon[37,40] (Fig. SI 20). This can be represented by a voltage source[32,41] or, equivalently, by a current source, $I_{ph}$, in parallel with a resistance of the photoactive interfacial region ($R_{Si}$).

Based on the described equivalent electrical circuit, we can derive a few critical equations concerning the device response (full details in "Methods"). These can be used to quantitatively analyze the experimental data and therefore clarify the response of this general model of an EDHV system. Remarkably, the experimental non-linear trend of $V_{OC}$ with temperature (Fig. 2b) can be finally elucidated thanks to the obtained expression for $V_{OC}$ (Eq. (4), and "Methods"), which includes the three contributions discussed earlier: (i) evaporation-induced ion streaming, (ii) the chemical potential difference, (iii) photovoltage.

$$V_{OC} = v_f \bar{\sigma} r_{3\Phi} + \Phi\left(1 + v_f r_{3\Phi} C_{tr}\right) + V_{ph} \qquad (4)$$

The resistive element $R_{3\Phi} = \frac{r_{3\Phi}}{W}$, where W is the perimeter of the meniscus region (see "Methods"). All quantities in the expression are defined in "Methods". Notably, both $\Phi$ and $v_f$ increase with temperature (Figs. SI 9 and SI 24), thus their product in the second term leads to the observed quadratic dependence of the open circuit voltage on temperature (Fig. S12 and Table S1).

## Power output analysis using an equivalent electrical circuit

Finally, we investigate the influence of various operational parameters on the power output of our device, emphasizing the effects of temperature, solar illumination, and electrolyte concentration. Figure 6a shows a schematic representation of power curve measurements by sweeping load resistances in the range 100 Ω–70 kΩ. Firstly, the peak value (position) of the power curves increases (shift towards lower resistance) as the environmental conditions change from ambient ($T_s$ = 25 °C, no light irradiation) to 1 Sun ($T_s$ = 25 °C, 1 kW/m² light irradiation) and then heat ($T_s$ = 70 °C, no light irradiation), as shown in Fig. 6b. Notably, the measured power curves indicate that increasing the concentration of KCl from 1 mM to 100 mM nearly doubles the maximum power output despite a lower open circuit voltage at higher concentrations. Furthermore, as observed in the power curves at varying temperatures (Fig. 6c), the maximum power at 70 °C surpasses

the ambient temperature output by over five times, demonstrating the profound impact of thermal conditions on device performance. Moreover, the parabolic dependence of power output and voltage (**SI 25**) across the load resistance underscores the importance of optimizing both aspects to achieve high performance.

The power output is governed by the equivalent electrical circuit, allowing us to derive an expression that captures the relationship between load resistance and output power. The simplification of this expression into a two-parameter family of curves $P = A(T, I, c_0) R_L / \left[B(T, I, c_0) + R_L\right]^2$ demonstrates the distinct roles of parameters A and B, which depend on temperature, electrolyte concentration, and solar intensity. By fitting this expression to the experimental data in Fig. 6b, c, we have compiled the values of the A and B parameters for a range of different conditions, as shown in Fig. S26, which can be used to obtain the current and voltage utilized to obtain the power density. The peak power output, represented by $P = A/4B$, occurs at a specific load resistance $R_L = B$, indicating that careful tuning of these parameters is crucial (see also Fig. SI 27 for load line analysis to estimating power). Notably, achieving double the maximum power output at 100 mM compared to 1 mM KCl concentration—despite a lower open circuit voltage—highlights the significant impact of internal resistance on peak power output, as indicated by parameter B in our study model. In fact, the increase in the open circuit voltage with temperature reinforces the need to consider material choices carefully; a transition from $TiO_2$ to $Al_2O_3$ yielded remarkable enhancements of 90% and 57% in $V_{OC}$ at 25 °C and 70 °C, respectively (Fig. 6d). Furthermore, the influence of silicon doping illustrates another pathway for enhancing power output (Fig. S16). High N-doped $Al_2O_3$ samples displayed a 28% increase in open circuit voltage over low N-doped counterparts, leading to a 1.6-fold improvement in power output. Lastly, long-term stability tests revealed that the device consistently maintained a steady voltage output for over 50 h. Remarkably, this stable performance persisted even under continuous light exposure, as demonstrated in Fig. 6e, underscoring the robustness and reliability of the EDHV architecture. This finding lays the groundwork for future research focused on material engineering to optimize device performance under various environmental conditions. In summary, the interplay between concentration and material properties presents a rich avenue for improving power output in EDHV devices operating under the influence of external heating and solar illumination.

The experimental results, consistent with the proposed model, demonstrate significant enhancements in power output due to the influence of heat and light. This improvement is not solely a result of increased evaporation but is predominantly driven by improved ion dynamics caused by light-induced photocharging at the silicon-oxide interface and thermally enhanced chemical potential at the oxide-electrolyte interface. Thus, it provides a unified perspective that incorporates thermal and photovoltaic effects, in addition to the electrical potential generated at the top evaporating surface.

## Discussion

In summary, this work establishes a spatially and functionally decoupled hydrovoltaic platform that disentangles the key physical processes—evaporation, ion transport, and interfacial chemical equilibrium—that govern energy generation in evaporation-driven water-based systems. By strategically introducing an intermediate electrolyte layer, we enabled independent yet coupled control over each of these phenomena, laying the foundation for a unified and generalizable framework to design hydrovoltaic operation beyond material-specific constraints. By systematically modulating experimental parameters such as doping concentration, oxide composition, and salinity, we effectively demonstrate predictable and tunable performance under illumination through the photocharging of charged interfaces. This approach transforms empirical observations into

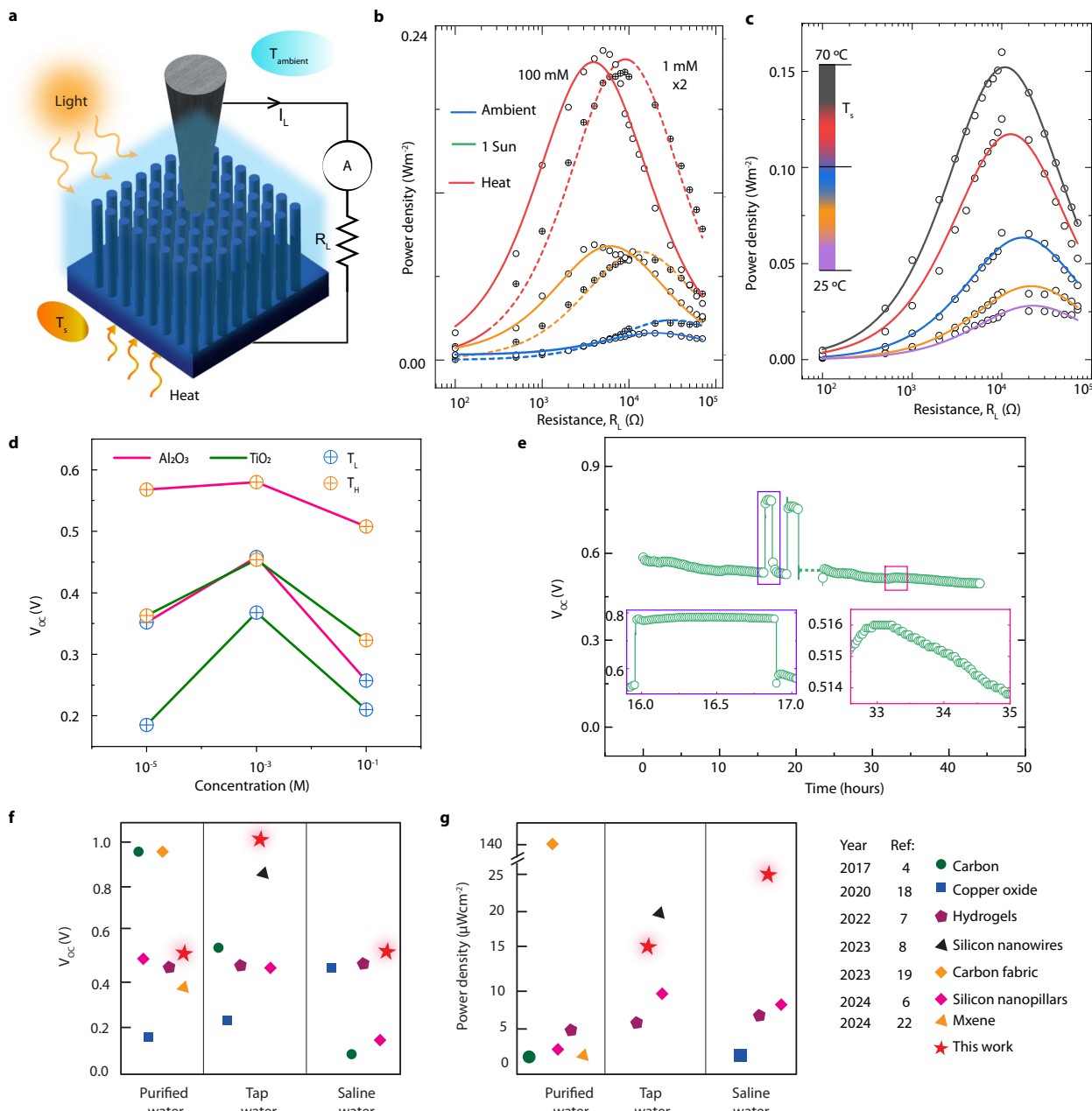

**Fig. 6 | Device performance as a function of materials, electrolyte, and environmental conditions (Heat/Light). a** Schematic representation of the device measurement configuration with various external loads for different surface temperatures and under illumination. The measurements were conducted at ambient temperature with or without solar illumination or elevated surface temperature. **b** Power-load resistance profiles for various conditions (ambient: $T_s = 25\,°C$; 1 Sun: $1\,kW/m^2$ solar intensity and $T_s = 25\,°C$; heat: $T_s = 70\,°C$) at 1 mM and 100 mM KCl. The profiles demonstrate an increase in peak and a shift of the toward lower resistances under illumination and at elevated temperatures. The empty (crossed) circles represent experimental values, while the solid (dashed) lines illustrate fits based on our model for 1 mM (100 mM) KCl; **c** Power-load resistance profiles for various surface temperatures ($T_s$) at 1 mM KCl demonstrate an increase in peak height and a shift in the peak toward lower resistance at elevated temperatures. The empty circles represent experimental data, while the solid lines show the model fit

(fitting parameters in S26). **d** Steady-state open circuit voltage values for two devices with $Al_2O_3$ and $TiO_2$ dielectric shell but with the same silicon core (Low N-doped: $1–20\,\Omega.cm$) at different salinity values. The blue and orange circle is measured at a surface temperature of $T_s = T_{ambient} = 25\,°C$ and $T_s = T_H = 70\,°C$, respectively. **e** Long-term stability testing of the device. The open circuit voltage was measured for 45 h using an Ag/AgCl electrode. We illuminated the sample with light (solar spectrum, $100\,mW/cm^2$) in 1-h on and off cycles for a total duration of around 15 h. The inset provides an enlarged view of the regions marked in purple and pink rectangles. The region marked in purple shows an enlarged light off-on-off cycle, clearly showing the instantaneous response of the $V_{oc}$ to illumination, which demonstrates photocharging rather than photothermal effects. The region marked in red shows stability of the measurement, with less than 2 mV change in 2 h. **f** Open circuit voltage, **g** Power density at different salt concentrations for various EDHV devices.

physically grounded design principles. We have developed a robust semi-quantitative predictive model based on an equivalent electrical circuit, emphasizing transfer capacitance as a central element. This model not only captures the interdependence between physical

structure and interfacial charge dynamics but also provides clear analytical expressions that link these phenomena directly to device output. Our study explicitly investigates the coupled influence of light and heat on solid-liquid interfacial equilibria in hydrovoltaic systems.

We reveal that capacitive photocharging and thermally driven modulation of surface ion equilibria are the dominant mechanisms underpinning energy conversion when interfacial environments are carefully engineered. Without relying on photothermal evaporation enhancement, our approach achieves state-of-the-art hydrovoltaic performance across a wide salinity range (Fig. 6f, g), underscoring the robustness and scalability of the design. Together, these insights provide a rigorous physical foundation for advancing hydrovoltaic technologies and offer a roadmap for designing next-generation devices capable of harvesting ubiquitous solar and low-grade thermal energy in a sustainable and efficient manner.

## Methods

### Electrical & thermal measurements

Figures 1a, 5a, S6, and S7 illustrate the configuration for electrical measurements, where SiNPs serve as the active substrate, and the Al layer functions as the back electrode, while Ag/AgCl (unless stated otherwise) is employed as the top electrode. We used an Ag/AgCl electrode as it is considered fully reversible, which ensures that the electrodes entirely consume the charges accumulated in its EDL at overpotentials, which are practically zero. There is no unwanted potential difference, which induces a conduction current. The open circuit voltage and complex impedance measurements were done using a CHI bipotentiostat. The impedance measurements were done at zero applied DC voltage with an amplitude of 5 mV at a fixed frequency of 1 kHz (Fig. SI 15). The resistance was added to the circuit to measure voltage and power at different load resistances. The bipotentiostat was connected in series or parallel for current or voltage measurements, respectively. For temperature-dependent measurements: by changing the voltage applied to the Peltier element, the device's temperature varied from an ambient temperature of 25 °C to 70 °C. Temperature is measured using an Infrared camera (Fortric 600 s) with 0.2 frames per second. A type-K thermocouple was used for calibration (Fig. S28). A solar simulator (Newport 66 984-300XF-R1 Xe lamp) with an AM 1.5 G filter was used as the light source for the light-dependent study. For a wavelength and intensity-dependent study, we used the LED solar simulator Ossila lamp (G2009A1). The intensity of the monochromatic light was recorded by a detector (Newport, 818-UV/DB) and a hand-held laser power meter (Newport, 843-R-USB). The electrical measurements were performed similarly to the ambient conditions described above. After each measurement, the samples were carefully rinsed with Isopropanol, followed by DI water. Then, the sample was placed in a DI water bath (maintained at 60 °C) for 5–10 min with continuous stirring at 500 RPM. Then, the sample was finally placed in the oven, which was maintained at 80 °C. Furthermore, to minimize the effect of salt precipitation on the output, the sweep of concentration was always performed starting from DI water, followed by low concentration and then higher concentrations. Therefore, for the present study, we expect the effect of salt precipitation at high concentrations to be minimal. We have verified the measurement repeatability at 2 M and 4 M KCl concentrations (Fig. S29).

### Numerical calculation to obtain the voltage-temperature dependence

We developed a 3D numerical model (COMSOL) to solve the Nernst-Planck-Poisson equation to determine the equilibrium distribution of ions and resulting electrostatic potentials. To perform these calculations, however, we must first identify an equivalent simplified geometry using the approach described in our prior work[6]. The simulation details, such as package use and boundary conditions, are given in Fig. SI 30 and Table S3. The following equilibrium reactions govern the amount of surface charge in an oxide-aqueous interface. $MOH + H_2O \Longleftrightarrow MO^- + H_3O^+$. The equilibrium constant for this reaction is given by $K_a = [H_3O^+][MO^-]/[MOH][H_2O]$, which is

temperature-dependent[42]. The temperature dependence is governed by:

$$K_a = K_{a0} \exp\left[-\frac{\Delta_H}{R}\left(\frac{1}{T} - \frac{1}{T_0}\right)\right] \qquad (5)$$

where the material's chemical characteristic, $\Delta_H$, is the enthalpy of dissociation of the surface groups. We used surface charge density as the boundary condition, which is not a fixed value but is instead governed by the above equilibrium reaction. This results in variable surface charge density that depends on the surface potential at the stern plane[35], given by the following equation:

$$\sigma = \frac{-e\Gamma}{1 + \frac{10^{-pH} e^{\frac{e\Phi_0}{k_B T}}}{K_a(T)}} \qquad (6)$$

This enabled us to obtain the open circuit voltage as a function temperature for various conditions, such as by varying electrolyte concentration, pH, and $K_{a0}$. Then, we swept $\Delta_H$, and obtained voltage-temperature lines for various conditions. Finally, we constructed the correlation between the slopes of the voltage-temperature lines and the value of $\Delta_H$ (refer to SI 9).

### Deriving analytical expression for the performance metrics

The total space charge density per unit area of the meniscus region is the sum of the self-charge density $\bar{\sigma} = \rho_f L_{3\phi}$ (when the electrode was inserted in a bulk electrolyte)[2] and the charge density arising due to intermediate electrolyte layer that is equal to transfer capacitance times the chemical potential difference ($C_{tr}\Phi$). Note that $\bar{\sigma}$ is not the bound surface charge, but instead free space charge per unit volume $\rho_f$, multiplied by length of the wetted portion of electrode exposed to air, denoted by $L_{3\phi}$ (refer Fig. S5). The term $I_{Evap}$ is equal to total charge times $v_f$, which is equal to the evaporative mass flux of the fluid (Figs. SI 5 and SI 20) divided by density of water.

$$I_{Evap} = v_f\left(\rho_f L_{3\phi} W + C_{tr}\Phi W\right) = W v_f(\bar{\sigma} + C_{tr}\Phi) \qquad (7)$$

The current driven through the load resistance is obtained by solving the equivalent electrical circuit. The resistive element $R_{3\Phi} = \rho_{3\Phi}\frac{L_{3\phi}}{W} = \frac{r_{3\phi}}{W}$, where $L_{3\phi}$ and W are the length and perimeter of the meniscus region. So, we use $r_{3\phi}$ instead, as it is independent of top electrode size.

$$I_L = \frac{I_{\Delta T}R_{Sol}\left(1 + v_f r_{3\phi}C_{tr}\right) + v_f\bar{\sigma}r_{3\phi} + V_{ph}}{R_{Si} + R_{Sol}(1 + v_f r_{3\phi}C_{tr}) + r_{3\phi}/W + R_C + R_L} \qquad (8)$$

The open-circuit voltage is obtained according to the conditions, $I_L R_L(R_L \to \infty)$, and under no current through the external load, we have the relation satisfying $I_{\Delta T}R_{Sol} = \Phi$. Thus,

$$V_{OC} = \Phi\left(1 + v_f r_{3\phi}C_{tr}\right) + v_f\bar{\sigma}r_{3\phi} + V_{ph} \qquad (9)$$

Using Ohm's law, we obtained the power consumed by the resistive load, and it is given by:

$$P(R_L) = \frac{\left[I_{\Delta T}R_{Sol}(1 + v_f r_{3\phi}C_{tr}) + v_f\bar{\sigma}r_{3\phi} + V_{ph}\right]^2 R_L}{\left[R_{Si} + R_{Sol}\left(1 + v_f r_{3\phi}C_{tr}\right) + r_{3\phi}/W + R_c + R_L\right]^2} = \frac{A(T, I, c_0)R_L}{[B(T, I, c_0) + R_L]^2} \qquad (10)$$

By incorporating the condition of maximum power, $\frac{d(I_L^2 R_L)}{dR_L} = 0$, we obtained the value of load resistance and voltage in terms of the

functions A and B, as follows:

$$R_L = B(T, I, c_0) \quad (11) \quad V = \frac{\sqrt{A(T, I, c_0)}}{2} \quad (12) \quad P_{max} = \frac{A(T, I, c_0)}{4B(T, I, c_0)} \quad (11)$$

### Material and solution preparation

The silicon wafers of different doping (SIEGERT Wafer) were used to fabricate the silicon nanopillar samples as described in Fig. SI 4. The aqueous solutions were prepared using Millipore water ($18.2\,M\Omega\,cm^{-1}$) and KCl (Alfa Aesar, 99% purity). The Ag/AgCl electrode was prepared from silver wire (wire, diam. 1.0 mm, 99.9% trace metals basis) by dipping silver wire into sodium hypochlorite solution for ~1 h.

### Reporting summary

Further information on research design is available in the Nature Portfolio Reporting Summary linked to this article.

### Data availability

Source data are provided as a Source Data file. Available at Zenodo at https://doi.org/10.5281/zenodo.14803626.

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

## Acknowledgements

T.A. and G.T. acknowledge the support of the Swiss National Science Foundation (Starting Grant 211695 and Korean-Swiss Science and Technology Cooperation Fund IZKSZ2_188341). T.A. also acknowledges the support of the Swiss Government Excellence fellowship. The authors acknowledge the support of Milad Sabzehparvar for his assistance in Scanning Electron Microscopy imaging and Vasily Artemov for the insightful discussions during the manuscript preparation. The authors also acknowledge the support of the following experimental facilities at EPFL: the Center for MicroNanoTechnology (CMi) and the Interdisciplinary Centre for Electron Microscopy (CIME).

## Author contributions

G.T. and T.A. conceived the study. T.A. conducted the experiments and developed the models under the supervision of G.T. All authors contributed to writing and revising the manuscript.

## Competing interests

The authors declare no competing interests.
