## [Transparent Peer Review file · Nature Communications]

Enhancing Hydrovoltaic Power Generation through Coupled Heat and Light-Driven Surface Charge Dynamics

Corresponding Author: Professor Giulia Tagliabue

Version 0:

Reviewer comments:

Reviewer #1

(Remarks to the Author)

For the manuscript titled “A Unified Framework for Harnessing Heat and Light with Hydrovoltaic Devices”, the author proposes a promising approach to converting heat and solar energy into electrical power by leveraging both thermodiffusion and photovoltaic effects. Specifically, an integrated hydrovoltaic device was constructed, and an equivalent electrical circuit model was developed to quantify the latter contribution. However, while this work is relatively comprehensive and innovative, it requires significant improvements in comprehensive analysis and in-depth validation, particularly in device design, material characterization, and theoretical model verification before it can be considered further. The following comments outline key areas that need to be addressed:

1. A major concern is the efficiency of the proposed system. The author states, “We demonstrate a state-of-the-art open-circuit voltage of 1V and an output power density of 0.25 W/m² at 0.1M concentration.” However, how does this efficiency compare to other similar photoelectric or photothermal-chemical systems? Comparing only with EDHV devices is insufficient. What about the Faraday efficiency and overall energy conversion efficiency of this system? A more detailed performance evaluation with relevant normalized comparison data should be provided against other related systems and methodologies. Refer to *Materials Today* 42, 178-191(2021).
2. Related to the above, another issue is the effect of auxiliary heating on system performance. In Figure 2, the output power only increases approximately twofold, from 0.25V to 0.5V, when the temperature is raised via external heating. This increase is not particularly significant when accounting for device contact design, error control, and additional energy input. How does this system compare to steam-liquid composite systems, independent photoelectric systems, or standard electrolyte-based electrochemical systems? A more thorough comparative analysis is needed.
3. The study does not sufficiently address light loss due to steam dissipation. How much light actually reaches the electrode surface? Additionally, what is the wavelength selectivity of light in the presence of vapor dissipation, and how do different wavelengths affect the photoelectrode? More discussion and quantitative analysis are needed in this area.
4. The characterization of electrode materials is insufficient. Only three concentration gradients were analyzed in the doping characterization, making the results less conclusive due to large experimental uncertainties. Furthermore, in Figure 3C, the measured photovoltage of TiO₂ is consistently lower than that of Al₂O₃, and its performance decreases with increasing concentration. This suggests that the concentration used was not optimized for maximum activity. The author should redesign the experiments to determine the optimal concentration and improve the accuracy of the results.
5. The study lacks an in-depth analysis of system stability. The author mentions a “partially wetted region”, but what are the wettability characteristics of the electrode? Will ions adsorb onto the electrode surface, potentially hindering its performance? Additionally, how do crystal structure and morphology change before and after the reaction? What is the rate of steam generation, and how does it affect electrode stability? Long-term stability tests and corresponding results are highly recommended.

Reviewer #2

(Remarks to the Author)

The paper focuses on the efficacy of evaporation-driven hydrovoltaic (EDHV) devices, considering a variety of driving forces, i.e., involving thermal and ion concentration gradients, as well as illumination. Given an electrolyte as the medium of the devices, it would be expected that potential differences would be generated.

At the very outset, the results and the analysis are/have been considered through individual phenomenology while there would be considerable coupling between the forces and inter-dependency. Consequently, the obtained voltages and power density would be sensitively dependent on the particular system, environment and testing conditions. While the noted phenomena have been analyzed in terms of performance metrics, it is unclear as to how they would result in the observed enhancements of the voltage, power, etc. For instance, could the observed voltage and power density be predicted from the indicated relations?

A few more specific comments are indicated next.

The decoupling of the top evaporating surface from the nanostructures, and related electrodes placed at the bottom, is not straightforward to understand, due to the intervening electrolyte and related electrochemical effects. Is there a Debye length related correlation, with respect to the various used concentrations? Would not there be a voltage across the EDL rather than "along the length of the NP"?

The time constants related to the temperature increase and decrease are not easy to understand. Was the Peltier heater placed in the liquid? What accounts for the stabilization of the temperature? How exactly does the convective heat transfer play a role?, say considering the related heat transfer coefficient? What is the role of the respective thermal conductivity values of the dielectrics used? Typically, TiO₂ has a lower thermal conductivity. The interplay of the temperature and the ion concentration is unclear. For instance, can the authors explain clearly what is meant by evaporation-induced ion streaming. To what extent would there be a modification of the liquid diffusion constants of the involved ions? The notion of "ion dynamics" needs to be further quantified! How would temperature increase the surface charge linearly? The dV/dT seems to be less sensitive when TiO₂ is used. Why would this be?

Have the authors performed an electrochemical impedance spectroscopy (EIS) analysis? Why was 1 kHz chosen? How was the capacitance monitored and estimated? Light induced charge increase could occur through a variety of factors and the possibilities should be enumerated better. Can the authors consider the possibility of electrolysis? The role of salinity, which is not clearly defined, should be further elucidated.

When Silicon doping is considered, it would have been worthwhile to use n- and p-doping of the same impurity concentration/resistivity. The magnitudes of the voltage change, say 35 mV, should be explained in terms of the band structure. Else, the values just seem incidental to the performed experiment. When it is stated that "surface charges of either electrode are perturbed by thermal effects or light irradiation", the meaning is unclear, with respect to whether the charges come off the surface, neutralized, contribute to the EDL, etc.? Are the authors claiming that the P-type Si is inverted, i.e., "through electron accumulation at the interface" – which is highly unlikely in the presented experiments! The instances of band bending and their causes should be reviewed again by the authors! Similarly, the "logarithmic" variation of the photovoltage with solar intensity is ambiguous!

The definition of transfer capacitance must be clearly quantified. What is the related dielectric permittivity and charge storage distance? A statement such as "thermal motion induces electron transfer between the solid's atoms and the water molecules, facilitated by overlapping of their electron cloud" is imprecise. For the posited "ionization reactions", what is the standard redox potential/s and how does it compare with the overpotentials or observed voltage? The role of solvents must be considered through the respective polar and/or non-polar attributes. What is the assumed value of the K_a in Eqn. (3)?

The authors have come up with an interesting electrical circuit model, in Figure 4(B). However, the coupling of the individual phenomena should be clearly indicated as well. The relating of an evaporation velocity to the surface charge is unclear, given that the surface charge would be bound to the surface. What is the role of electroosmosis? There should be a load-line analysis for the use of the load resistances for the power estimations. evaporation-rate-dependent contact potential difference between the wet and dry sides of

Many typos should be corrected through the manuscript, e.g., not sure how tunneling is involved, as indicated in the caption of Fig. 1(c), as well as several unclear phrases and statements, e.g., "The surface nature is crucial in this relationship", "appears logarithmic-like in nature", etc.

In summary, the paper indicates fascinating interplay of light and heat in an electrolyte, with respect to the generation of electrical voltages and power. However, the interdependence of the various forces is not well explained. Further, the models should indicate or come close to predicting the observed voltage and power values. The authors should then aim, in the next revision, to address these shortcomings.

Reviewer #3

(Remarks to the Author)

I have carefully reviewed this manuscript. The authors have conducted innovative research on hydrovoltaic systems. By introducing an intermediate liquid layer, the authors divided the hydrovoltaic system into a surface evaporation layer, an intermediate ion-conducting layer, and a bottom silicon nanostructure layer, enabling a relatively independent investigation of each component's function. Through a combination of experimental and numerical simulations, the manuscript proposes an equivalent circuit model for this system and assigns corresponding circuit components and parameters to each functional part. By altering the system's temperature, illumination conditions, and silicon nanostructure surface properties, the study reveals mechanisms of ion dynamics, such as light-induced photocharging and thermally-enhanced chemical potential. Through optimization of these conditions, the system achieves an open-circuit voltage of 1V and an output power of 0.25W/m². Overall, the manuscript presents a rich and innovative study. However, due to concerns regarding clarity and

other aspects of the manuscript, I recommend major revisions before publication. Below are my specific comments:

1. Lack of clarity in logic. The study's innovation lies in introducing an intermediate electrolyte layer to separate the hydrovoltaic device into three distinct regions for investigation and in establishing a systematic equivalent circuit for ion transport mechanism analysis. However, the manuscript presents the research in a manner that first describes a large volume of experimental results, then builds the equivalent circuit, conducts numerical calculations, and finally performs output power testing and optimization. This structure fails to highlight the study's novelty and advantages, making it challenging for readers to follow. I recommend reorganizing the presentation to emphasize the key contributions more effectively.
2. Lack of conciseness in content. In line with the previous point, the manuscript includes a substantial number of experiments to support the proposed model. However, this results in an overly lengthy and complex presentation that compromises clarity. Thus, I suggest moving non-essential content to supplementary materials to enhance the manuscript's conciseness. For example, the fabrication of hydrovoltaic devices using silicon nanostructures has been previously reported, so details on device preparation and material characterization could be transferred to the supplementary information (SI). Similarly, in the discussion of temperature and illumination effects on open-circuit voltage, the introduction of electrolyte concentration as a variable appears unnecessary. Please clarify its necessity for explaining the mechanisms, or consider placing it in the SI.
3. In Section 2.1, the device contains only 250 μL of electrolyte. As the temperature rises, how can it be proven that the performance variations result from temperature changes rather than the rapid evaporation of the solution, which could lead to increased ion concentration?
4. In Section 3.1, Figure S1 suggests that the introduction of electrodes significantly affects the evaporation rate. However, considering that the diameters of the introduced electrodes (0.25 mm, 1 mm) are relatively small compared to the original evaporation area (~ 1.3 mm in diameter), how do such small electrodes achieve a twofold or fourfold increase in the evaporation rate? Please provide a more detailed explanation.
5. Ambiguity in the presentation of certain experimental data Figures. Figures 3c and 5e simultaneously use both line and bar graphs for the same dataset, making the intended message unclear. In Figure 5b, two separate y-axes represent output power, which is confusing and may mislead readers.
6. Missing or incorrect legends in some experimental data Figures. In Figure 2c, the plot lines and legend color blocks do not correspond correctly. Figure 2e lacks legends. Figures 3a and 3b also lack sufficient labeling. In Figures 5c and 5d, the temperature indicators are unclear. Please refine these elements for improved clarity.
7. Lack of explanation for certain schematic elements. In Figure 1, the meaning of the circular dots is not explained. Additionally, the colors of the dots in Figures 1d and 1a are different without explanation. In Figure 4d, the significance of the red-white gradient color bar is not clarified.
8. Missing scale bars and unclear indicators in some Figures. Figure 1b lacks a scale bar. In Figure 1c, the meaning of the arrows and horizontal lines is unclear.
9. The reference in Figure 5h is incorrect. Reference 14 describes a device that does not use a silicon nanowire system but rather a silica nanosphere system. Please double-check and ensure the citations are accurate.

Reviewer #4

(Remarks to the Author)

Version 1:

Reviewer comments:

Reviewer #1

(Remarks to the Author)

The authors have taken good efforts in revising the manuscript. I have no further input.

Reviewer #2

(Remarks to the Author)

I thank the authors for responding to my earlier comments in detail. I still think that the analysis of the coupled system investigated is quite complex and am unsure of the extent to which "the unravelling of the complex interplay of different

interfacial phenomena in 410 determining the hydrovoltaic device output” was accomplished in the present work. As indicated in the paper, there are thermal phenomena, interfacial chemistry, electrochemical aspects, and in addition light-induced effects the sum total influence of which is truly hard to understand the coupled interactions and the related simple linear models.

Further, the aspect that “heat and light input have major impacts on the solid/liquid interface due to the temperature sensitivity of the surface group dissociation and the ions desorption/adsorption” is expected qualitatively and the modeling is essentially based on linear approximation/s. For instance, I do not consider statements, such as “An increase in ambient temperature, for example, will lead to a higher temperature of silicon even when no external heating is applied” insightful. The proposed “decoupling” of the top evaporative interface with the bottom electrode- through a change of the “manuscript’s structure” does not do much to enhance the understanding. What accounts for the

New data, such as Fig. R 12, is not particularly meaningful – as only diffusive behavior seems to have been considered. What about the R_{ct} and the C_{dl} components? The related analysis is superficial. In another instance, what is the rationale for indicating that the dV/dT is proportional to the enthalpy of dissociation for the surface groups – seems to be an example of correlation and not causation? What is the basis for fitting the voltage as a function of temperature to the second order? Can the authors indicate more clearly the basis for the 0.25 W/cm^2 in terms of the utilized voltage, current, and area involved?

In summary, I do appreciate the immense effort involved in the experimentation. However, the obtained results still seem to be phenomenological, given the very nature of the experiment with several coupled phenomena. I am unsure as to how well the conducted experiment may be reproduced by other scientific groups.

Reviewer #3

(Remarks to the Author)

I have carefully read the authors’ responses to the reviewer comments, as well as the revised version of the manuscript. I am pleased to see that the clarity of the manuscript has been significantly improved, the logic has become more coherent, and the authors have provided detailed and convincing responses to the previous concerns. Based on this, I believe the article can be accepted for publication after some minor revisions. My comments are as follows:

Regarding the logical coherence of the article:

1. Although the logical clarity of the manuscript has been greatly improved, there are still some weak transitions between certain sections. In Section 3.3, the photocharging phenomenon at the solid–liquid interface is described in detail, along with the corresponding capacitive mechanism. However, in Section 3.4, when discussing the equivalent circuit of the silicon–oxide–water interface, the capacitor component is not reflected. How should this be explained?
2. Following the previous question, in the abstract, the main strategies proposed for enhancing EDHV are “increasing silicon doping” and “switching the dielectric shell from TiO_2 to Al_2O_3 .” In the equivalent circuit, which parameters correspond to these strategies, and how should they be interpreted?

I also have some specific questions regarding certain content in the manuscript:

3. Lines 233–236:

“Within the linear regime, we can extract the dV/dT slope and relate it to the enthalpy of dissociation, confirming the expected trend due to the lower enthalpy and surface site density (Γ in Eq. 1) of TiO_2 compared to Al_2O_3 (Figure 2D–E).” I understand the relationship between the dV/dT slope and the enthalpy of dissociation, but how is this related to the surface site density?

4. Lines 281–283:

“A differential capacitance of the double layer, CDL, can be defined as the change in surface charge (σ) due to a change in Φ , and mathematically expressed as $C_d = (\partial\Phi/\partial\sigma)^{-1}$, which increases with surface charge (see SI 14 for derivation).” What is the relationship between CDL and C_d ? Are they the same quantity? What does “c” represent in SI 14? Why is the Stern layer capacitance (C_s) not reflected in the equation?

5. Regarding the capacitance measurement: the authors converted the measured impedance at a fixed frequency into capacitance using the pure capacitor formula. What is the basis for assuming that the system exhibits pure capacitive behavior at 1 kHz?

6. Lines 459–461:

“Notably, the measured power curves indicate that increasing the concentration of KCl from 1 mM to 100 mM nearly doubles the maximum power output despite a lower open circuit voltage at higher concentrations.”

According to earlier data (e.g., Figure 2c), the open circuit voltage at 100 mM is higher than at 1 mM across various temperatures. Why is it described here as “a lower open circuit voltage at higher concentrations”?

7. In addition, possibly due to the extensive revision of the manuscript, there are several instances of inappropriate figure references and formatting issues that could be optimized. The following points were noted:

- The layout of Figure 3 (with panel b below panel a) is inconsistent with other figures, where panel b is placed to the right of panel a.

- In Figure 3b, only three concentrations are shown, which appears insufficient. Would adding error bars help improve the reliability of the data?

- In Figure 3c, temperature variation is not clearly indicated. Could background shading be used to distinguish heating and cooling phases?

- For Figure 3d, since there is only a single y-axis, placing it on the left would be more conventional.

- Lines 315–317:

“As shown in Figure 3D (blue bars, blue shaded areas), the capacitance for the low-doped silicon sample (left two columns)

was significantly lower than that of the high-doped silicon samples (right two columns), in agreement with the expected trend.”

While the meaning is understandable, the phrasing contains redundancy and inconsistency between “blue bars, blue shaded areas” and “left two columns/right two columns.”

- Subscripts are missing in Lines 320 and 458.
- In Figure 4c, the symbol $\alpha\lambda$ appears but is not clearly explained in the text. Please clarify its meaning and purpose.
- Line 437: Figure 4d is not present in the manuscript.
- Line 482: The reference to Figure 3c does not match the content described in the text.
- Figures 6d and 6f appear to be unmentioned in the manuscript.

I understand that such minor oversights are possible in the process of extensive revisions, but I urge the authors to pay closer attention to these details, especially those that may affect the reader’s understanding of the article.

Reviewer #4

(Remarks to the Author)

Version 2:

Reviewer comments:

Reviewer #3

(Remarks to the Author)

The author has addressed all the concerns I raised in the previous round of review with sufficient and reasonable explanations, which have clarified my doubts. After reviewing, I believe that the manuscript meets the journal's publication standards. Therefore, I recommend the acceptance of the manuscript and look forward to its contribution to the relevant field.

Reviewer #4

(Remarks to the Author)

Reviewer #5

(Remarks to the Author)

The authors have made tremendous efforts to reply to referee 2 and referee 3's comments. In particular, they have convincingly answered to the main concerns raised by referee 2 and edited Section 3.4 to summarize the different coupled phenomena. Although the physical coupling is complex, their description sounds reasonable and beyond a mere phenomenological approach. I therefore recommend publication of the manuscript after the authors have taken into account the following minor corrections:

- Eq.2 : please define all terms. A reference may be given to understand where does the factor 8000 come from.
- Does the evaporation velocity depend on the electrolyte concentration or is it taken to be that of the pure fluid ? Please clarify this point in the main text.
- Eq. 4: please replace $r_3\phi$ by $R_3\phi$ to keep consistent notations

**Response to reviewers**

We thank the reviewers for their valuable comments and constructive suggestions. We
have carefully revised the manuscript in accordance with the feedback provided. Below,
we provide point-by-point responses to each comment. Reviewer comments are in
**black**; our responses are in **blue**, and changes made to the manuscript are indicated in
**red** in the revised text.

**Reviewer #1**

For the manuscript titled “A Unified Framework for Harnessing Heat and Light with
Hydrovoltaic Devices”, the author proposes a promising approach to converting heat and
solar energy into electrical power by leveraging both thermodiffusion and photovoltaic
effects. Specifically, an integrated hydrovoltaic device was constructed, and an equivalent
electrical circuit model was developed to quantify the latter contribution. However, while
this work is relatively comprehensive and innovative, it requires significant
improvements in comprehensive analysis and in-depth validation, particularly in device
design, material characterization, and theoretical model verification before it can be
considered further. The following comments outline key areas that need to be addressed:

Thank you for the reviewer’s thoughtful and detailed evaluation of our manuscript titled
“A Unified Framework for Harnessing Heat and Light with Hydrovoltaic Devices.” We
sincerely appreciate the reviewer’s recognition of the novelty and promise of our approach
in integrating thermal and photovoltaic effects for sustainable energy conversion. We are
especially grateful for the time and effort dedicated to providing in-depth feedback on the
device design, material characterization, and theoretical model validation.

We fully acknowledge the need for comprehensive analysis and rigorous validation to
further strengthen our work. In response, we have carefully revised the manuscript and
taken concrete steps to address each of the concerns raised. To improve the clarity and
logical flow of the manuscript, we have significantly reorganized the structure to highlight
the central contributions better and guide the reader through our rationale and findings
more effectively. The revised manuscript now follows a clear and logical progression:

*Introduction of Key Interfacial Processes* – We begin by identifying the central interfacial
phenomena that govern the device’s behavior, which is the central theme of our work.

This section lays the groundwork for understanding how each process can be modulated
to affect the voltage generation via thermal or light inputs.

*Experimental Platform of the Decoupled System* – Here we introduce the device
architecture and experimental platform that enable the spatial and functional separation
of evaporation, ion transport, and charge generation processes, a key step to achieve a
comprehensive understanding of the operation of hydrovoltaic devices.

*Effects of Temperature Change on Chemical Equilibrium* – We analyze how temperature
alters interfacial surface charge dynamics, leading to observable changes in voltage. This
goes beyond simple thermal enhancement of the evaporation rate previously exploited to
improve device performance.

*Effect of Irradiation on Chemical Equilibrium* – This section examines how light
influences chemical equilibrium and modulates voltage output. As we discuss later,
contrary to prior works, we do not exploit light to induce photothermal effects and
enhance the evaporation rate. Instead, we focus on how light alters the interfacial
chemical equilibrium.

*Photocharging of Charged Interfaces* – We present detailed experiments that show how
capacitive photocharging varies with salinity and illumination, providing critical
mechanistic insight into the photovoltaic contribution.

*Decoupling Strategy and Rationale for Equivalent Electrical Circuit* – Based on the
physical and experimental insights, we introduce an equivalent circuit model, explaining
how it maps to each functional component of the device as well as the interplay of these
complex interfacial phenomena.

*Power Output Analysis Using Equivalent Electrical Circuit* – We then use the model to
analyze voltage and power output, showing how the interplay between physical
parameters can be used to predict and optimize device performance.

We believe this restructuring improves the overall readability of the manuscript and
ensures that the novelty and logic of our approach are communicated clearly and
effectively. Our detailed point-by-point responses are provided below.

1. A major concern is the efficiency of the proposed system. The author states, “We
demonstrate a state-of-the-art open-circuit voltage of 1V and an output power density of
0.25 W/m² at 0.1M concentration.” However, how does this efficiency compare to other
similar photoelectric or photothermal-chemical systems? Comparing only with EDHV
devices is insufficient. What about the Faraday efficiency and overall energy conversion
efficiency of this system? A more detailed performance evaluation with relevant
normalized comparison data should be provided against other related systems and
methodologies. Refer to Materials Today 42, 178-191(2021).

We thank the reviewer for this insightful comment. While we fully agree that a broader
contextualization of performance is important for evaluating a device potential, we would
like to clarify two key points that make direct comparisons with non-hydrovoltaic systems
non-trivial.

Firstly, our electrode is passivated with oxides and operates in a dilute KCl (neutral pH)
electrolyte, making the system fundamentally capacitive—no faradaic reactions occur, as
we've firmly established through our photoelectrochemical tests discussed in the
manuscript. Even in non-passivated conditions and basic electrolyte scenarios, the
possibility of faradaic transformations, such as oxygen or hydrogen evolution, is highly
unlikely under these specific conditions. The overpotential required for these reactions
far surpasses the voltage generated in various scenarios. For example, oxygen evolution
in silicon demands an applied potential of at least 1.2-1.4 V, even under optimal electrode
and electrolyte conditions (Chem. Soc. Rev., 2019, 48, 2158-2181). Thus, our system
distinctly contrasts with typical photoelectrochemical systems, where metrics like
Faraday efficiency and photocatalytic yield are relevant due to the occurrence of redox
reactions. In our case, enhancement under illumination occurs through charge
accumulation and interfacial capacitive effects, not via redox reactions (**Figure R1**),
rendering Faraday efficiency an irrelevant metric.

Secondly, while our device shows enhanced performance under illumination, it does not
rely on photothermal heating. Instead, we exploit photocharging effects, where light
enhances charge accumulation at the interface by modifying the surface proton

distribution and interfacial potential. This is distinct from systems that rely on enhanced
evaporation via broadband photothermal absorption by resorting to a black light
absorber.

Overall, this work focuses on hydrovoltaic devices as promising platforms for passive
energy recovery across scales, focusing on the fundamental processes that determine their
operation. By using heat and light to selectively modulate interfacial interactions—rather
than driving photothermal evaporation or faradaic reactions—we demonstrate that
interfacial effects in hydrovoltaic systems can be harnessed with better performance than
previously recognized. For this reason, a direct comparison with non-hydrovoltaic
systems would be misleading in this case, as photothermal or additional photochemical
effects could be coupled with the existing system to further enhance its performance.

These distinctions are now emphasized more clearly in the manuscript to avoid
misinterpretation and to highlight the novelty of our approach.

Yet, to further address the reviewer's request, we have added an estimated energy
conversion efficiency relative to Carnot's efficiency of approximately **3.5%** (refer to **SI**
**10**), which, although modest, is comparable to other capacitive hydrovoltaic platforms
under similar operating conditions.

Published Data Figure Redacted

**Figure R1:** A) Photoelectrochemical behavior of p+-Si|NiOx, n-Si|NiOx and np+-Si|NiOx electrodes in 1.0 M
potassium hydroxide under simulated AM 1.5G illumination (100 mW cm⁻²) (Reproduced: Chem. Soc. Rev.,
2019,48, 2158-2181). B) Current-Voltage curves under AM 1.5 G illumination of our SiNP electrode in 1 mM KCl.

Changes to the manuscript

**Section 1**

“...In particular, evaporation-induced hydrovoltaic devices (EDHV) stand out due to
their ability to produce a continuous (24hr) electricity output, operating without
external mechanical energy input across a wide range of environmental conditions. This
is achieved through the synergy of spontaneous capillary action and evaporation,
enabling autonomous, low-intensity energy supply solutions ideal for portable
devices...”

“...In particular, beyond enhancing the evaporation rate through photothermal effects,
it is crucial to comprehend how external heat, and light sources impact the interfacial
processes that are essential for effective hydrovoltaic energy conversion...”

**Section 2.1**

“...Consequently, under illumination, photogenerated charges (electrons or holes)
accumulate at the interface^{32,33}. Due to the capacitive effect of the oxide layer, a
concurrent change in the surface proton concentration will occur, leading to a change in
surface charge. This also highlights the importance of the oxide layer in passivating the
silicon surface³⁴ and preventing any chemical reaction between silicon and water. Our
photoelectrochemical test on the device indeed shows no evidence of faradaic activity,
thereby confirming a capacitive charging, rather than faradaic, process (SI 3)...”

2. Related to the above, another issue is the effect of auxiliary heating on system
performance. In Figure 2, the output power only increases approximately twofold, from
0.25V to 0.5V, when the temperature is raised via external heating. This increase is not
particularly significant when accounting for device contact design, error control, and
additional energy input. How does this system compare to steam-liquid composite
systems, independent photoelectric systems, or standard electrolyte-based
electrochemical systems? A more thorough comparative analysis is needed.

Thank you for this insightful comment regarding the role of auxiliary heating and the
need for comparative performance analysis.

We agree that the observed increase in open-circuit voltage—from ~0.25 V to ~0.5 V
with external heating—may appear modest at first glance. However, we would like to
emphasize that the system's power output increases much more significantly, by more
than fivefold under moderate heating conditions (as shown in Figure 6B of the
manuscript). This is because the power output depends on the square of voltage and
internal resistance of the system. With increase in temperature, there is a synergistic
increase in both factors that leads to a very significant increase in power output.

**Section 3.5**

“The peak power output, represented by $P = A/4B$, occurs at a specific load
resistance $R_L = B$, indicating that careful tuning of these parameters is crucial (See also
**SI 26** for load line analysis to estimating power). Notably, achieving double the
maximum power output at 100 mM compared to 1 mM KCl concentration—despite a
lower open circuit voltage—highlights the significant impact of internal resistance on
peak power output, as indicated by parameter B in our study model.”

In comparison to steam-liquid composite systems, which often require high thermal
input, complex system designs, and insulation to achieve boiling and phase transition,
our approach is distinct and advantageous. It operates entirely through surface-driven
thermodiffusion and interfacial effects, using low-grade heat (typically <70 °C)—making
it suitable for integration with ambient heat sources and solar thermal input without the
need for additional infrastructure.

**Section 1**

“...In particular, evaporation-induced hydrovoltaic devices (EDHV) stand out due to
their ability to produce a continuous (24hr) electricity output, operating without
external mechanical energy input across a wide range of environmental conditions...”

Unlike standard electrolyte-based electrochemical systems, which rely on faradaic redox
reactions, our system operates via a non-faradaic mechanism based on capacitive charge
separation and redistribution at solid-liquid interfaces. As we discussed in response to

comment #1, our device differs fundamentally in that it produces voltage through
capacitive ion redistribution. Previous studies have largely enhanced performance
through photothermal evaporation or photoinduced faradaic reactions. In contrast, we
demonstrate that capacitive photocharging and thermally modulated surface equilibria
are dominant mechanisms when interfacial effects are properly engineered.

3. The study does not sufficiently address light loss due to steam dissipation. How much
light reaches the electrode surface? Additionally, what is the wavelength selectivity of
light in the presence of vapor dissipation, and how do different wavelengths affect the
photoelectrode? More discussion and quantitative analysis are needed in this area.

We thank the reviewer for this important comment. As we mentioned in response to
comment #1, we first of all want to emphasize that under light irradiation our
performance change is driven by photocharging effects at the solid/liquid interface and
not by photothermal effects (i.e. evaporation enhancement) or photochemical reactions.

Concerning the light intensity reaching the electrode surface, we have included the
absorption measurement data for the silicon nanopillar sample in **SI 19**. Additionally,
we have provided the absorption measurement of the original silicon substrate as a
reference (**Figure R2**).

**Figure R2:** Absorption measurement for the silicon wafer and the two silicon nanopillar array sample.

Regarding the light impact onto the system, we have now added a dedicated section in
the manuscript, where we explain in detail the intensity variation, wavelength selectivity
on photovoltage, and the amount of light reaching the electrode surface. Our results

confirm that light has predominantly a photocharging effect on the system, an
 observation never reported before for hydrovoltaic devices.

 **Figure 4: Role of Light intensity and wavelength selectivity.** B) Measured photovoltage for different
 monochromatic incidences as a function of light intensity, plotted in logarithmic scale. I_0 is equal to 100 mW/cm^2 .
 The points corresponding to solar is the measured photovoltage for the full solar spectrum. The experimental data
 points are shown in circle, while the line is the linear fit. C) The black curve is the measured photovoltage for
 different at 7 mW/cm^2 . The red curve is the estimated β_λ from the linear fit. The blue curve is the value of β_λ
 normalized to absorptance α_λ of the sample.

Section 3.3

“To further investigate the interplay between photocharging and interfacial processes, we
 finally examined how variations in light intensity influence the photovoltage response. As
 shown in **Figure 4D**, due to capacitive effect, accumulating holes at the silicon-oxide
 interface drive the dissociated cations of water, primarily hydronium ions (H_3O^+), away
 from the interface. This leads to a reduction in the concentration of $[H^+]_S$ enabling higher
 surface charge (σ) that in turn produces a higher V_{oc} or a positive photovoltage (V_{ph}). The
 number of electron-holes pairs generation is directly proportional to the intensity of the
 light. However, not all generated carriers will lead to the charging effect as there will be
 surface or bulk recombination process, which depends on the interface structure and the
 energy of generated carriers³⁸. While the detailed analysis of these phenomena is beyond
 the scope of this manuscript, we notably observe that the number of generated charge
 carriers is directly proportional to the light intensity, expressed mathematically as
 $n_h \sim \beta_\lambda I_\lambda / E_\lambda$, where β_λ is the effective number of carriers generated per unit incident

photon, I_λ is the incident intensity, and E_λ is the energy of the photon. On the other hand,
the concentration of protons on the liquid side is given by the Boltzmann distribution³⁶
as $[H^+]_S \sim \exp(-\frac{e\phi}{k_B T})$, which influences the chemical potential according to the
relationship³⁰ $\phi \sim -\log([H^+]_S)$. Assuming the change in $[H^+]_S$ is directly proportional to
the change in n_h we can establish the dependence of photovoltage on intensity as
$V_{ph} \sim \Delta\phi \sim \log(\beta_\lambda I_\lambda / E_\lambda)$. We conducted measurements of photovoltage across various
monochromatic light wavelengths, ranging from 450 nm to 950 nm, as well as the full
solar spectrum, across a broad intensity range. The results reveal that the photovoltage
exhibits a clear logarithmic dependence on light intensity, with a linear trend observed
when plotted on a logarithmic scale (**Figure 4B**). Additionally, we can derive estimates
for the parameter β_λ at different wavelengths, as illustrated in **Figure 4C**, highlighting
the selectivity of the wavelengths.”

4. The characterization of electrode materials is insufficient. Only three concentration
gradients were analyzed in the doping characterization, making the results less conclusive
due to large experimental uncertainties. Furthermore, in Figure 3C, the measured
photovoltage of TiO₂ is consistently lower than that of Al₂O₃, and its performance
decreases with increasing concentration. This suggests that the concentration used was
not optimized for maximum activity. The author should redesign the experiments to
determine the optimal concentration and improve the accuracy of the results.

We thank the reviewer for this valuable and insightful comment. We acknowledge that
only three Si doping concentrations were included in the initial experimental set, which
may appear insufficient to fully map the concentration-performance relationship.

However, we would like to clarify that the trends of lower photovoltage with a higher
silicon dopant concentration, lower photovoltage of TiO₂ compared to Al₂O₃, and the
decrease in photovoltage with increasing electrolyte concentration—are not primarily due
to non-optimal concentrations or experimental artifacts. Rather, these behaviors are
governed by a consistent physical mechanism: the effect of initial surface charge and
interfacial capacitance on photocharging behavior. To address this more clearly, we have

added a dedicated section to the revised manuscript titled “Photocharging of Charged
Interfaces”, in which we explain:

- 1. How increasing the dopant concentration alters the capacitance and results in
lower photovoltage.
- 2. Why TiO₂, with its different surface charge properties and dielectric
characteristics, inherently produces a lower photovoltage under identical
conditions.
- 3. Why higher electrolyte concentrations show lower photovoltage.

We have also included additional experimental data in the revised Figure 3 to better
explain the above points (see below). We agree that an expanded experimental set could
better define the optimal concentration range; however, we believe that the observed
monotonic trends and supporting theoretical analysis provide a sufficiently robust
foundation for the conclusions drawn in the current manuscript. We have clarified this in
the revised discussion.

Finally, we would like to comment on the *large experimental uncertainties* mentioned by
the reviewer as we recognize that experimental uncertainty in characterizing electrode
materials and doping gradients can impact data interpretation. To effectively address this
issue, we have included an additional figure in the Supplementary Information (**Figure**
**S17, S28**) that clearly demonstrates the reproducibility of our measurements. Here we
measured the open-circuit voltage using the same electrode and electrolyte combination
during two distinct tests under identical environmental and testing conditions. The blue
curve shows the first test in which we measured V_{OC} during the heating and cooling cycle.
After completely drying out the sample, we started the second test by dispensing a fresh
volume of the same electrolyte concentration (red curve). We again measured the V_{OC}
while subjecting the system to the heating and cooling cycle. In addition, during this test
we periodically turned on and off the light input (solar light at an intensity of 100
267 mW/cm²). From the time-trace of the voltage in the red curve it is clear that the dark
values are perfectly consistent with the blue curve, while the periodic illumination leads
to significant and repeatable changes in V_{OC} . Overall, the results unequivocally show

consistent trends with minimal deviation, confirming the stability of our experimental
setup and the reliability of our measurements.

**Figure R3: Repeatability test:** Measured V_{oc} during the heating and cooling cycle (depicted by the blue curve).
Measured V_{oc} subjecting the system to the heating and cooling cycle with periodic activation and deactivation of
solar light at an intensity of $100\text{mW}/\text{cm}^2$ (red curve). The experiments were performed at different absolute times
after the sample was dried and replenished with fresh electrolyte.

**Changes to Manuscript**

**Figure 1: Role of Light for different coatings and salinity levels.** A) Time trace of the measured open-
 circuit voltage and capacitance at 1 mM KCl for two devices with different dielectric shells and low N-doping (1-20
 $\Omega\cdot\text{cm}$). The test was conducted under ambient conditions and 100 mW/cm² solar illumination (yellow-shaded
 region). B) Measured photovoltage under 100 mW/cm² intensity for samples with Al_2O_3 and TiO_2 shells (same low
 N-doped silicon core) at different concentrations of KCl. C) Measured open circuit voltage in the dark and the
 corresponding photovoltage measured as the device is heated and then allowed to cool down. D) Steady-state
 capacitance (blue bars) values for two devices with Al_2O_3 shell, but with different doping of silicon core (low N-
 doped: 1-20 $\Omega\cdot\text{cm}$ and high N-doped: <0.005 $\Omega\cdot\text{cm}$) at 1 mM KCl. The blue and red shaded region is measured at a
 surface temperature of $T_s = T_{\text{ambient}} = 25^\circ\text{C}$ and $T_s = 70^\circ\text{C}$, respectively.

Section 3.3

“We then measured the open circuit voltage of the same two low N-doping Si (1-20 $\Omega\cdot\text{cm}$)
 samples for different salinities (from 0.01 mM to 100 mM). **Figure 3B** presents the
 photovoltage, defined as $V_{ph} = V_{oc}^{\text{light}} - V_{oc}^{\text{dark}}$, for both materials across varying salinities.
 Our observations clearly demonstrate that the photovoltage recorded for samples with
 Al_2O_3 shells consistently outperformed that of the TiO_2 samples across all salinity levels
 tested. Our observations also reveal a significant decline in photovoltage as salinity levels
 increase for both sample types. Intriguingly, we further noted that High N-doped silicon

samples exhibit photovoltage values that are consistently over 150 mV lower than their
low N-doped counterparts (**Figure S16**). Finally, we assessed the photovoltage as a
function of combined heating/cooling and irradiation (**Figure 3C** and **SI 17**).
Interestingly, we observed that, contrary to the open circuit voltage which increases with
temperature, the photovoltage decreases with increasing temperature.

To better understand all of these critical findings and their relationship to the solid-liquid
surface charge, we conducted capacitance measurements under all these different
conditions. As shown in **Figure 1E**, the measured capacitance is linked to 3 capacitances
in series. Firstly, the space charge layer capacitance is directly proportional to silicon
doping³⁶. We thus compared the response of 2 samples with different silicon doping,
specifically Low N-doping (1-20 $\Omega\cdot\text{cm}$) and high N-doping (<0.005 $\Omega\cdot\text{cm}$). As shown in
**Figure 3D (blue bars, blue shaded areas)**, the capacitance for the low-doped silicon
sample (left two columns) was significantly lower than that of the high-doped silicon
samples (right two columns), in agreement with the expected trend. Secondly, the
capacitance of the oxide layer is given by $C_{ox} = \epsilon_0 \epsilon_r / d$, where ϵ_0 is the permittivity of free
space, ϵ_r is the dielectric constant of the material, and d is the thickness of the oxide. The
higher dark capacitance of TiO₂ compared to Al₂O₃ (**Figure 3A**, right axis) can be thus
related to the dielectric constant of anatase TiO₂ being 3-5 times higher than that of
Al₂O₃³⁷, explaining the consistent lower photovoltage for TiO₂ samples across all the
salinity values (**Figure 3B**). Thirdly, the EDL capacitance must increase with surface
charge (see **SI 14** for the analytical expression). By measuring the capacitance at different
electrolyte concentrations and temperatures (**SI 18**), we confirm that it rises with
increased electrolyte concentration and surface charge. This is also in agreement with the
observed trend of decreasing photovoltage with an increase in electrolyte concentration
due to higher capacitance (**Figure 3B**) as well as with the increase in capacitance with
temperature for both Low- and High N-doping of Si (**Figure 3D, blue bars, red**
**shaded areas**). Overall, these results confirm the key role of light-triggered alterations
of the chemical equilibrium at the interface via capacitive photocharging, concurrently
underscoring the complex interplay between doping levels, salinity, and capacitance.”

5. The study lacks an in-depth analysis of system stability. The author mentions a
“partially wetted region”, but what are the wettability characteristics of the electrode?
Will ions adsorb onto the electrode surface, potentially hindering its performance?
Additionally, how do crystal structure and morphology change before and after the
reaction? What is the rate of steam generation, and how does it affect electrode
stability? Long-term stability tests and corresponding results are highly recommended.

We sincerely thank the reviewer for raising important points regarding the stability and
wettability of the electrode, as well as potential effects of ion adsorption and
morphological changes.

**Wettability of the Electrode:**

The term “partially wetted region” does not accurately describe our system as our silicon
is completely wetted with liquid as evidenced by the immediate wicking and
disappearance of any droplet placed on the surface. This high wettability ensures
intimate contact between the electrolyte and the electrode, which we have now clearly
highlighted in the revised manuscript.

**Ion Adsorption and Its Effect on Performance:**

Concerning the ions, as we discussed in the prior comments and now hopefully more
clearly in the manuscript, ion adsorption and interfacial capacitance are the key
mechanisms driving the operation of the hydrovoltaic devices. Ions will adsorb
according to the chemical equilibrium of the surface (which is different for Al_2O_3 and
TiO_2) and instead of hindering the performance, they are the key factor making
operation possible. Due to the oxide layer on the surface, the silicon surface is
passivated and thus no Faraday reaction takes place under illumination as described in
detail in the response to comment#1.

**Morphology and Crystal Structure Stability:**

**Figure R4:** Scanning Transmission Electron Microscopy image of the Silicon Nanopillars showing passivating
oxide shell post operation.

Post-operation characterization using TEM-EDX confirms that the oxide passivation
shell remains preserved, indicating that the morphology and crystal structure of the
silicon nanostructure remain intact after device operation. There are no observed signs
of degradation or structural changes. These characterization data have been included in
the revised manuscript to demonstrate the structural stability of the electrode.

**Rate of evaporation:**

Regarding the next comment, we would like to emphasize that our device operates on
natural evaporation driven by molecule escape at the interface. Even when irradiating
the system, we are not driving photothermal steam generation as the main effect. The
rate of steam generation is significantly low for inducing any morphological changes in
the electrode. Corresponding evaporation rate data are now included in **SI 23** and
shown here in **R5**.

**Figure R5: Evaporation rate measurement.** A) Mass loss from the system due to evaporation. Prior to
 ~2000s, the the system was at ~35 oC, after ~2000 s (highlighted in pink), the system undergoes natural cooling to
 room temperature. Finally the system undergoes a temperature increase (highlighted in yellow). See SI 10 for
 details on temperature measurements and the explanation for different slopes during the temperature increase and
 decrease. B) The derivative of mass change. The pink and yellow lines show an exponential fit during the cooling
 and heating cycle, respectively. C) The corresponding temperature during cooling(pink) and heating (yellow).

**Long-term stability test:**

Lastly, we have now performed long-term stability tests. These revealed that the device
 consistently maintained a steady voltage output for 50 hours. Remarkably, this stable

performance persisted even under continuous light exposure, as demonstrated in Figure
R6 (Figure 6E), underscoring the robustness and reliability of the EDHV architecture.

A)

**Figure R6: Long-term stability testing of the device:** A) The open circuit voltage was measured
continuously for 45hrs using an Ag/AgCl electrode as utilized in this study. We illuminated the sample with light
(solar spectrum, 100 mW/cm²) in 1-hour on and off cycles for a total duration of around 15 hours. The inset
provides an enlarged view of the regions marked in purple and pink rectangles. The region marked in purple shows
an enlarged light off-on-off cycle, clearly showing the instantaneous response of the V_{OC} to illumination, which
demonstrate photocharging rather than photothermal effects. The region marked in red shows stability of the
measurement, with less than 2 mV change in 2 hours.

We appreciate the reviewer's valuable feedback, which allowed us to further strengthen
the discussion on electrode wettability, ion adsorption, morphological stability, and
long-term performance in our manuscript.

Reviewer #2 (Remarks to the Author):

The paper focuses on the efficacy of evaporation-driven hydrovoltaic (EDHV) devices,
considering a variety of driving forces, i.e., involving thermal and ion concentration
gradients, as well as illumination. Given an electrolyte as the medium of the devices, it
would be expected that potential differences would be generated.

At the very outset, the results and the analysis are/have been considered through
individual phenomenology while there would be considerable coupling between the
forces and inter-dependency.

We thank the reviewer for the careful reading of our manuscript and for highlighting
both the relevance of the topic and the complexity inherent in analyzing evaporation-
driven hydrovoltaic (EDHV) devices operating under multiple driving forces.

We would like to emphasize that our choice of the experimental platform was deliberate
to enable the unravelling of the complex interplay of different interfacial phenomena in
determining the hydrovoltaic device output, beyond the specificity of a given device. In
fact, contrary to prior works where a single material served all purposes, we established
a device architecture that decouples the three main components of an hydrovoltaic
system: (1) the true hydrovoltaic component, controlled by the chemical equilibrium at
the solid/liquid interface, (2) the electrolyte layer, where thermodiffusion can occur and
(3) the evaporating interface, affected by environmental conditions (humidity,
temperature). Our experimental results are thus not simply phenomenological, but they
rather disentangle the different mechanisms at play in all hydrovoltaic devices, their
interdependence represented by the equivalent electric circuit. Furthermore, going
beyond a simple photothermal effect to enhance evaporation rates, our results show that
heat and light input have major impacts on the solid/liquid interface due to the
temperature sensitivity of the surface group dissociation and the ions
desorption/adsorption driven by photocharging of the interface. These observations, to
the best of our knowledge, are new and have never been reported before. They can also
be generalized for a wide range of materials used in hydrovoltaic devices, accounting for
their specific chemical equilibrium and band structure.

To further emphasize the role of coupling between forces, we note that in real
operational conditions (i.e., non-open-circuit mode), the net current in the system
depends simultaneously on all driving forces and based on our model we can write the
expression for current flowing through the circuit as:

$$430 \quad I_L = \frac{I_{\Delta T} R_{Sol} (1 + v_f r_{3\phi} C_{tr}) + v_f \bar{\sigma} r_{3\phi} + V_{ph}}{R_{Si} + R_{Sol} (1 + v_f r_{3\phi} C_{tr}) + r_{3\phi} / W + R_C + R_L} \quad (M4)$$

Consequently, the charge stored (Q) in the capacitor (C_{tr}), is also clearly coupled to both
interfacial and ionic properties as shown below:

$$433 \quad Q = R_{Sol} C_{tr} (I_L - I_{\Delta T})$$

Based on all of the reviewers' comments, we acknowledge that our initial draft did not
highlight well the focus and main results of the work. In our revised manuscript, we
have strived to emphasize from the start our emphasis on the solid/liquid interface,
which motivates the chosen experimental architecture. In addition, following the
reviewer's comments, we have expanded both the theoretical framework and the
experimental validation to address the couplings explicitly and to demonstrate how the
observed performance enhancements can be predicted using our analytical expressions.

Consequently, the obtained voltages and power density would be sensitively dependent
on the particular system, environment and testing conditions. While the noted
phenomena have been analyzed in terms of performance metrics, it is unclear as to how
they would result in the observed enhancements of the voltage, power, etc. For instance,
could the observed voltage and power density be predicted from the indicated relations?

We now address in more detail the reviewer's comment about the environmental and
testing conditions as well as the voltage and power outputs predictions. It is clear that
environmental conditions like temperature and humidity play a crucial role, as they
directly influence interfacial charge dynamics and evaporation rates in a strongly
interdependent manner. Indeed, the system under investigation is fundamentally
governed by coupled physicochemical phenomena, including thermally induced
interfacial chemistry, photogenerated charge dynamics, electrolyte interactions, and
evaporation processes.

To address this complexity, we have now revised the manuscript to include a dedicated
subsection discussing the coupling of these interfacial processes and how they
collectively contribute to voltage and power generation. Notably, the observed nonlinear
trend of V_{oc} with temperature and light cannot be explained by a single mechanism
alone. Instead, a unified model is required that captures the synergy among:

- 1. Evaporation-induced flow (v_f)
- 2. Chemical potential differences (Φ)
- 3. Photovoltage from photocharging effects (V_{ph})

$$462 \quad V_{OC} = v_f \bar{\sigma} r_{3\phi} + \Phi (1 + v_f r_{3\phi} C_{tr}) + V_{ph} \quad (2)$$

This equation highlights how the evaporation rate (v_f) and chemical potential difference
(Φ), both of which are influenced by environmental conditions are coupled. An increase
in ambient temperature, for example, will lead to a higher temperature of silicon even
when no external heating is applied. This, in turn, will increase both the chemical
potential and evaporation velocity, and their product in the second term leads to the
observed quadratic increase in V_{oc} , as shown in **Figure 2C and S12**. Furthermore, we
include experimental evaporation rate data as a function of temperature (**Figure S23,**
**R7**), capturing the transition region and confirming that performance depends not only
on surface charge dynamics but also critically on evaporation rate.

**Figure R7: Evaporation rate measurement.** A) The derivative of mass change. The pink and yellow lines

*show an exponential fit during the cooling and heating cycle, respectively. B) The corresponding temperature*
*during cooling (pink) and heating (yellow).*

While the analytical expressions for V_{OC} and I_{evap} are based on simplified assumptions
(e.g., capacitive interfaces, equilibrium ion dynamics), we emphasize that they enable
semi-quantitative predictions of voltage and power output. For instance, Eq. M3 shows
that:

$$480 \quad I_{Evap} = W v_f (\bar{\sigma} + C_{tr} \Phi)$$

Both terms are experimentally tunable via temperature changes (**Figure S9 and S23**),
enabling predictive modeling of performance trends.

To better reflect this complexity, we have revised the manuscript to include a dedicated
subsection that discusses the interplay between different interfacial forces and how their
coupling contributes to voltage and power output. Specifically, we elaborate on the link
between temperature-dependent chemical equilibrium (through ΔH), the influence of
solvent and electrolyte on capacitance, light-driven photocharging, and the
electrokinetic contribution from evaporation-induced flow. We have also added
measurements concerning the change in evaporation rate with temperature in the **SI**
**23**.

A few more specific comments are indicated next.

The decoupling of the top evaporating surface from the nanostructures, and related
electrodes placed at the bottom, is not straightforward to understand, due to the
intervening electrolyte and related electrochemical effects.

We agree that the decoupling between the top evaporative interface and the
nanostructured silicon electrode at the bottom—mediated by the intervening
electrolyte—can be conceptually challenging due to the complex electrochemical
interactions involved. To address this and improve clarity, we have significantly revised
the manuscript's structure to highlight better how each region contributes to the overall
device behavior. The introduction of the intermediate electrolyte layer enables
systematic decoupling into three distinct functional regions, allowing us to isolate and

investigate the roles of evaporation-driven ion flow, interface-specific charge
accumulation, and photocharging effects independently. The revised manuscript now
follows a clear and logical progression:

*Introduction of Key Interfacial Processes* – We begin by identifying the central
interfacial phenomena that govern the device’s behavior. This section lays the
groundwork for understanding how each process can be modulated to affect the voltage
generation.

***Experimental Platform of the Decoupled System*** – Here we introduce the device
architecture and experimental platform that enable the spatial and functional separation
of evaporation, ion transport, and charge generation processes.

***Effects of Temperature Change on Chemical Equilibrium*** – We analyze how
temperature alters interfacial surface charge dynamics, leading to observable changes in
voltage.

*Effect of Irradiation on Chemical Equilibrium* – This section examines how light
influences chemical equilibrium and modulates voltage output.

***Photocharging of Charged Interfaces*** – We present detailed experiments that
show how capacitive photocharging varies with salinity and illumination, providing
critical insight into the photovoltaic contribution.

***Decoupling Strategy and Rationale for Equivalent Electrical Circuit*** –
Based on the physical and experimental insights, we introduce the equivalent circuit
model, explaining how it maps to each functional component of the device.

***Power Output Analysis Using Equivalent Electrical Circuit*** – We then use the
model to analyze voltage and power output, showing how the interplay between physical
parameters can be used to predict and optimize device performance.

We believe this restructuring improves the overall readability of the manuscript and
ensures that the novelty and logic of our approach are communicated clearly and
effectively.

Is there a Debye length related correlation, with respect to the various used
concentrations? Would not there be a voltage across the EDL rather than “along the
length of the NP”?

We appreciate the reviewer's comment and want to address the important points raised.
The phrasing in the original manuscript may imply an overly simplified view of the
system. To be clear, the electrical potential difference we measure is a direct
consequence of the formation of the electric double layer (EDL). However, the voltage
drop is due to a longitudinal imbalance in ionic chemical potential along the length of
the silicon nanopillar (SiNP) structures. This imbalance results from the asymmetries in
surface charge distribution between the top and bottom of the nanostructure (**Figure**
**R8**).

**Figure R8: Electrical potential distribution.** *A) (Left) Three-dimensional schematic of the triangular unit cell*
*of the hexagonally arranged NPs showing the geometrical parameters of the nanostructures (pillar diameter, D_{np} ;*
*pillar length, L_{np} ; and mean pore diameter, D_p). (Right) Annular cylindrical nanopore geometry was used for*
*simulations, including the calculated electrical potential distribution. B) Vertical cut plane of the simulated*
*cylindrical nanopore in (D) showing the counterion concentration distribution, with ion flux (left), and the electrical*
*potential distribution, with electric field lines (right). The bulk ionic concentration is $10 \mu\text{M}$ KCl for the given*
*simulation results. Reproduced: Device 2, 100287 (2024).*

As we explain in Supplementary Information (**SI 8**), the closed bottom of the
nanochannel introduces an intrinsic geometric and chemical asymmetry. This
asymmetry generates a persistent chemical potential gradient (Φ) along the length of the

nanostructure, even in the absence of convective or evaporative flow, leading to a
measurable open-circuit voltage (V_{OC}).

Regarding the correlation with Debye length, it is indeed true that the EDL thickness is
inversely correlated with electrolyte concentration. However, due to the chemical
equilibrium at the solid-liquid interface, the relationship between measured voltage and
concentration is not strictly monotonic with the Debye length correlation. We have
thoroughly discussed these aspects in our previous work. **Figure R8** clearly illustrates
the distribution of electrical potential and cation concentration, demonstrating a
distinct gradient of potential both across the EDL and along the length of the
nanopillars.

This has now been explicitly clarified in the manuscript to improve interpretability.

Changes to the manuscript

**Section 3**

“Due to the chemical equilibrium controlled surface charge, at the oxide-liquid interface
an EDL is formed, and any resulting imbalance in the ion distribution along the SiNP
length contributes to the measured electrical potential difference. Interestingly, we
previously observed that, due to the closed bottom surface of the nanochannel, the
studied geometry presents an intrinsic asymmetry in the surface-charge distribution
(see **SI 8**). As a result, even in the absence of an evaporation-induced flow, a chemical
potential difference (Φ) exists, and therefore, a V_{OC} can be measured between the top
electrode and the bottom SiNPs surface⁶.”

The time constants related to the temperature increase and decrease are not easy to
understand. Was the Peltier heater placed in the liquid? What accounts for the
stabilization of the temperature? How exactly does the convective heat transfer play a
role?, say considering the related heat transfer coefficient? What is the role of the
respective thermal conductivity values of the dielectrics used? Typically, TiO_2 has a
lower thermal conductivity.

We thank the reviewer for this thorough and constructive question.

To address the concerns raised, we have now clarified the experimental setup, added
relevant data and modeling, and expanded the discussion in the manuscript (**SI 10**).

**Heating Configuration:**

The Peltier heater was not placed in the liquid but was mounted beneath the SiNPs
electrode, in direct contact with the substrate. This configuration ensured that heating
was localized and directed upward toward the solid–liquid interface without direct
immersion of the heater in the electrolyte.

**Temperature Monitoring and Time Constants:**

We monitored the temperature evolution in real time using an infrared (IR) thermal
camera positioned above the system. We have now included the full temperature–time
traces for both heating and cooling cycles (**Figure S10a, R9**) and extracted the
corresponding time constants using exponential fits. These reveal the characteristic
thermal response of the system and are discussed in the revised manuscript.

**Figure R9: Temperature measurement of the system.** A) Measured temperature-time tract during heating
and cooling phases. The red and blue curves are the fit to the data points. The distinct value of the heating and
cooling time constants was obtained.

***Convective Heat Transfer Role:***

We performed a thermal circuit analysis of the system to evaluate the contribution of
convective heat losses to the environment. The analysis includes the convective heat
transfer coefficient (h) based on natural convection from the top surface of the liquid
and the electrode. This coefficient plays a crucial role in the cooling dynamics and
temperature stabilization, as it governs the rate of heat dissipation into the surrounding
air. We developed a Comsol model and critically assessed the heat transfer coefficients
h1 and h2 from the top surfaces (see SI 10 for more details).

***Thermal Conductivity of Dielectric Oxides:***

While it is correct that TiO₂ typically exhibits lower thermal conductivity (~2–10
W/m·K) than materials like Al₂O₃, in our geometry, the oxide layers are very thin and
do not serve as the primary pathway for thermal transport. Instead, the total thermal
resistance is primarily dominated by the bulk silicon and the liquid (which has a larger
thickness and a much lower thermal conductivity). Consequently, the overall thermal
response is *minimally affected by the oxide's intrinsic thermal conductivity*

Based on one-dimensional heat transfer analysis, we can use the cylindrical pin-fin
configuration to estimate the heat transfer coefficients considering the effective thermal
conductivity as shown below (see **SI 10** for more details):

$$k_{eff} = k_{Si} \left(1 + \frac{4k_{ox}t_{ox}}{k_{Si}D_{NP}} \right)$$

For $t_{ox} \approx 10 \text{ nm}$, $D_{NP} = 400 \text{ nm}$, $k_{Si} = \frac{W}{mK}$, and $k_{ox} = 1 - 50 \frac{W}{mK}$

$$\frac{4k_{ox}t_{ox}}{k_{Si}D_{NP}} \ll 1$$

$$k_{eff} \approx k_{Si}$$

*Therefore, the thermal conductivity of the oxide layer does not significantly affect*
*overall heat transfer coefficients due to its minimal thickness compared to the*
*diameter of the nanopillars.*

***Chemical vs. Thermal Contributions:***

Our analysis indicates that the observed response time and performance differences
between TiO₂ and Al₂O₃-modified electrodes are not governed by thermal transport,
but are more directly linked to thermally induced changes in interfacial chemical
equilibria, specifically affecting ion migration and surface potential dynamics.

**Section 3.1**

“...This behavior clearly stems from the distinct chemical properties of the two
materials. Furthermore, the thermal conductivity of the oxide surface—commonly low in
TiO₂—has minimal impact on the system's overall behavior (see SI 10).”

The interplay of the temperature and the ion concentration is unclear. For instance, can
the authors explain clearly what is meant by evaporation-induced ion streaming. To
what extent would there be a modification of the liquid diffusion constants of the
involved ions? The notion of “ion dynamics” needs to be further quantified!

We have now added a dedicated first section in the manuscript where we guide the
reader on the role of key interfacial phenomena in device performance. Here we
specifically introduce how heat and light, through changes in temperature and the
photovoltaic effect, can modify the system's behavior. In the later sections of the
manuscript, we proceed to the experimental demonstration together with the analysis of
their interplay.

How would temperature increase the surface charge linearly?

We thank the reviewer for this insightful question.

In our system, the observed linear relationship between temperature and surface charge
(**Figure R10**) arises from the thermally driven shift in the equilibrium of surface
ionization reactions. Specifically, temperature increases promote the dissociation of
surface hydroxyl groups (e.g., $\equiv\text{TiOH} \rightleftharpoons \equiv\text{TiO}^- + \text{H}^+$), thereby increasing the density of
surface charges. We note that the temperature dependence of the ionization equilibrium
is central to our analysis. In our study, we relate the temperature-dependent

equilibrium constant $K(T)$ to the thermodynamic parameter-enthalpy (ΔH)-via the
Van't Hoff equation

$$\ln K_1 - \ln K_2 = - \left[\frac{\Delta H}{RT_1} - \frac{\Delta H}{RT_2} \right]$$

Equivalently, we can write it in form that is given in the manuscript Eq. M1

$$K = K_0 \exp \left[- \frac{\Delta H}{R} \left(\frac{1}{T} - \frac{1}{T_0} \right) \right] \quad (M1)$$

We have clarified this interpretation in the revised manuscript and expanded the related
thermodynamic discussion. We have added a figure based on our COMSOL simulation
to evaluate the Surface charge as a function of temperature, showing linear dependence.

**Figure R10:** Surface charge dependence on temperature shown for different values of K_0

The dV/dT seems to be less sensitive when TiO_2 is used. Why would this be?

We appreciate the reviewer's insightful observation. The reduced dV/dT sensitivity
observed for TiO_2 can be directly attributed to its surface chemistry's thermodynamic
properties. Specifically, the enthalpy change (ΔH) associated with temperature-
dependent surface ionization reactions is lower for TiO_2 than for Al_2O_3 . This difference
results in a distinct thermal driving force for modulating surface charge with
temperature, consequently diminishing the temperature-voltage response (**Figure**
**R11**).

To substantiate this claim, we employed the COMSOL model to analyze the linear
regions of the experimental V_{oc} - T_s curves, allowing us to extract ΔH values for each
material. Our findings, now clearly presented in **Figure S9**, demonstrate that the
slopes of V_{oc} - T_s increase in correlation with higher ΔH values. This outcome
unequivocally supports our interpretation regarding the underlying chemical
equilibrium at the interface.

**Figure R11:** Chemical potential difference dependence on temperature shown for different values of ΔH . It shows a
linear trend whose slope is determined by ΔH , which is the material's surface chemistry thermodynamic properties.

Section 3.1

“We first use the COMSOL model to fit the linear part of the experimental V_{oc} - T_s
curves, that allow us to extract the value of ΔH for each material and electrolyte
concentration. This confirms that Al_2O_3 exhibits a higher ΔH than TiO_2 (Figure 2E, right
axis). This behavior clearly stems from the distinct chemical properties of the two
materials. Furthermore, the thermal conductivity of the oxide surface—commonly low in
TiO_2 —has minimal impact on the system's overall behavior (**SI 10**). Thus, the
temperature-dependent alterations in chemical equilibrium at the liquid-solid interface
play a critical role in enhancing ion migration, thereby providing a clearer insight into
the thermally induced dynamics of the system we have presented.”

Have the authors performed an electrochemical impedance spectroscopy (EIS) analysis?

We thank the reviewer for highlighting the relevance of electrochemical impedance
spectroscopy (EIS).

Yes, we have performed EIS measurements to better understand the interfacial
properties and capacitive behavior of the system. A full EIS scan was conducted over a
frequency range of 10^6 to 10^2 Hz (**Figure R12**). The resulting Bode and Nyquist plots
are provided in the Supplementary Information (**SI 15**).

A detailed analysis using the EIS was reported in our previous work, where we
established a clear dependence of ionic mobility on the observed open-circuit voltage.
Notably, the lowest-mobility cation, Li^+ , exhibits the strongest VOC peak. Through our
EIS measurements in 1 mM concentrations of LiCl , KCl , CsCl , KCl , KBr , and KI . We
determined the diffusivity, and consequently the ionic mobility, determined by fitting
the Nyquist plots with an appropriate electrical circuit. We found a significant variation
in diffusivity among different cations, compared to the minor differences among anions,
which unequivocally confirms that cations dominate ion transport in our studied
system, as anticipated in a scenario characterized by a negative surface charge.

Here is a sample of EIS data for 1mM KCl , presented below. For comprehensive details,
we direct the reviewers to our previous work (*Device 2, 100287 (2024)*).

Figure R12: Nyquist plot obtained from the EIS measurement in 1 mM KCl electrolyte.

Why was 1 kHz chosen? How was the capacitance monitored and estimated?

We thank the reviewer for this insightful question regarding the selection of
measurement frequency and capacitance monitoring methodology.

*1. Rationale for Selecting 1 kHz:*

To determine the optimal frequency for real-time impedance monitoring, we initially
conducted a full electrochemical impedance spectroscopy (EIS) scan over a frequency
range of 10^6 to 10^2 Hz. Based on the typical nature of the EIS plots, the high frequency
region is related to the charge kinetics while low frequency region is the diffusion-
controlled region. Furthermore, we analyzed the Bode plots obtained from this scan to
understand the system's characteristic impedance behavior. We noticed a transition for
both the phase and amplitude plots around 1kHz (**Figure R13 A**). Based on this
analysis, 1 kHz was selected as the monitoring frequency for time-resolved impedance
measurements (commonly referred to as single-frequency impedance (SFI)
monitoring)(Figure R13 B). This frequency captures the relevant interfacial dynamics
with sufficient signal stability.

*2. Capacitance Estimation Method:*

The capacitance was estimated using the imaginary component of impedance (Z'') at
the selected frequency, following the standard relation:

$$C = \frac{1}{2\pi f |Z''|}$$

$f=1kHz$ is the measurement frequency. This approach allows for the dynamic tracking of
capacitive behavior in response to changes in illumination or temperature conditions.

We have clarified these points in the revised manuscript and have added supplementary
information (**SI 15**) showing the Bode plot EIS analysis that guided the frequency
selection.

**Figure R13: Impedance measurements.** A) Bode plot obtained from the EIS analysis. B) Measured time trace
 of the capacitance at different frequencies. The temperature varied during the heating and cooling regimes.

Light induced charge increase could occur through a variety of factors and the
 possibilities should be enumerated better. Can the authors consider the possibility of
 electrolysis?

We appreciate the reviewer's suggestion to consider electrolysis as a potential
 contributor to the observed light-induced charge increase. We have carefully evaluated
 this possibility and conclude that electrolysis, specifically oxygen or hydrogen evolution
 reactions, is highly unlikely under our experimental conditions.

Our system operates in a dilute KCl electrolyte with a maximum observed open-circuit
 voltage of ~ 0.5 - $1V$, which are significantly below the threshold potentials required for
 water electrolysis. For silicon-based electrodes, the oxygen evolution reaction typically
 requires an overpotential of at least 1.2 - $1.4 V$, even under optimized electrode and
 electrolyte conditions (*Chem. Soc. Rev.*, 2019, 48, 2158-2181).

To further support this, we conducted cyclic voltammetry (CV) measurements under 1
 sun (1 kW/m^2) illumination across the 0 - $1 V$ range. The CV curves show no peaks and
 onset potentials indicative of faradaic processes, such as water splitting or redox
 reactions, confirming that no faradaic electrochemical transformations are occurring in
 the system.

Instead, the observed behavior is consistent with a capacitive charging mechanism,
driven by interfacial charge accumulation under illumination. This is further reinforced
by the shape of the CV curves (as shown in **Figure R14 B and C**), which exhibit
rectangular profiles typical of non-faradaic, capacitive systems.

We have included this clarification in the revised manuscript, emphasizing
electrochemical behavior and excluding the likelihood of electrolysis under the applied
conditions.

Section 2.1

“This also highlights the importance of the oxide layer in passivating the silicon surface³²
and preventing any chemical reaction between silicon and water. Our
photoelectrochemical test on the device indeed shows no evidence of faradaic activity,
thereby confirming a capacitive charging, rather than faradaic, process (SI 3).”

**Figure R14:** A) Photoelectrochemical behavior of p+-Si|NiOx, n-Si|NiOx and np+-Si|NiOx electrodes in 1.0 M
potassium hydroxide under simulated AM 1.5G illumination (100 mW cm⁻²) (Chem. Soc. Rev., 2019,48, 2158-2181).
B) Current-Voltage curves under AM 1.5 G illumination of the SiNP electrode in 1 mM KCl. C) Current-Voltage
curves of the SiNP electrode under increasing temperature in 1 mM KCl.

The role of salinity, which is not clearly defined, should be further elucidated.

We appreciate the reviewer’s comment highlighting the need for a more precise
explanation of the role of salinity in our system.

In the revised manuscript, we have expanded our discussion to clarify this relationship.
Specifically, we observed that increasing the salinity (electrolyte concentration) leads to

a notable decrease in photovoltage output for all different sample types studied. To better
understand this trend, we performed capacitance measurements under dark conditions
(pre-illumination) and found that dark capacitance increases with higher electrolyte
concentrations.

This is consistent with the well-known Debye correlation, which states that higher ionic
strength compresses the electrical double layer, resulting in a greater surface charge
density and, consequently, a higher double-layer capacitance.

Our results establish a clear inverse relationship:

*Higher electrolyte concentration* → *Higher surface charge* → *Higher capacitance* →
*Lower photovoltage*

We have added a dedicated section (**Section 3.3**) in the manuscript to clarify the role of
salinity.

When Silicon doping is considered, it would have been worthwhile to use n- and p-doping
of the same impurity concentration/resistivity. The magnitudes of the voltage change, say
35 mV, should be explained in terms of the band structure, Else, the values just seem
incidental to the performed experiment.

We thank the reviewer for this comment. We acknowledge that only three doping
concentrations were included in the initial experimental set, which may appear
insufficient to map the concentration-performance relationship fully.

However, we would like to clarify that the trends of lower photovoltage with a higher
silicon dopant concentration, aligns with the observed trend of lower photovoltage of
TiO₂ compared to Al₂O₃, and the decrease in photovoltage with increasing electrolyte
concentration. These are not due to non-optimal concentrations or experimental artifacts.
Rather, these behaviors are governed by a consistent physical mechanism: the effect of
initial surface charge and interfacial capacitance on photocharging behavior. To address
this more clearly, we have added a dedicated section to the revised manuscript titled
“Photocharging of Charged Interfaces”.

**Section 3.3**

“...Intriguingly, we further noted that High N-doped silicon samples exhibit photovoltage
values that are consistently over 150 mV lower than their low N-doped counterparts
(**Figure S16**). Finally, we assessed the photovoltage as a function of combined
heating/cooling and irradiation (**Figure 3C and SI 17**). Interestingly, we observed that,
contrary to the open circuit voltage, which increases with temperature, the photovoltage
decreases with increasing temperature.

To better understand all of these critical findings and their relationship to the solid-liquid
surface charge, we conducted capacitance measurements under all these different
conditions. As shown in **Figure 1E**, the measured capacitance is linked to 3 capacitances
in series. Firstly, the space charge layer capacitance is directly proportional to silicon
doping³⁶. We thus compared the response of 2 samples with different silicon doping,
specifically Low N-doping (1-20 $\Omega\cdot\text{cm}$) and high N-doping ($<0.005 \Omega\cdot\text{cm}$). As shown in
**Figure 3D (blue bars, blue shaded areas)**, the capacitance for the low-doped silicon
sample (left two columns) was significantly lower than that of the high-doped silicon
samples (right two columns), in agreement with the expected trend...”

We agree that an expanded experimental set could better define the optimal
concentration range; however, the observed monotonic trends and supporting theoretical
analysis provide a sufficiently robust foundation for the conclusions drawn in the current
manuscript. We have clarified this in the revised discussion.

When it is stated that “surface charges of either electrode are perturbed by thermal
effects or light irradiation”, the meaning is unclear, with respect to whether the charges
come off the surface, neutralized, contribute to the EDL, etc.?

Thank you for this valuable comment. We acknowledge that the original phrasing was
unclear and lacked specificity regarding the underlying physical processes.

In response, we have revised the manuscript to provide a clearer and more precise
explanation in the first section, where we now explicitly describe the relevant interfacial
processes.

**Section 2.1**

**Introduction of Key Interfacial Processes**

“...Overall, the chemical equilibrium at the interface is strongly dependent on the
material properties as well as the temperature and the ion concentration in the EDL.
Light and heat triggers can have a multi-faceted influence on the interfacial chemical
equilibrium, therefore affecting device behavior and performance...”

Are the authors claiming that the P-type Si is inverted, i.e., “through electron
accumulation at the interface” – which is highly unlikely in the presented experiments!
The instances of band bending and their causes should be reviewed again by the
authors!

We thank the reviewer for pointing out this important clarification regarding the band
bending and carrier dynamics in P-type silicon.

We acknowledge that the original wording in the manuscript may have led to confusion.
To address this, we have carefully revised the relevant text to clarify that we are *not*
claiming inversion of the P-type silicon under illumination. Rather, the observed band
bending arises due to Fermi-level equilibration at the solid–liquid interface, which is
influenced significantly by surface states and oxide-mediated interface properties.

Specifically, for P-type silicon, the band bending remains downward, consistent with the
well-established Fermi-level pinning effects at the Si/SiO₂/electrolyte interface (**Figure**
**R15**). Under illumination, photogenerated electrons are driven toward the interface due
to this band bending and accumulate there, contributing to the observed photovoltage.
Importantly, this accumulation does not indicate inversion but reflects the interfacial
charge dynamics inherent to such systems.

We have also included references that support this explanation and reflect the known
physical behavior:

- Mizsei, J. *Vacuum*, 67 (2002), 59–67.

- • Li, J. et al., *Nat. Commun.* 12, 4998 (2021).
- • “pH Sensor Using Protein-Mediated Gold Nanocrystal Array.”

**Section 2.1**

“...The equilibration of the Fermi level across the solid–liquid interface, primarily driven
 by surface states, determines the band bending at the silicon-oxide interface (Figure 1A,
 panel iii). Consequently, under illumination, photogenerated charges (electrons or
 holes) accumulate at the interface...”

**Figure R15:** Band bending of silicon due to Fermi level pinning. (Left) Downward band bending of p-type silicon.
 (Right) Upward band bending of p-type silicon.

We hope this revision satisfactorily addresses the reviewer’s concern, and we thank
 them again for drawing attention to this important point.

Similarly, the “logarithmic” variation of the photovoltage with solar intensity is
 ambiguous!

We appreciate the reviewer’s feedback regarding the description of the photovoltage
 dependence on solar intensity.

We agree that the term “logarithmic” was used too loosely in the earlier version of the
 manuscript.

Now, a separate section has been added to explain clearly the physics and the
 dependence of photovoltage on the intensity.

**Figure 4: Role of Light intensity and wavelength selectivity.** B) Measured photovoltage for different
 monochromatic incidences as a function of light intensity, plotted in logarithmic scale. I_0 is equal to 100 mW/cm².
 The points corresponding to solar is the measured photovoltage for the full solar spectrum. The experimental data
 points are shown in circle, while the line is the linear fit.

Section 3.3

“To further investigate the interplay between photocharging and interfacial processes, we
 finally examined how variations in light intensity influence the photovoltage response. As
 shown in **Figure 4D**, due to capacitive effect, accumulating holes at the silicon-oxide
 interface drive the dissociated cations of water, primarily hydronium ions (H_3O^+), away
 from the interface. This leads to a reduction in the concentration of $[H^+]_S$ enabling higher
 surface charge (σ) that in turn produces a higher V_{OC} or a positive photovoltage (V_{ph}). The
 number of electron-holes pairs generation is directly proportional to the intensity of the
 light. However, not all generated carriers will lead to the charging effect as there will be
 surface or bulk recombination process, which depends on the interface structure and the
 energy of generated carriers³⁸. While the detailed analysis of these phenomena is beyond
 the scope of this manuscript, we notably observe that the number of generated charge
 carriers is directly proportional to the light intensity, expressed mathematically as
 $n_h \sim \beta_\lambda I_\lambda / E_\lambda$, where β_λ is the effective number of carriers generated per unit incident
 photon, I_λ is the incident intensity, and E_λ is the energy of the photon. On the other hand,
 the concentration of protons on the liquid side is given by the Boltzmann distribution³⁶
 as $[H^+]_S \sim \exp(-\frac{e\phi}{k_B T})$, which influences the chemical potential according the
 relationship³⁰ $\phi \sim -\log([H^+]_S)$. Assuming the change in $[H^+]_S$ is directly proportional to

the change in n_h we can establish the dependence of photovoltage on intensity as
$V_{ph} \sim \Delta\phi \sim \log(\beta_\lambda I_\lambda / E_\lambda)$. We conducted measurements of photovoltage across various
monochromatic light wavelengths, ranging from 450 nm to 950 nm, as well as the full
solar spectrum, across a broad intensity range. The results reveal that the photovoltage
exhibits a clear logarithmic dependence on light intensity, with a linear trend observed
when plotted on a logarithmic scale (**Figure 4B**). Additionally, we can derive estimates
for the parameter β_λ at different wavelengths, as illustrated in **Figure 4C**, highlighting
the selectivity of the wavelengths.”

The definition of transfer capacitance must be clearly quantified. What is the related
dielectric permittivity and charge storage distance?

Thank you for this important comment. We agree that a clear definition and
quantification of the transfer capacitance is essential for accurately describing the
underlying ion–electrode interactions and their role in voltage generation.

In response to the reviewer’s comment, we have now derived an expression for the
transfer capacitance that accounts for the intricate geometry of the various interfacial
charge storage layers.

As illustrated in **Figure R16**, the total transfer capacitance is clearly decomposable into
contributions from both the bottom and top regions, represented by C1 and C2,
respectively.

We begin by identifying the interfacial region at the bottom electrode. The yellow
rectangle in **Figure R16** highlights both flat and cylindrical regions. We can accurately
estimate the capacitance using the established formulations for flat plate and cylindrical
capacitors, as detailed below.

$$C_1 = \frac{2\pi\epsilon_0\epsilon_r N L_{np}}{\log\left(\frac{d_{np} + 2\lambda_D}{d_{np}}\right)} + \frac{\epsilon_0\epsilon_r}{\lambda_D}$$

Where N represents the density of Nanopillars per unit area of the sample. L_{np} and d_{np}
are the length and diameter of the nanopillars. $\epsilon_0\epsilon_r$ respectively, of the dielectric

permittivity of the electrolyte. By considering a hexagonal arrangement of pillars with
pitch, p , and diameter d_{np} , we can estimate the number density as:

$$N = \frac{2}{\sqrt{3}p^2}$$

Furthermore, the Debye length, which can be used as the size of the capacitor on the
electrolyte side, is expressed as:

$$\lambda_D = \sqrt{\frac{\epsilon_0 \epsilon_r k_B T}{4\pi e^2 C_b}}$$

Where, C_b is the bulk ionic concentration of the electrolyte, e is the electronic charge, k_B
is the Boltzmann constant, T is the temperature in Kelvin.

Next we consider the interfacial region in the upper section, which consists of three
distinct components, as illustrated in **Figure R16**:

- 1. The flat portion of the top electrode surface
2. The cylindrical segment of the top electrode surface
3. Two segments representing the spherical region at the liquid-vapor meniscus
interface

**Figure R16:** Transfer capacitance estimation by identifying different interfacial regions of the bottom and top
 segments of the system.

By considering the contributions of each component, we can derive an expression for
 capacitance that encompasses the capacitance formulations for the flat plate, cylindrical,
 and spherical shells with radius 'a'. This comprehensive approach ensures that we
 capture the full complexity of the system and its impact on performance.

$$C_2 = \frac{1}{\pi a^2} \left[\frac{2\pi\epsilon_0\epsilon_r L_e}{\log\left(\frac{d_e + 2\lambda_D}{d_e}\right)} + \frac{\epsilon_0\epsilon_r \pi d_e^2}{\lambda_D} + 2\pi\epsilon_0\epsilon_r \frac{a(a + \lambda_d)}{\{(a + \lambda_d) - a\}} \right]$$

As $a \gg \lambda_D$ we can simplify it to get:

$$C_2 = \frac{1}{\pi a^2} \left[\frac{2\pi\epsilon_0\epsilon_r L_e}{\log\left(\frac{d_e + 2\lambda_D}{d_e}\right)} + \frac{\epsilon_0\epsilon_r \pi d_e^2}{\lambda_D} + 2\pi\epsilon_0\epsilon_r \frac{a^2}{\lambda_D} \right]$$

Further simplification led to:

$$C_2 = \frac{2\pi\varepsilon_0\varepsilon_r L_e}{\log\left(\frac{d_e + 2\lambda_D}{d_e}\right)} \frac{1}{\pi a^2} + \frac{\varepsilon_0\varepsilon_r}{\lambda_D} \frac{d_e^2}{4a^2} + \frac{2\varepsilon_0\varepsilon_r}{\lambda_D}$$

By considering that $d_e^2 \ll a^2$ as $\pi a^2 = 1\text{cm}^2$, and $d_e = 0.1\text{ cm}$, we can further simplify the
expression by neglecting the second term in the above expression

$$C_2 = \frac{2\pi\varepsilon_0\varepsilon_r L_e}{\log\left(\frac{d_e + 2\lambda_D}{d_e}\right)} \frac{1}{\pi a^2} + \frac{2\varepsilon_0\varepsilon_r}{\lambda_D}$$

Finally, as noted above, $\pi a^2 = 1\text{cm}^2$. If all physical properties are reported in cm, can
write as the final expression for the capacitance C_2 :

$$C_2 = \frac{2\pi\varepsilon_0\varepsilon_r L_e}{\log\left(\frac{d_e + 2\lambda_D}{d_e}\right)} + \frac{2\varepsilon_0\varepsilon_r}{\lambda_D}$$

We repeat the final expression relevant for estimating the transfer capacitance below:

$$C_1 = \frac{2\pi\varepsilon_0\varepsilon_r N L_{np}}{\log\left(\frac{d_{np} + 2\lambda_D}{d_{np}}\right)} + \frac{\varepsilon_0\varepsilon_r}{\lambda_D}$$

$$C_2 = \frac{2\pi\varepsilon_0\varepsilon_r L_e}{\log\left(\frac{d_e + 2\lambda_D}{d_e}\right)} + \frac{2\varepsilon_0\varepsilon_r}{\lambda_D}$$

Using Taylor expansion of natural logarithm function, we can write:

$$\log(x) = x - \frac{x^2}{2} + \frac{x^3}{3} - \frac{x^4}{4} \dots$$

Neglecting higher-order terms as $\lambda_D/d_{np} \ll 1$, (Here we keep the terms up to third order,
as at very low concentration, Debye length can be $\sim 100\text{ nm}$, which is similar to the
diameter of the nanopillars, $\sim 300\text{-}400\text{ nm}$. We can simplify the expression for C_1 as:

$$C_1 = \frac{\varepsilon_0 \varepsilon_r}{\lambda_D} \left[1 + \frac{\pi N d_{np} L_{np}}{1 - \frac{2\lambda_D}{d_{np}} + \frac{4}{3} \frac{\lambda_D^2}{d_{np}^2}} \right] = \frac{\varepsilon_0 \varepsilon_r}{\lambda_D} (1 + K_1)$$

K1 depends on geometrical parameters of the nanopillar array and the Debye length as:

$$K_1 = \frac{\pi N d_{np} L_{np}}{1 - \frac{2\lambda_D}{d_{np}} + \frac{4}{3} \frac{\lambda_D^2}{d_{np}^2}}$$

Neglecting non-linear terms as $\frac{\lambda_D}{d_{np}} \ll 1$, we can simplify the expression for C2 as:

$$C_2 = \frac{\varepsilon_0 \varepsilon_r}{\lambda_D} \left[2 + \frac{\pi d_e L_e}{1(cm^2)} \right] = \frac{\varepsilon_0 \varepsilon_r}{\lambda_D} (2 + K_2)$$

K2 depends on geometrical parameters of the electrodes, its meniscus shape, and the
Debye length as:

$$K_2 = \frac{\pi d_e L_e}{\pi a^2}$$

We can finally obtain the expression for transfer capacitance that depends on the
geometrical parameters of the bottom nanostructures as well as the size of the top
electrode and the features of the related meniscus regions.

$$C_{tr} = \frac{\varepsilon_0 \varepsilon_r}{\lambda_D} \left[\frac{(1 + K_1)(2 + K_2)}{3 + K_1 + K_2} \right]$$

We have added the above analysis for estimating transfer capacitance in the
supplementary information of the manuscript (SI 22).

A statement such as “thermal motion induces electron transfer between the solid's
atoms and the water molecules, facilitated by overlapping of their electron cloud” is
imprecise.

Thank you for pointing out this important issue. We agree that the original phrasing was
imprecise and could lead to misinterpretation of the underlying mechanism. We have

revised the text in the manuscript to more accurately reflect the physical processes
involved. The modified text now reads:

**Section 2.1**

“The SiNP’s solid-liquid interface, in particular, plays a key role in the system behavior
and performance due to the presence of a net surface charge⁶, which can originate from
both electronic and ionic contributions^{27,28}. In this work, we focus on the ionic
contribution arising from dissociation reactions on the surface...”

For the posited “ionization reactions”, what is the standard redox potential/s and how
does it compare with the overpotentials or observed voltage?

We appreciate the reviewer’s comment and the opportunity to clarify the nature of the
ionization reactions discussed in our study.

The surface dissociation reactions we refer to—such as the deprotonation of surface
hydroxyl groups or adsorption/desorption equilibria of ionic species—do not involve
electron transfer between species. As such, they do not constitute redox reactions, and
therefore, the definition of a standard redox potential is not meaningful or applicable in
this context.

Instead, these processes are best described using acid–base equilibrium
thermodynamics, where the relevant thermodynamic potential is the standard Gibbs
free energy change, defined as:

$$1010 \quad \Delta G^\circ = -RT \ln K$$

Here, K is the equilibrium constant of the surface dissociation reaction, R is the
universal gas constant, and T is the temperature in Kelvin. This contrasts with redox
reactions, for which the Gibbs free energy is defined as:

$$1014 \quad \Delta G^\circ = -nFE_{redox}$$

where n is the number of electrons transferred, F is Faraday’s constant, and E_{redox} is the
standard redox potential. Since no electron transfer is involved in the processes under
discussion, E_{redox} does not apply.

Additionally, we note that the temperature dependence of the ionization equilibrium is
central to our analysis. In our study, we relate the temperature-dependent equilibrium
constant $K(T)$ to the thermodynamic parameter-enthalpy (ΔH)-via the van't Hoff
equation:

$$1022 \quad \ln K_1 - \ln K_2 = - \left[\frac{\Delta H}{RT_1} - \frac{\Delta H}{RT_2} \right]$$

This expression forms the basis of Eq. M1 in our manuscript and allows us to extract the
enthalpy of surface reactions from experimental temperature-dependent voltage data.

$$1025 \quad K = K_0 \exp \left[-\frac{\Delta H}{R} \left(\frac{1}{T} - \frac{1}{T_0} \right) \right] \quad (M1)$$

The role of solvents must be considered through the respective polar and/or non-polar
attributes.

We thank the reviewer for highlighting the importance of solvent properties, particularly
the role of polar versus non-polar characteristics, in influencing the device performance.

In our system, the charge flux associated with evaporation-driven ion transport is
governed by the relation:

$$1033 \quad I_{Evap} = v_f (\rho_f L_{3\phi} W + C_{tr} \Phi W) = W v_f (\bar{\sigma} + C_{tr} \Phi) \quad (M3)$$

Polar solvent molecules play a crucial role in modulating I_{Evap} (Joule 6, 690–701
(2022)), as well as ionic species such as H^+ , OH^- , K^+ , and Cl^- through $\bar{\sigma}$ and Φ
respectively.

To experimentally assess the solvent's role, we conducted comparative measurements
using two different polar solvents—deionized water and ethanol—as well as 1mMKCl in
Deionized water, under otherwise identical conditions. The results show a clear
difference in open-circuit voltage (V_{oc}):

- • For DI water: $V_{oc} = 0.552 \text{ V}$ (25 °C) and 0.690 V (65 °C)

- • For ethanol: $V_{oc} = 0.195 \text{ V}$ (25 °C) and 0.336 V (65 °C)
- • For 1 mM KCl in DI water: $V_{oc} = 0.652 \text{ V}$ (25 °C) and 0.840 V (65 °C)

These results, now presented in Figure S21 of the Supplementary Information, highlight
the significant influence of solvent polarity and the ionic species on surface charge
interaction and charge flux.

We also note that our electrolyte does not contain any non-polar components, and thus
the role of non-polar solvents is beyond the scope of the current study. This clarification
and the supporting data have been added to the revised manuscript.

**Section 3.4**

“To confirm the role of polar solvent molecules in the process, we tested the device using
water or ethanol under otherwise identical conditions. The measured VOC values for DI
water and ethanol at T_s equal to 25 °C (65 °C) are 0.552 V (0.690 V), and 0.195 V (0.336
1054 V), respectively (see SI 21 for more details on the effect of solvent), confirming a clear
dependence on the polar solvent molecules interaction with the charged surfaces.”

What is the assumed value of the K_a in Eqn. (3)?

We thank the reviewer for this question.

The assumed value of the acid dissociation constant used in Equation (3) ranges
between 5 and 10, depending on the specific surface chemistry of the oxide interface.
This range is consistent with reported values for surface hydroxyl groups on common
oxide materials such as SiO_2 , Al_2O_3 , and TiO_2 , which typically exhibit surface
protonation/deprotonation equilibria in this range. We have now clarified this point in
the main text and also direct readers to **Table S29** in the Supplementary Information,
where the assumed values used for the calculations are explicitly listed.

The authors have come up with an interesting electrical circuit model, in Figure 4(B).
However, the coupling of the individual phenomena should be clearly indicated as well.

We thank the reviewer for highlighting this important point regarding the clarity of the
coupling between individual physical processes in our equivalent electrical circuit model
(now Figure 5B). In the revised manuscript, we have now explicitly discussed the model
to indicate the coupling between key physical phenomena in **section 3.4** of the
manuscript. We have also derived the expression for transfer capacitance, which shows
the coupling between the top and bottom electrodes and the intermediate electrolyte
layer (see **SI 22**).

The relating of an evaporation velocity to the surface charge is unclear, given that the
surface charge would be bound to the surface.

We appreciate the reviewer’s thoughtful observation and agree that the terminology in
the initial manuscript version may have led to confusion.

To clarify, the “surface charge” was wrongly referenced in the discussion of evaporation
velocity. We wanted to refer to the free space charge per unit area of the meniscus
region—that is, the net mobile charge density accumulated near the contact line because
of interfacial chemical potential gradients and ion accumulation.

We have made the following changes in the manuscript.

“The total space charge density per unit area of the meniscus region is the sum of the self-
charge density $\bar{\sigma} = \rho_f L_{3\phi}$ (when the electrode was inserted in a bulk electrolyte)² and the
charge density arising due to intermediate electrolyte layer that is equal to transfer
capacitance times the chemical potential difference ($C_{tr}\Phi$). Note that $\bar{\sigma}$ is not the bound
surface charge, but instead free space charge per unit volume ρ_f , multiplied by length of
the wetted portion of electrode exposed to air, denoted by $L_{3\phi}$ (refer Figure S5). The term
I_{Evap} is equal to total charge times v_f , which is equal to the evaporative mass flux (SI 5 and
23) divided by density of water.”

$$I_{Evap} = v_f(\rho_f L_{3\phi} W + C_{tr}\Phi W) = W v_f(\bar{\sigma} + C_{tr}\Phi) \quad (M3)$$

What is the role of electroosmosis?

We thank the reviewer for raising this important point.

Electroosmosis typically requires an external electric field on the order of ~ 100 V/cm or
higher to induce significant fluid flow across charged interfaces. In our system, however,
the maximum voltage generated is below 1 V, which translates to an internal field that is
orders of magnitude smaller than that typically required for measurable electroosmotic
flow. As such, we believe that electroosmosis is unlikely to contribute significantly to the
observed fluid dynamics or interfacial phenomena in our setup. Nonetheless, we
acknowledge that localized electric fields and nanoscale interfacial effects could, in
principle, play a role under certain configurations. Thus, the role of electroosmosis was
not explicitly analyzed in this study, and it remains an interesting direction for future
investigation.

There should be a load-line analysis for the use of the load resistances for the power
estimations.

We thank the reviewer for this valuable suggestion. We agree that performing a load-
line analysis provides a comprehensive understanding of the device's power output and
ensures accurate estimation of maximum power transfer conditions.

In response, we have now included a detailed load-line analysis in the revised **SI 26**. In
this analysis, we systematically varied the external load resistance across a wide range
and estimated the corresponding output voltage and current. This enables us to obtain
the power output at a given loading condition. Additionally, we present the measured
current–voltage (I–V) characteristics at different temperatures to highlight the
influence of thermal conditions on device performance and load matching. These
analyses collectively provide a comprehensive view of the electrical performance of our
device under realistic operating conditions.

We have added the analysis below (Figure R 17) and in the supplementary information
(**SI 26**) believe this addition significantly strengthens the manuscript and provides a
more robust foundation for assessing the power generation capabilities of the EDHV
system.

Figure R17: Load line analysis of the device tested under different temperatures.

evaporation-rate-dependent contact potential difference between the wet and dry sides
 of

We apologize to the reviewer. Due to some missing text, it was not clear what the
 question was.

Many typos should be corrected through the manuscript, e.g., not sure how tunneling is
 involved, as indicated in the caption of Fig. 1(c), as well as several unclear phrases and
 statements, e.g., “The surface nature is crucial in this relationship”, “appears
 logarithmic-like in nature”, etc.

Thank you for this important observation. We carefully proofread the entire manuscript
 and corrected all identified typographical errors and awkward phrasing.

Regarding the specific examples:

Caption of Fig. 1(c): We acknowledge the confusion regarding the term "tunneling." The
 term was inaccurately used. We revised the caption to accurately reflect the mechanism
 being illustrated.

Phrase “The surface nature is crucial in this relationship”: We rephrased the contents to
 clarify the specific role of surface properties.

Phrase “appears logarithmic-like in nature”: We revised this to explicitly state the
 observed behavior.

All revised phrases are now more precise and scientifically appropriate. These changes
have been marked in red in the manuscript.

Caption of Fig. 1(c):

Original: "Scanning tunneling electron microscopy (STEM) image..."

Revised: "Scanning transmission electron microscopy (STEM) image...."

Revised unclear sentence:

Original: "The surface nature is crucial in this relationship."

Revised: "...This behavior clearly stems from the distinct chemical properties of the two
materials..."

Revised vague language:

Original: "...appears logarithmic-like in nature."

Revised: "...The results reveal that the photovoltage exhibits a clear logarithmic
dependence on light intensity, with a linear trend observed when plotted on a
logarithmic scale..."

In summary, the paper indicates fascinating interplay of light and heat in an electrolyte,
with respect to the generation of electrical voltages and power. However, the
interdependence of the various forces is not well explained. Further, the models should
indicate or come close to predicting the observed voltage and power values. The authors
should then aim, in the next revision, to address these shortcomings.

We thank the reviewer for the interest in our work, we hope that the revisions have
clarified the interdependence of the various forces and the models are now satisfactory.

Reviewer #3 (Remarks to the Author):

I have carefully reviewed this manuscript. The authors have conducted innovative
research on hydrovoltaic systems. By introducing an intermediate liquid layer, the
authors divided the hydrovoltaic system into a surface evaporation layer, an

intermediate ion-conducting layer, and a bottom silicon nanostructure layer, enabling a
relatively independent investigation of each component's function. Through a
combination of experimental and numerical simulations, the manuscript proposes an
equivalent circuit model for this system and assigns corresponding circuit components
and parameters to each functional part. By altering the system's temperature,
illumination conditions, and silicon nanostructure surface properties, the study reveals
mechanisms of ion dynamics, such as light-induced photocharging and thermally-
enhanced chemical potential. Through optimization of these conditions, the system
achieves an open-circuit voltage of 1V and an output power of 0.25W/m². Overall, the
manuscript presents a rich and innovative study. However, due to concerns regarding
clarity and other aspects of the manuscript, I recommend major revisions before
publication. Below are my specific comments:

We sincerely thank the reviewer for their thoughtful and detailed evaluation of our
manuscript. We are especially grateful for the recognition of the novelty in our
approach—particularly our strategy of introducing an intermediate liquid layer to
decouple and systematically analyze the contributions from different components of the
hydrovoltaic system. We appreciate the reviewer's acknowledgment of our integration of
experimental results with numerical simulations and the development of an equivalent
circuit model to elucidate the underlying ion dynamics. We also thank the reviewer for
highlighting the significance of our findings in achieving high open-circuit voltage and
power output. In response to the concerns raised regarding clarity and other aspects, we
have carefully revised the manuscript to address each point in detail, as outlined below.

1. Lack of clarity in logic. The study's innovation lies in introducing an intermediate
electrolyte layer to separate the hydrovoltaic device into three distinct regions for
investigation and in establishing a systematic equivalent circuit for ion transport
mechanism analysis. However, the manuscript presents the research in a manner that
first describes a large volume of experimental results, then builds the equivalent circuit,
conducts numerical calculations, and finally performs output power testing and
optimization. This structure fails to highlight the study's novelty and advantages,

making it challenging for readers to follow. I recommend reorganizing the presentation
to emphasize the key contributions more effectively.

Thank you for this insightful comment. We appreciate the reviewer's recognition of the
novelty of our approach—namely, the introduction of an intermediate electrolyte layer
that decouples the hydrovoltaic device into three distinct functional regions and the
development of a systematic equivalent circuit to analyze ion transport mechanisms. In
response to the reviewer's concern regarding the clarity and logical flow of the
manuscript, we have significantly reorganized the structure to better highlight the
central contributions and guide the reader through our rationale and findings more
effectively. The revised manuscript now follows a clear and logical progression:

*Introduction of Key Interfacial Processes* – We begin by identifying the central
interfacial phenomena that govern the device's behavior. This section lays the
groundwork for understanding how each process can be modulated to affect the voltage
generation.

*Experimental Platform of the Decoupled System* – Here we introduce the device
architecture and experimental platform that enable the spatial and functional separation
of evaporation, ion transport, and charge generation processes.

*Effects of Temperature Change on Chemical Equilibrium* – We analyze how
temperature alters interfacial surface charge dynamics, leading to observable changes in
voltage.

*Effect of Irradiation on Chemical Equilibrium* – This section examines how light
influences chemical equilibrium and modulates voltage output.

*Photocharging of Charged Interfaces* – We present detailed experiments that show how
capacitive photocharging varies with salinity and illumination, providing critical insight
into the photovoltaic contribution.

*Decoupling Strategy and Rationale for Equivalent Electrical Circuit* – Based on the
physical and experimental insights, we introduce the equivalent circuit model,
explaining how it maps to each functional component of the device.

*Power Output Analysis Using Equivalent Electrical Circuit* – We then use the model to
analyze voltage and power output, showing how the interplay between physical
parameters can be used to predict and optimize device performance.

We believe this restructuring improves the overall readability of the manuscript and
ensures that the novelty and logic of our approach are communicated clearly and
effectively.

2. Lack of conciseness in content. In line with the previous point, the manuscript
includes a substantial number of experiments to support the proposed model. However,
this results in an overly lengthy and complex presentation that compromises clarity.
Thus, I suggest moving non-essential content to supplementary materials to enhance
the manuscript's conciseness. For example, the fabrication of hydrovoltaic devices using
silicon nanostructures has been previously reported, so details on device preparation
and material characterization could be transferred to the supplementary information
(SI). Similarly, in the discussion of temperature and illumination effects on open-circuit
voltage, the introduction of electrolyte concentration as a variable appears unnecessary.
Please clarify its necessity for explaining the mechanisms, or consider placing it in the
SI.

Thank you for this constructive suggestion. We appreciate the reviewer's emphasis on
improving the clarity and conciseness of the manuscript. In response, we have carefully
restructured the content to streamline the presentation to highlight the novelty and
main findings of our work. Specifically, we have moved the detailed *fabrication*
*procedure of the core–shell silicon nanopillar arrays*, which has been previously
reported, to the Supplementary Information (SI) to avoid redundancy. Additionally, we
have clarified in the main text the *importance of electrolyte concentration (salinity)* in
modulating the photocharging effect (**Section 3.3**), particularly its impact on
interfacial capacitance and the resulting photovoltage. This justification supports its
inclusion in the analysis of *photocharging effects*. We believe these revisions
significantly improve the readability and focus of the manuscript while preserving the
key mechanistic insights.

3. In Section 2.1, the device contains only 250 μL of electrolyte. As the temperature
rises, how can it be proven that the performance variations result from temperature
changes rather than the rapid evaporation of the solution, which could lead to increased
ion concentration?

We thank the reviewer for this important observation. Indeed, the small volume of
electrolyte (250 μL) used in the device raises a valid concern regarding the potential
impact of evaporation-induced ion concentration during heating. To address this, we
performed repeated heating–cooling cycles while simultaneously monitoring both the
temperature and the open-circuit voltage (V_{OC}) over time. *Please note that the*
*electrolyte was not replenished during subsequent cycles.*

Our results, now shown in **Figure R18**, demonstrate that although some water loss is
inevitable during prolonged heating, the observed voltage variations are predominantly
governed by temperature-induced effects rather than changes in ion concentration.
Specifically, we note the following key observations:

- 1. *Reproducibility Across Heating–Cooling Cycles:* The voltage–time traces show
consistent trends across multiple heating and cooling cycles. This indicates that
the dominant mechanism affecting V_{OC} is not the gradual change in ion
concentration, but rather the direct and reversible influence of temperature on
interfacial properties, particularly surface charge modulation of the silicon
nanopillar (SiNP) electrode.
- 2. *Small Drift in V_{OC} Amplitude:* While a slight decrease in the peak voltage is
observed in subsequent cycles—likely due to minor evaporation-induced changes
in ionic strength—this effect is secondary. The primary temperature-induced
trend in V_{OC} remains clearly visible and consistent, suggesting that the system’s
performance is not strongly dependent on cumulative electrolyte loss over the
short-term experimental window.
- 3. *Direct Correlation Between Temperature and V_{OC} :* We observe a strong and
repeatable correlation between temperature rise and voltage increase within each
cycle, reinforcing the conclusion that the thermally modulated surface chemistry
is the principal driver of voltage generation.

Figure R18: Voltage with temperature during heating and cooling phase.

4. In Section 3.1, Figure S1 suggests that the introduction of electrodes significantly
 affects the evaporation rate. However, considering that the diameters of the introduced
 electrodes (0.25 mm, 1 mm) are relatively small compared to the original evaporation
 area (~11.3 mm in diameter), how do such small electrodes achieve a twofold or fourfold
 increase in the evaporation rate? Please provide a more detailed explanation.

We appreciate the reviewer's insightful observation. We agree that the *geometric cross-*
 *section* of the electrodes is small relative to the bulk free-surface area. However, the
 observed enhancement in evaporation rate arises not from simple surface coverage but
 from the *physicochemical enhancement* due to the liquid meniscus along the electrode.

1. *Meniscus-Enhanced Flux:*

When an electrode is inserted, a thin liquid film (meniscus) forms along its
 entire *vertical length* ($L_{3\phi}$). Evaporation from this film domain—denoted E_{film}
 can significantly exceed the bulk evaporation flux, as shown in prior studies (e.g.,
 *Sci. Rep.* 12, 1087 (2022)).

2. *Physicochemical Effects:*

Beyond geometry, the curved meniscus region concentrates thermal and mass-
 transfer gradients:

- Thinner films exhibit lower thermal resistance and higher local temperature at the liquid–air interface, boosting evaporation.
- Steeper vapor-pressure gradients exist near the three-phase contact line, further accelerating local mass flux.

Analytical formulation for the evaporation rates from the bulk and the meniscus region:

$$\frac{dm_{film}}{dt} = \pi D \int_0^{L_{3\phi}} \rho E_{film}(y) dy$$

$$\frac{dm_{bulk}}{dt} = \rho E_{bulk} A$$

We have revised **Figure S5** to incorporate this clarification and added relevant literature to support this mechanism. The updated text is highlighted in red in the manuscript.

Figure S5: Measured evaporative flux of water in different conditions. The change in mass due to evaporation in an ambient environment was recorded using a microbalance. The Hydrovoltaic cell was placed in the microbalance, then the mass change was monitored in different conditions. i) No electrode was placed in the cell (black curve), ii) 0.25 mm diameter ($W=0.8$ mm) electrode (red curve), iii) 1 mm diameter ($W=3.2$ mm) electrode (blue and green curves). The inset shows the meniscus region with perimeter W and length $L_{3\phi}$.

5. Ambiguity in the presentation of certain experimental data Figures. Figures 3c and 5e
simultaneously use both line and bar graphs for the same dataset, making the intended
message unclear. In Figure 5b, two separate y-axes represent output power, which is
confusing and may mislead readers.

We thank the reviewer for pointing out these important concerns regarding figure
clarity.

Figures 3c and 5e: We agree that the simultaneous use of both line and bar plots for the
same dataset created unnecessary visual complexity. To improve clarity, we have revised
both figures to display the data using a single, consistent format. Specifically, we
retained the line-symbol plot representation, which more clearly emphasizes the
comparative nature of the results, and removed the overlapping bar plots.

Figure 5b: We acknowledge that the use of dual y-axes for similar units (output power)
can lead to confusion. We have revised this figure to consolidate the data into a single y-
axis. We have also clarified the axis labels and provided a more detailed explanation in
the figure caption.

These revisions aim to enhance readability and eliminate any ambiguity in data
interpretation. The updated figures and captions are included in the revised manuscript.

6. Missing or incorrect legends in some experimental data Figures. In Figure 2c, the plot
lines and legend color blocks do not correspond correctly. Figure 2e lacks legends. Figures
3a and 3b also lack sufficient labeling. In Figures 5c and 5d, the temperature indicators
are unclear. Please refine these elements for improved clarity.

We appreciate the reviewer's careful attention to the clarity and accuracy of our figure
labeling and legends. We have thoroughly reviewed and revised the relevant figures to

ensure proper labeling, legend accuracy, and alignment between graphical elements and
their descriptions.

Figure 2c: We corrected the mismatch between plot line colors and legend blocks so that
they now correspond accurately.

Figure 2e: A clear legend has been added to identify each plotted data series.

Figures 3a and 3b: We have added labels directly on the curves or adjacent to key
features and improved axis titles to clearly indicate units and variable names.

Figures 5c and 5d: The temperature indicators were ambiguous in the original version.
We have now added precise annotations, color-coded legends, and temperature units to
clarify the plotted values and their significance.

These adjustments were made to enhance clarity and prevent any potential
misinterpretation. The updated figures and captions are included in the revised
manuscript.

7. Lack of explanation for certain schematic elements. In Figure 1, the meaning of the
circular dots is not explained. Additionally, the colors of the dots in Figures 1d and 1a
are different without explanation. In Figure 4d, the significance of the red-white
gradient color bar is not clarified.

Thank you for highlighting these critical points regarding the schematic representations.
In response to the reviewer's helpful observation, we have carefully revised the
manuscript and figure captions to improve clarity:

In **Figure 1**, we have added explanations for the circular dots in the schematic. These
dots represent bound surface charge, ions or charge carriers involved in the formation of
the electric double layer (EDL) or surface charge redistribution.

We have standardized the color coding of the dots across different figures.

In **Figure 4A**, we have added a detailed explanation of the red-white gradient color bar,
which illustrates that the red-white gradient represents the filling of surface states
leading to the Fermi level pinning.

All these clarifications are now clearly described in the figure legends and main text,
ensuring that readers can interpret the schematic and graphical data without ambiguity.

8. Missing scale bars and unclear indicators in some Figures. Figure 1b lacks a scale bar.
In Figure 1c, the meaning of the arrows and horizontal lines is unclear.

Thank you for pointing this out. We have added appropriate scale bars to Figure 1b to
provide a clear spatial context for the visualized nanostructures.

In Figure 1c, we have clarified the meaning of the arrows and horizontal lines:

“Scanning transmission electron microscopy (STEM) image of a single NP. The cross-
sectional cut of a single NP reveals the presence of a silicon core and Al₂O₃ shell.
Intensity mapping performed in the rectangular region (see Figure S1).”

To further aid clarity, we have included an annotated schematic in the Supplementary
Information (SI 1), where the function and significance of each graphical element are
explicitly explained.

*Figure S1: A) Scanning Transmission Electron Microscopy image of a single Nanopillars surrounded by Al₂O₃ shell.*
*B) Line scan of the intensity to measure the thickness of the shell, which is around 10 nm.*

9. The reference in Figure 5h is incorrect. Reference 14 describes a device that does not
use a silicon nanowire system but rather a silica nanosphere system. Please double-

check and ensure the citations are accurate.

We thank the reviewer for identifying this error. Upon re-examination, we confirm that
Reference 14 indeed describes a silica nanosphere-based system, not a silicon nanowire
system. This citation was incorrectly included in the comparison presented in Figure 5h.

We have taken the following steps to correct this:

Replaced reference 14 with a correct citation in Figure 5h.

**Reviewer #4 (Remarks to the Author):**

I co-reviewed this manuscript with one of the reviewers who provided the listed reports.
This is part of the Nature Communications initiative to facilitate training in peer review
and to provide appropriate recognition for Early Career Researchers who co-review
manuscripts.

We appreciate the constructive feedback provided, and we have addressed all comments
in detail.

**Response to reviewers**

We thank the reviewers for their valuable comments and constructive suggestions.
We have carefully revised the manuscript in accordance with the feedback provided.
Below, we provide point-by-point responses to each comment. Reviewer comments
are in **black**; our responses are in **blue**, and changes made to the manuscript are
indicated in **red** in the revised text.

**Reviewer #1 (Remarks to the Author):**

The authors have taken good efforts in revising the manuscript. I have no further
input.

We sincerely thank the reviewer for their time and positive feedback. We appreciate
the acknowledgment of our efforts in revising the manuscript.

**Reviewer #2 (Remarks to the Author):**

I thank the authors for responding to my earlier comments in detail. I still think that
the analysis of the coupled system investigated is quite complex and am unsure of the
extent to which “the unravelling of the complex interplay of different interfacial
phenomena in 410 determining the hydrovoltaic device output” was accomplished in
the present work. As indicated in the paper, there are thermal phenomena, interfacial
chemistry, electrochemical aspects, and in addition light-induced effects the sum
total influence of which is truly hard to understand the coupled interactions and the
related simple linear models.

We understand the reviewer concern however we need to disagree with the
considerations and conclusions. Our work in fact deconstructs the complexity
underlying hydrovoltaic devices, including thermal effects, interfacial chemistry,
electrochemical processes, and light-induced responses, and reveals the key factors
and coupling phenomena that control the device response. In particular, light and
heat inputs are present in practical applications, and it is essential to explore how
they can enhance performance instead of hindering it.

More specifically, in this work, we clearly identify two key interplays: (1) the coupling
of three capacitances at the solid/liquid interface, shown in **Figure 1E**, which

regulates the surface charge and, (2) the coupling of the solid/liquid interface with
the liquid/vapor interface, shown in **Figure 5**, regulated by the transfer capacitance,
which controls the ion migration from bottom to top electrode and thus the
evaporation-induced potential. Contrary to all prior literature, these effects are
thoroughly characterized and modelled in relation to the physical properties of the
materials, the solution, and the light and heat inputs.

Importantly, we must emphasize that the complex and coupled interactions at play in
the system are **not described in our work by simple linear models**. Instead,
we uniquely capture both the highly non-linear phenomena, i.e., the chemical
equilibrium at the interface, and their non-linear interplay. For example, in Section
3.3 of the manuscript, we elaborate on the interconnected effects of heat and light,
particularly due to the capacitive effect. We show that the **photovoltage depends**
**non-linearly on the light intensity (Eq. 3)**, influencing the surface charge. **The**
**surface charge**, in turn, **alters in a non-linear manner the chemical**
**equilibrium on the electrolyte side (Eq. 1 and S2)**. Thus, light irradiation has
a highly non-linear impact on the predicted and measured hydrovoltaic device
voltage output.

More generally, as mentioned at the beginning, one of the key factors we identified is
the interplay of the chemical equilibria at the interface, represented in **Figure 1E** by
three coupled capacitances. This coupling leads to a charge regulation at the interface
whenever either the solid or the liquid side of the interface is perturbed, establishing
a dynamic feedback loop, which is highly non-linear.

To better highlight the non-linearity of our model, we have modified the
supplementary section (**SI 2**) to include additional explanations as:

$$56 \quad \sigma = \frac{-e \Gamma}{1 + \frac{10^{-pH} e^{-\frac{e(V_{ox-el} - V_{el})}{k_B T}}}{K_a(T)}} \quad (S2)$$

“...Where, V_{ox-el} is the potential on the oxide–electrolyte interface (**Figure 1 E**), V_{el}
is the bulk potential of the electrolyte, Γ is the surface site density, K_a is the
temperature-dependent equilibrium constant.

The above equation is a more general representation of equation 1 in the manuscript.
The relationship between the surface charge and the potential gradient at the
interface is well-defined by Eq. S2, which links the surface potential, $V_{\text{ox-el}}$, to the
surface charge. It highlights how heat and light are coupled in a highly non-linear
manner through the boundary conditions; specifically, illumination and temperature
modulate the surface charge through the generation of surface photovoltage (i.e., by
changing $V_{\text{ox-el}}$, which depends logarithmically on the light intensity, Eq. 3) and a
change in the dissociation constant (K_a , which depends on temperature in a non-
linear manner, Eq. M1), respectively...”

For added clarity, below we show a quantitative calculation of the silicon–oxide–
electrolyte coupled interfacial capacitances that includes the potential profile, band
bending in silicon, and the electrical double layer. As illustrated in **Figure S20-I**,
our calculations demonstrate the variation in surface charge as a function of the
potential difference between the silicon bulk (V_{si}) and the electrolyte solution (V_{el}).
More details can be obtained in the supplementary information (**SI 20-I**)

Furthermore, band bending influences the maximum photovoltage. We have added a
dedicated section in the supplementary information (**S20-II**) to elaborate on how
the effect of initial band bending (i.e., before illumination) is tied to the generation of
photovoltage and how this, in turn, impacts the surface charge at the solid-liquid
interface in a non-linear manner.

**Figure S20-I: Influence of band bending at the silicon-oxide side on interfacial charge and**
 **electric field regulation.** A) Calculated surface charge at the solid-liquid interface for n-type silicon for
 different dopant concentrations (1/cm³). B) The potential profile across the semiconductor-oxide electrolyte
 system for various bulk potentials of silicon and dopant concentrations. The inset illustrates how changes in
 band bending modify the interfacial potential profile, thereby shifting the chemical equilibrium on the
 electrolyte side. C) Surface charge variation as a function of bulk potential of silicon for various electrolyte
 concentrations. D) Corresponding potential profile for various bulk potentials of silicon and electrolyte
 concentrations.

Finally, we have edited the introductory paragraph of Section 3.4 to further clarify
 the identified and modelled coupled phenomena:

“As illustrated in Figure 5A, the studied system consists of three key components: the
 top electrode, the electrolyte layer, and the bottom nanostructured electrode, each
 defined by its solid/liquid interfaces. At the bottom nanostructured electrode, the
 coupling of the three surface capacitances (Figure 1E) leads to a surface charge
 regulation, whenever either the solid or the liquid side of the interface is perturbed,
 establishing a non-linear dynamic feedback loop. It is essential to recognize that the

interaction of surface charges with ion distribution in the electrolyte layer further
results in a coupling between the bottom and top electrodes. As we outline below, the
equivalent circuit depicted in Figure 5B accurately reflects this multi-level coupling,
capturing the complex behavior of the hydrovoltaic devices.”

Overall, while we agree with the reviewer that the studied system is complex and
involves several interconnected non-linear mechanisms, we believe that our work
significantly advances the understanding of which factors play a key role, including
solid properties, electrolyte concentration, heat, and light inputs, providing guidance
on how to better harness external inputs for improved device performance.

Further, the aspect that “heat and light input have major impacts on the solid/liquid
interface due to the temperature sensitivity of the surface group dissociation and the
ions desorption/adsorption” is expected qualitatively and the modeling is essentially
based on linear approximation/s.

We thank the reviewer for the comment, but, in line with the answer above, we
disagree with the statement that the modelling is essentially based on linear
approximations. In fact, heat and light input strongly affect the solid/liquid interface
through temperature-dependent dissociation of surface groups and ion
desorption/adsorption, which are highly non-linear.

In the context of our equivalent electrical circuit, **the non-linearity is hidden in**
**the different elements affected by the interdependence of solid and**
**liquid properties, as well as light and heat inputs.** In fact, it is crucial to note
that while illumination and temperature may appear as independent factors,
**temperature directly influences the photovoltage element (V_{ph})** included in
the model. The equivalent electric circuit not only serves as an effective framework
for visualizing the essential physical phenomena at play, but it also facilitates a
general understanding of the device's behavior. The equations governing each
element, however, unequivocally preserve the non-linear characteristics of the
phenomena involved, along with their couplings. In particular, as discussed earlier,
the capacitive photocharging phenomenon described in **Section 3.3** is strongly
modulated by parameters such as doping, salinity, oxide layer characteristics, and

temperature. These effects are not represented as a separate capacitor but are
inherently embedded in the magnitude of V_{ph} within the equivalent circuit.

Thus, our model extends beyond a simple linear approximation by capturing the
intertwined influence of thermal and light effects on interfacial charge dynamics. To
clarify this point in the manuscript, we have revised Section 3.4.

Line 441-445: “As highlighted in Section 3.3, critical factors such as doping, salinity,
and the oxide layer non-linearly influence the capacitive photocharging mechanism,
which, in turn, has a profound effect on the magnitude of V_{ph} . Therefore, the strength
and effectiveness of the photocharging capacitive component are fundamentally
encapsulated in the equivalent circuit by the value of V_{ph} , underscoring its crucial
role in this phenomenon^{36,40} (SI 20).”

For instance, I do not consider statements, such as “An increase in ambient
temperature, for example, will lead to a higher temperature of silicon even when no
external heating is applied” insightful. The proposed “decoupling” of the top
evaporative interface with the bottom electrode- through a change of the
“manuscript’s structure” does not do much to enhance the understanding. What
accounts for the

We appreciate the reviewer’s feedback and acknowledge that our earlier phrasing
regarding environmental conditions may not have been sufficiently precise. The
original statement, “An increase in ambient temperature, for example, will lead to a
higher temperature of silicon even when no external heating is applied,” was
intended to highlight the role of ambient conditions in influencing the system
temperature and performance.

To improve clarity, we have revised this passage to:

“A higher ambient temperature has a twofold effect on a hydrovoltaic device. First, as
typically expected, it will increase the evaporation rate and therefore a higher open-
circuit voltage and short-circuit current. Secondly, it will also modify the
temperature of the silicon oxide surface, thereby affecting its chemical potential and
altering the surface charge. This latter effect has been largely overlooked despite its
profound influence on the device performance, shown in this work. Assuming the

device is at equilibrium with its environment, an increase in ambient temperature
from 25 °C to 35 °C can increase the surface charge by ~50% (Figure S2). Thus,
ambient thermal effects can favorably affect hydrovoltaic energy generation in more
complex ways beyond evaporative enhancements.”

This revision emphasizes how this work elucidates more complex interplays between
environmental factors and hydrovoltaic performance than previously expected.

Finally, we emphasize that, as stated explicitly in **Section 2.1**, it is thanks to the
decoupled structured of the studied device that we can identify the two separate key
couplings occurring in the system, i.e., the three-capacitance coupling at the
solid/liquid interface (**Figure 1E**) and the charge transfer capacitance coupling
between the solid/liquid and solid/vapor interfaces (**Figure 5**). Importantly, these
couplings are present in all hydrovoltaic devices. Yet, the studied geometry, which
introduces a thicker electrolyte layer between the solid-liquid and liquid-vapor
interfaces, allows for a better evaluation of their separate contributions and
interplay.

We note that we were not able to address the last sentence of the reviewer’s comment
due to several missing words in the sentence.

New data, such as Fig. R 12, is not particularly meaningful – as only diffusive
behavior seems to have been considered. What about the R_{ct} and the C_{dl}
components? The related analysis is superficial.

We thank the reviewer for addressing this point that closely relates to the chosen
experimental design. In fact, our approach first considers the coupling at the solid-
liquid interface (Figure 1E) and subsequently couples it to the full device behavior
through the charge transfer capacitance (Figure 5B) and an equivalent electrical
circuit. The complete analytical expression for the charge transfer capacitance is
included in the supplementary section (S23), which shows how the properties of the
bottom nanostructured electrode and top electrode affect its values. Importantly, we
have now added a detailed discussion of surface charge regulation supported by a full
numerical calculation (**S 20-I**). By considering the carrier distribution in the space
charge layer and EDL, we analyze how charges and potential vary in response to

perturbations in either the solid or liquid side of the system connected through the
interfacial capacitances (C_{DL} , C_{SC} , and C_{ox}).

In another instance, what is the rationale for indicating that the dV/dT is
proportional to the enthalpy of dissociation for the surface groups – seems to be an
example of correlation and not causation?

We thank the reviewer for bringing this important point to our attention. While this
relationship is derived numerically, it underscores a significant physical dependence
on the key thermodynamic parameters at play.

As described in the manuscript, the surface charge is expressed in terms of the
surface proton concentration, which is related to the potential difference between the
surface (V_s) and the bulk solution (V_{el}) as shown in eq. R1

Furthermore, the temperature dependence of the ionization equilibrium is related to
the thermodynamic parameter-enthalpy (ΔH)-via the Van't Hoff equation (eq. R2)

$$201 \quad \ln K_{a1} - \ln K_{a2} = - \left[\frac{\Delta H}{RT_1} - \frac{\Delta H}{RT_2} \right] \quad (R2)$$

We solved numerically using COMSOL these coupled equations together with the ion
transport within the highly confined liquid region. Despite the high non-linearity of
the charge-regulating equations, these simulations reveal a nearly linear correlation
between dV/dT and ΔH . Therefore, while this finding is grounded in numerical
modeling, it highlights the fundamental physical relationship to the thermodynamics
of the surface and establishes a causation link between the change in different
enthalpies of dissociation of surface groups at the solid/liquid interface and the
change in voltage output.

To clarify this point, we have revised the text to explicitly state that although this
relationship is obtained numerically, it represents a physical dependence on the
governing thermodynamic parameter.

Line 224-225: “While this relationship is derived numerically, it underscores a
significant physical dependence on the key thermodynamic parameters at play.”

What is the basis for fitting the voltage as a function of temperature to the second
order?

We thank the reviewer for this question. The basis for using a second-order fit arises
from both experimental observations and the underlying physical model. The voltage–
temperature curves exhibit clear super-linear behavior. According to our proposed
model, the V_{oc} is expressed as:

$$221 \quad V_{OC} = v_f \bar{\sigma} r_{3\phi} + \Phi (1 + v_f r_{3\phi} C_{tr}) + V_{ph} \quad (R3)$$

As discussed in the manuscript, Φ increases approximately linearly with temperature
(see **S9**), while v_f , which depends on the evaporative flux (Eq. **M3**), also increases
with temperature (see **S24**). Therefore, their product introduces a quadratic
dependence on temperature. This nonlinear contribution justifies fitting the
experimental data with a quadratic function. The fit shows excellent agreement, with
an R^2 value greater than 0.9 (**S12**).

**We note that the prediction of a quadratic dependence of the voltage**
**output with temperature, confirmed by the measurements, highlights the**
**value of our model for understanding and directing the design of**
**hydrovoltaic devices.**

Can the authors indicate more clearly the basis for the 0.25 W/cm² in terms of the
utilized voltage, current, and area involved?

As explained in the Methods section of the manuscript, equations M7-M9 describe
the voltage and current used to obtain the power density, considering a fixed area of 1
236 cm² for our cell design. The current and voltage are:

$$237 \quad I = \frac{\sqrt{A(T, I, c_0)}}{2B(T, I, c_0)} \quad (R4)$$

$$238 \quad V = \frac{\sqrt{A(T, I, c_0)}}{2} \quad (R5)$$

Furthermore, in the supplementary information **S26**, we have detailed the values of
parameters A and B for different conditions. In particular, for the conditions under
which the power density is 0.25 W/m², we have $A=0.35 \text{ V}^2$, and $B=3.5 \text{ k}\Omega$.

**This gives $I = 84.6 \mu\text{A}$, and $V = 0.3 \text{ V}$.**

We have modified the text in the manuscript to clarify this:

**Line 478-481: “By fitting this expression to the experimental data in Figure 6B-C, we**
**have compiled the values of the A and B parameters for a range of different**
**conditions, as shown in Figure S26, which can be used to obtain the current and**
**voltage utilized to obtain the power density.”**

In summary, I do appreciate the immense effort involved in the experimentation.
However, the obtained results still seem to be phenomenological, given the very
nature of the experiment with several coupled phenomena. I am unsure as to how
well the conducted experiment may be reproduced by other scientific groups.

We thank the reviewer for acknowledging the effort involved in the experimentation.
However, we strongly disagree with the outlook for our work. The model we present
is not merely phenomenological. We don't simply fit curves with arbitrary
parameters; instead, we establish clear predictive trends and offer robust physical
interpretations, linking causation to various physical properties of the system. This is
all thoroughly supported by comprehensive numerical calculations and experimental
results. In fact, it is based on key physical relations of interfacial and transport
phenomena, like the chemical equilibrium at the interface or the photovoltage
generation under irradiation. It correctly predicts experimental trends based on the
physical properties of the solid, changes in the temperature and light irradiation (i.e
the quadratic dependence of the voltage output based on the temperature increase or
the change in magnitude and sign of the surface charge and hence output voltage
based on the semiconductor doping), despite the high complexity, non-linearity and
high coupled nature of these phenomena.

Regarding reproducibility, we have demonstrated that our experimental results can
be repeated consistently over time. Furthermore, in the manuscript, we provide
detailed experimental descriptions, including materials, procedures, and analysis
methods, to facilitate replication by other research groups. In addition, the complete
data set will be made available in the appropriate public repository to further support
transparency and reproducibility.

Overall, our primary objective was to capture and describe the effects underlying
complex hydrovoltaic devices to enhance the fundamental understanding. The
capability of our approach to explain all observed experimental trends despite the
complexity of the system and the highly non-linear couplings involved is a testimony
to the value of these results for the broader community.

**Reviewer #3 (Remarks to the Author):**

I have carefully read the authors' responses to the reviewer comments, as well as the
revised version of the manuscript. I am pleased to see that the clarity of the
manuscript has been significantly improved, the logic has become more coherent,
and the authors have provided detailed and convincing responses to the previous
concerns. Based on this, I believe the article can be accepted for publication after
some minor revisions.

We sincerely thank the reviewer for the positive evaluation of our revised
manuscript. We are pleased that the clarifications and improvements have addressed
the previous concerns, and we appreciate the constructive feedback that helped
enhance the clarity and coherence of our work. We will carefully implement the
remaining minor revisions to improve the manuscript further.

My comments are as follows:

Regarding the logical coherence of the article:

1. Although the logical clarity of the manuscript has been greatly improved, there are
still some weak transitions between certain sections. In Section 3.3, the
photocharging phenomenon at the solid–liquid interface is described in detail, along
with the corresponding capacitive mechanism. However, in Section 3.4, when
discussing the equivalent circuit of the silicon–oxide–water interface, the capacitor
component is not reflected. How should this be explained?

We thank the reviewer for this insightful comment. We agree that there could be
some ambiguity in connecting the discussion in **Section 3.3** to the equivalent circuit
presented in **Section 3.4**. To clarify, the capacitive photocharging phenomenon
described in **Section 3.3** is strongly influenced by doping, salinity, and the oxide

layer. In the equivalent circuit, this capacitive effect is not represented as a discrete
capacitor but is inherently reflected in the photovoltage element, V_{ph} .

To address this point, we have revised the manuscript to make this connection more
straightforward. The following sentence has been added to **Section 3.4**:

*Line 446-450: “As highlighted in Section 3.3, critical factors such as doping, salinity,*
*and the oxide layer non-linearly influence the capacitive photocharging mechanism,*
*which, in turn, has a profound effect on the magnitude of V_{ph} . Therefore, the strength*
*and effectiveness of the photocharging capacitive component are fundamentally*
*encapsulated in the equivalent circuit by the value of V_{ph} , underscoring its crucial*
*role in this phenomenon (SI 20).”*

We believe this clarification strengthens the logical transition between Sections 3.3
and 3.4.

We note that we have also now added a new Supplementary Section that
quantitatively addresses the capacitive coupling at the solid/liquid interface (S20-I).
In this section, we show a complete calculation of the silicon–oxide–electrolyte
coupled interfacial capacitances that includes the potential profile, band bending in
silicon, and the electrical double layer. As illustrated in **Figure S20-I**, our
calculations demonstrate the variation in surface charge as a function of the potential
difference between the silicon bulk (V_{si}) and the electrolyte solution (V_{el}).

2. Following the previous question, in the abstract, the main strategies proposed for
enhancing EDHV are “increasing silicon doping” and “switching the dielectric shell
from TiO_2 to Al_2O_3 .” In the equivalent circuit, which parameters correspond to these
strategies, and how should they be interpreted?

We thank the reviewer for this important question regarding the connection between
the strategies highlighted in the abstract and the parameters in the equivalent circuit.
The two main strategies proposed—(1) increasing silicon doping and (2) switching
the dielectric shell from TiO_2 to Al_2O_3 —both affect the initial band bending and
capacitance of the silicon–oxide–electrolyte interface, which **directly influences**
**the magnitude of the photovoltage, V_{ph} , in the equivalent circuit.**

In response to the last two comments, we have added a dedicated section in the
supplementary information (SI 20-I and II) to elaborate on how the effect of initial
band bending (i.e, before illumination) is tied to the generation of photovoltage.

**SI 20-II:** The absorption of light above the semiconductor's band gap generates
electron-hole pairs, effectively separated by the electric field induced by the initial
band-bending in the absence of light. This process significantly decreases band-
bending, ultimately leading to flattening at high intensity. As a result, this
phenomenon culminates in the generation of surface photovoltage. The maximum
photovoltage is directly correlated with the initial band-bending observed in the
dark, which is heavily influenced by the doping levels of silicon. As illustrated in
**Figure S20-II D**, as band-bending levels escalate from $2k_B T$ to $5k_B T$, the saturation
value of photovoltage correspondingly increases. Furthermore, **Figure S20-II B**
demonstrates the relationship between band bending and surface state density
across various doping concentrations. It is evident that low-doped silicon
experiences rapid saturation in band-bending, resulting in a significantly larger
band-bending compared to high-doped silicon. Consequently, this leads to a greater
photovoltage under illumination, clearly explaining why low-doped silicon achieves
larger photovoltage values than its high-doped counterparts. *Note: For simplicity,*
*the analysis presented here doesn't take into account the effect of the electrical*
*double layer.*

**Figure S23: Band structure and photoresponse at the semiconductor-oxide interface.**

(A) Band diagrams before (left) and after (right) equilibration, showing how surface states fill to align

the Fermi levels across the interface. The red–white gradient illustrates progressive state filling during

equilibration. (B) Calculated surface band bending for n-type silicon as a function of surface state

density and dopant concentration. Band bending saturates once the Fermi level approaches the

surface states. (C) Under illumination, splitting of the quasi-Fermi levels for electrons and holes

occurs, leading to band flattening. The separation between these quasi-Fermi levels corresponds to

the measurable photovoltage. (D) Calculated photovoltage as a function of light intensity for different

initial band-bending conditions. The saturation value increases with the magnitude of dark band

bending.

Together, the panels illustrate how surface states govern equilibrium band

alignment, how doping influences band bending, and how illumination modulates

interfacial energetics to generate photovoltage.

I also have some specific questions regarding certain content in the manuscript:

3. Lines 233–236:

“Within the linear regime, we can extract the dV/dT slope and relate it to the

enthalpy of dissociation, confirming the expected trend due to the lower enthalpy
and surface site density (Γ in Eq. 1) of TiO_2 compared to Al_2O_3 (Figure 2D–E).”
I understand the relationship between the dV/dT slope and the enthalpy of
dissociation, but how is this related to the surface site density?

We thank the reviewer for raising this point. We acknowledge that the original
phrasing was misleading. The dV/dT slope is related only to the enthalpy of
dissociation and not to the surface site density (Γ). We have corrected this statement
in the revised manuscript to remove any reference to Γ in this context.

4. Lines 281–283:

“A differential capacitance of the double layer, CDL, can be defined as the change in
surface charge (σ) due to a change in Φ , and mathematically expressed as $C_d =$
$(\partial\Phi/\partial\sigma)^{-1}$, which increases with surface charge (see SI 14 for derivation).”

What is the relationship between CDL and C_d ? Are they the same quantity? What
does "c" represent in SI 14? Why is the Stern layer capacitance (C_s) not reflected in
the equation?

We thank the reviewer for this comment and for catching the inconsistency. The
notation “ C_d ” was a typographical error; it should indeed be C_{DL} throughout. We
have corrected this in both the main manuscript and the supplementary information.

To clarify:

• C_{DL} refers to the differential capacitance of the electrochemical double layer

and is defined as $C_{DL} = \left(\frac{\partial\Phi}{\partial\sigma}\right)^{-1}$

• The Stern layer capacitance C_s was not explicitly included in this equation
because SI 14 focuses on deriving the general expression for the differential
double-layer capacitance from the Gouy–Chapman (diffuse layer) model. In
the full equivalent circuit, however, the total double-layer capacitance would
be the series combination of C_{stern} and the diffuse layer capacitance (C_d).

We believe these corrections and clarifications resolve the confusion.

5. Regarding the capacitance measurement: the authors converted the measured
impedance at a fixed frequency into capacitance using the pure capacitor formula.
What is the basis for assuming that the system exhibits pure capacitive behavior at 1
405 kHz?

We thank the reviewer for this important question. We would like to clarify that we
did not assume purely capacitive behaviour a priori. At 1 kHz, the impedance was
measured by recording both the amplitude and the phase. From these data, the real
(resistive) and imaginary (capacitive) components of the impedance were extracted.
The capacitance was then calculated from the imaginary part of the impedance,
which directly corresponds to the capacitive response of the system at that frequency.
This clarification has been added to the revised manuscript to avoid any ambiguity
regarding our measurement approach.

SI 15: “At 1 kHz, the impedance was measured by recording both the amplitude and
the phase. From these data, the real (resistive) and imaginary (capacitive)
components of the impedance were extracted. The capacitance was estimated using
the imaginary component of impedance (Z'') at the selected frequency, following the
standard relation:”

6. Lines 459–461:

“Notably, the measured power curves indicate that increasing the concentration of
KCl from 1 mM to 100 mM nearly doubles the maximum power output despite a
lower open circuit voltage at higher concentrations.”

According to earlier data (e.g., Figure 2c), the open circuit voltage at 100 mM is
higher than at 1 mM across various temperatures. Why is it described here as “a
lower open circuit voltage at higher concentrations”?

We thank the reviewer for pointing out this inconsistency. Upon revisiting the data
and figure legends, we found that the original legends were somewhat misleading.
After correction, it is clear that the open-circuit voltage at 100 mM is indeed **lower**
compared to that at 1 mM across various temperatures, consistent with the
discussion in the manuscript. We have revised the figure with clearer legends to
make this trend unambiguous.

7. In addition, possibly due to the extensive revision of the manuscript, there are
several instances of inappropriate figure references and formatting issues that could
be optimized. The following points were noted:

- The layout of Figure 3 (with panel b below panel a) is inconsistent with other
figures, where panel b is placed to the right of panel a.

We thank the reviewer for noting this inconsistency. We have modified the layout of
Figure 3 so that panel (b) is placed to the right of panel (a), consistent with the
arrangement in other figures. We believe this improves visual clarity and uniformity
across the manuscript.

- In Figure 3b, only three concentrations are shown, which appears insufficient.
Would adding error bars help improve the reliability of the data?

We thank the reviewer for this suggestion. We have revised Figure 3b to improve
clarity and reliability by including the appropriate error bars for the presented data.

- In Figure 3c, temperature variation is not clearly indicated. Could background
shading be used to distinguish heating and cooling phases?

We thank the reviewer for this suggestion. To improve clarity, we have added
background shading in Figure 3c to distinguish between the heating and cooling
phases. Additionally, we have included explanatory text in the figure caption to
clearly indicate these phases.

- For Figure 3d, since there is only a single y-axis, placing it on the left would be more
conventional.

We thank the reviewer for this suggestion. The y-axis in Figure 3d has been moved to
the left side to follow the conventional placement, improving readability and
consistency with other figures.

- Lines 315–317:

“As shown in Figure 3D (blue bars, blue shaded areas), the capacitance for the low-

doped silicon sample (left two columns) was significantly lower than that of the high-
doped silicon samples (right two columns), in agreement with the expected trend.”
While the meaning is understandable, the phrasing contains redundancy and
inconsistency between “blue bars, blue shaded areas” and “left two columns/right
two columns.”

We thank the reviewer for pointing out the redundancy and inconsistency in the
original phrasing. To improve readability and clarity, the sentence has been revised
as follows:

Line 319-322: “As shown in **Figure 3D (blue ($T_s=25^\circ\text{C}$) and red ($T_s =70^\circ\text{C}$)**
**shaded areas)**, the capacitance for the low-doped silicon sample (left two columns)
was significantly lower than that of the high-doped silicon samples (right two
columns), in agreement with the expected trend.”

- Subscripts are missing in Lines 320 and 458.

This revised phrasing eliminates redundancy and clearly specifies the color coding
and sample grouping.

- In Figure 4c, the symbol $\alpha\lambda$ appears but is not clearly explained in the text. Please
clarify its meaning and purpose.

We thank the reviewer for this observation. We have clarified the meaning and
purpose of the symbol $\alpha\lambda$ in the revised manuscript. The following explanation has
been added:

Line 360-362: “Lastly, we have shown in **Figure 4C (blue line)** the value of β_λ
normalized to the absorption of the sample α_λ (S19) to exclude the variation in the
amount of light reaching the surface for different wavelengths.”

- Line 437: Figure 4d is not present in the manuscript.

We thank the reviewer for pointing out this typo. The reference to Figure 4d has been
corrected in the revised manuscript.

- Line 482: The reference to Figure 3c does not match the content described in the
text.

We thank the reviewer for noticing this discrepancy. The reference has been updated
to point to the correct figure in the revised manuscript.

- Figures 6d and 6f appear to be unmentioned in the manuscript.

We thank the reviewer for pointing this out. References to Figures 6d and 6f have
now been added in the manuscript to ensure all panels are properly discussed.

I understand that such minor oversights are possible in the process of extensive
revisions, but I urge the authors to pay closer attention to these details, especially
those that may affect the reader's understanding of the article.

We thank the reviewer for this constructive feedback and fully acknowledge the
importance of careful attention to detail. We have carefully reviewed the manuscript
to correct all identified inconsistencies, typos, and figure references to ensure clarity
and improve the reader's understanding. We appreciate the reviewer's guidance in
helping us strengthen the presentation of our work.

Reviewer #4 (Remarks to the Author):

I co-reviewed this manuscript with one of the reviewers who provided the listed
reports. This is part of the Nature Communications initiative to facilitate training in
peer review and to provide appropriate recognition for Early Career Researchers who
co-review manuscripts.

We appreciate the thoughtful feedback and constructive comments provided through
this process.

Response to reviewers

We thank the reviewers for their valuable comments and constructive suggestions. We have carefully revised the manuscript in accordance with the feedback provided. Below, we provide point-by-point responses to each comment. Reviewer comments are in **black**; our responses are in **blue**, and changes made to the manuscript are indicated in **red** in the revised text.

Reviewer #3 (Remarks to the Author):

The author has addressed all the concerns I raised in the previous round of review with sufficient and reasonable explanations, which have clarified my doubts. After reviewing, I believe that the manuscript meets the journal's publication standards. Therefore, I recommend the acceptance of the manuscript and look forward to its contribution to the relevant field.

We sincerely thank the reviewer for the positive evaluation and thoughtful feedback throughout the revision process. We appreciate your recommendation for acceptance and are grateful for your constructive comments, which helped us improve the clarity and quality of the manuscript.

Reviewer #4 (Remarks to the Author):

We appreciate the thoughtful feedback and constructive comments provided through this process.

Reviewer #5 (Remarks to the Author):

The authors have made tremendous efforts to reply to referee 2 and referee 3's comments. In particular, they have convincingly answered to the main concerns raised by referee 2 and edited Section 3.4 to summarize the different coupled phenomena. Although the physical coupling is complex, their description sounds reasonable and beyond a mere phenomenological approach. I therefore recommend

publication of the manuscript after the authors have taken into account the following minor corrections:

We thank the reviewer for the positive assessment of our revisions and for recognizing our efforts to address the concerns raised by the referees. We appreciate your thoughtful feedback and will carefully incorporate the suggested minor corrections to further improve the manuscript.

-Eq.2 : please define all terms. A reference may be given to understand where does the factor 8000 come from.

Thank you for this helpful suggestion. In the revised manuscript, we have defined all terms appearing in Eq. 2 for clarity. Additionally, we have added a reference explaining the origin of the factor 8000.

“where, k_B , T_s , e , ϵ_0 , ϵ_r , c_0 are Boltzmann constant, surface temperature of solid, electronic charge, dielectric permittivity of free space, relative permittivity of the medium and molar concentration of bulk electrolyte respectively. σ is the surface charge and C_{Stern} is the capacitance of the Stern layer³⁵.”

-Does the evaporation velocity depend on the electrolyte concentration or is it taken to be that of the pure fluid ? Please clarify this point in the main text.

Thank you for pointing out this issue. The evaporation velocity could, in principle, depend on the salt concentration; however, such effects lie beyond the scope of the present study. To clarify this in the manuscript, we have revised the text as follows:

“The term I_{Evap} is equal to total charge times v_f , which is equal to the evaporative mass flux of the fluid (**SI 5 and SI 24**) divided by density of water.”

-Eq. 4: please replace $r_{3\phi}$ by $R_{3\phi}$ to keep consistent notations

Thank you for this comment. We originally used the lowercase $r_{3\phi}$ to emphasize the removal of the meniscus width dependence from the open-circuit voltage expression. However, to avoid confusion and improve consistency in the notation, we have clarified this point in the text as follows:

“The resistive element $R_{3\phi} = \frac{r_{3\phi}}{W}$, where W is the perimeter of the meniscus region (see Methods)....”